# The fate of pyruvate dictates cell growth by modulating cellular redox potential

Ashish G Toshniwal[1], Geanette Lam[1], Alex J Bott[1], Ahmad A Cluntun[1†], Rachel Skabelund[1], Hyuck-Jin Nam[2], Dona R Wisidagama[2], Carl S Thummel[2], Jared Rutter[1,3*]

[1]Department of Biochemistry, University of Utah, Salt Lake City, United States; [2]Department of Human Genetics, University of Utah, Salt Lake City, United States; [3]Howard Hughes Medical Institute, University of Utah School of Medicine, Salt Lake City, United States

*For correspondence:
rutter@biochem.utah.edu

Present address: †Department of Biochemistry & Molecular Biology, Rutgers Robert Wood Johnson Medical School, New Jersey, United States

Competing interest: The authors declare that no competing interests exist.

## eLife Assessment

This **fundamental** work demonstrates that compartmentalized cellular metabolism is a dominant input into cell size control in a variety of mammalian cell types and in Drosophila. The authors show that increased pyruvate import into the mitochondria in liver-like cells and in primary hepatocytes drives gluconeogenesis but reduces cellular amino acid production, suppressing protein synthesis. The evidence supporting the conclusions is **compelling**, with a variety of genetic and pharmacologic assays rigorously testing each step of the proposed mechanism. This work will be of interest to cell biologists, physiologists, and researchers interested in cell metabolism, and is significant because stem cells and many cancers exhibit metabolic rewiring of pyruvate metabolism.

**Abstract** Pyruvate occupies a central node in carbohydrate metabolism such that how it is produced and consumed can optimize a cell for energy production or biosynthetic capacity. This has been primarily studied in proliferating cells, but observations from the post-mitotic *Drosophila* fat body led us to hypothesize that pyruvate fate might dictate the rapid cell growth observed in this organ during development. Indeed, we demonstrate that augmented mitochondrial pyruvate import prevented cell growth in fat body cells in vivo as well as in cultured mammalian hepatocytes and human hepatocyte-derived cells in vitro. We hypothesize that this effect on cell size was caused by an increase in the NADH/NAD$^+$ ratio, which rewired metabolism toward gluconeogenesis and suppressed the biomass-supporting glycolytic pathway. Amino acid synthesis was decreased, and the resulting loss of protein synthesis prevented cell growth. Surprisingly, this all occurred in the face of activated pro-growth signaling pathways, including mTORC1, Myc, and PI3K/Akt. These observations highlight the evolutionarily conserved role of pyruvate metabolism in setting the balance between energy extraction and biomass production in specialized post-mitotic cells.

## Introduction

Cells must appropriately allocate available nutrients to optimize their metabolic programs—for energy extraction and for the generation of building blocks that enable cell growth. The balance between these processes not only maintains cellular health under varying nutritional conditions but also plays an important role in determining cell fate and function (*Baker and Rutter, 2023*; *DeBerardinis and Thompson, 2012*; *Ghosh-Choudhary et al., 2020*; *Metallo and Vander Heiden, 2013*). For example, hepatocyte metabolism changes considerably between fed and fasted conditions. In the fed state, hepatocytes use the majority of their nutrients to synthesize proteins, lipids, and glycogen, which

**eLife digest** Cells require nutrients to produce energy and build vital biomolecules, such as nucleic acids, proteins, lipids and carbohydrates to support organ functions. An imbalance of nutrients can significantly affect cell growth and behavior, which has been seen in diseases like cancer, diabetes and heart failure.

Pyruvate is a key product of sugar metabolism. It can enter cell components called mitochondria to fuel energy production or be converted into lactate and other molecules to support the production of essential molecules (called biosynthesis) and maintain redox balance (i.e., the right proportion of electron carriers).

The mitochondrial pyruvate carrier (MPC) controls how much pyruvate enters the mitochondria, and changes in its activity have been linked to abnormal cell growth and impaired heart function. Most of our understanding of this balance comes from studies of dividing cells, but it is less clear how pyruvate affects cells that no longer divide, such as liver cells or hepatocytes in the liver.

To determine whether pyruvate metabolism directly influences biosynthetic capacity and growth of cells, Toshniwal et al. studied the fat body of fruit fly larvae, a metabolic organ analogous to the mammalian liver. The results showed that pyruvate metabolism affects cell size through changes in the redox balance of a cell.

In genetically modified cells that overexpressed MPC, more pyruvate entered the mitochondria. This restricted the growth of cells and limited protein synthesis in the fat body cells of fruit fly larvae. Restricted cell growth was also observed in spheroids derived from human liver cancer cells and in primary rat hepatocytes.

Further experiments revealed that the restricted cell growth was not the result of engaging the normal growth signaling pathways. Instead, it was due to changes in the way cells use pyruvate in the mitochondria. Rather than using it for energy production, the cells redirected it toward gluconeogenesis, producing and releasing glucose. This shift was caused by a more reduced redox state in the cell (more precisely, an imbalance in $NADH/NAD^+$ levels).

Because of this redox shift, the cells made fewer amino acids, and with fewer building blocks for proteins, the cells could not make as much protein, limiting their growth and size.

These results demonstrate how specialized cells, such as hepatocytes, employ distinct metabolic adaptations to meet the organismal demands under varying nutritional conditions. The fate of pyruvate could profoundly influence the redox state and gluconeogenic capacity of the mammalian liver. Understanding these mechanisms may eventually benefit patients with metabolic disorders, such as diabetes. Until then, further studies are needed to translate these findings from model systems to human physiology and to evaluate the safety and efficacy of targeting these pathways in living organisms.

---

results in increased cell size and liver biomass (*Kast et al., 1988*; *Reinke and Asher, 2018*; *Sinturel et al., 2017*). To meet the energy demands of other tissues during fasting, the liver undergoes a metabolic shift to produce glucose from biosynthetic precursors, thereby decreasing hepatocyte and liver size (*Lang et al., 1998*; *Sinturel et al., 2017*).

The metabolic pathways that support biosynthetic metabolism can be inappropriately activated in diseases such as cancer and heart failure to promote pathological growth. For example, many of the primary oncogenic adaptations in tumors prioritize anabolic metabolism over ATP production, which facilitates rapid cell proliferation (*Lunt and Vander Heiden, 2011*; *Vander Heiden et al., 2009*; *Zhu and Thompson, 2019*). Metabolic rewiring during heart failure similarly results in greater biosynthetic potential and less efficient ATP production, which in post-mitotic cardiomyocytes leads to increased cell size and insufficient cardiac pumping (*Bornstein et al., 2024*; *Henry et al., 2024*; *Weiss et al., 2023*).

The fate of pyruvate, which is primarily generated from glucose via glycolysis in the cytoplasm, is a critical node that can determine the balance between energetic and biosynthetic metabolism (*Baker and Rutter, 2023*; *Yiew and Finck, 2022*). In most differentiated cells, the majority of pyruvate is transported into the mitochondria via the mitochondrial pyruvate carrier (MPC) complex, a heterodimer composed of MPC1 and MPC2 proteins (*Bricker et al., 2012*; *Herzig et al., 2012*). Once

in the mitochondria, pyruvate is converted to acetyl-CoA by the pyruvate dehydrogenase complex (PDH), fueling the tricarboxylic acid (TCA) cycle and supporting efficient ATP production. In cancer, stem, and other proliferative cells, more pyruvate is converted to lactate and exported from the cell, a process that regenerates $NAD^+$, a cofactor necessary for glycolytic flux (*Baker and Rutter, 2023*; *Lunt and Vander Heiden, 2011*; *Zhu and Thompson, 2019*). In some specialized cells, including hepatocytes, pyruvate is imported into mitochondria but converted to oxaloacetate, which can feed the TCA cycle but also serves as a precursor for glucose production via gluconeogenesis (*Holeček, 2023b*; *Jitrapakdee et al., 2008*). We and others have demonstrated that these alternative fates of pyruvate—energy generation, cell proliferation, or glucose production—differentially impact the metabolic and fate decisions of multiple cell types in varying contexts (*Bensard et al., 2020*; *Cluntun et al., 2021*; *Wei et al., 2022*; *Yiew and Finck, 2022*). Loss of MPC function, which shifts pyruvate metabolism toward lactate production and thus expedites glycolysis and the production of biosynthetic precursors, has been shown to increase cell proliferation in mouse and *Drosophila* intestinal stem cells as well as in various tumors (*Bensard et al., 2020*; *Schell et al., 2017*; *Zangari et al., 2020*). MPC expression is also reduced in human and mouse models of heart failure, and genetic deletion of the MPC in cardiomyocytes is sufficient to induce hypertrophy and heart failure (*Cluntun et al., 2021*; *Fernandez-Caggiano et al., 2020*; *McCommis et al., 2020*; *Zhang et al., 2020*). Conversely, MPC overactivation or overexpression restricts intestinal stem cell proliferation and limits the growth of cardiomyocytes under hypertrophic stimuli, with excess mitochondrial pyruvate fueling the TCA cycle (*Bensard et al., 2020*; *Schell et al., 2014*; *Schell et al., 2017*). These observations suggest that mitochondrial pyruvate metabolism is central to cell proliferation as well as the size and specialized functions of post-mitotic cells.

The mechanisms regulating cell size include well-characterized cellular signaling pathways and transcriptional programs (*Grewal, 2009*; *Liu et al., 2022*; *Lloyd, 2013*). The CDK4–Rb pathway monitors cell size in proliferating cells by coupling cell growth with cell division (*Amodeo and Skotheim, 2016*; *Ginzberg et al., 2018*; *Tan et al., 2021*; *Zhang et al., 2022*). In response to insulin and other growth factors, the PI3K/Akt pathway activates mTORC1, leading to increased biosynthesis of proteins, lipids, and nucleotides, and consequently, increased cell size (*Gonzalez and Rallis, 2017*; *Gonzalez and Rallis, 2017*; *Saxton and Sabatini, 2017*). The pro-growth transcription factor Myc drives a gene expression program that enhances metabolic activity and protein synthesis, resulting in larger cells (*Baena et al., 2005*; *Dang, 1999*; *Iritani and Eisenman, 1999*; *Stine et al., 2015*; *van Riggelen et al., 2010*). However, we only partially understand how metabolic pathways regulate physiological (or pathophysiological) growth, particularly in cells that have distinct and specialized roles in organismal metabolism.

Here, we investigated whether pyruvate metabolism influences biosynthetic capacity and cell size, using the *Drosophila* fat body as a model of the mammalian liver. We found that MPC overexpression and increased mitochondrial pyruvate transport restrict cell growth and limit protein synthesis in larval fat body cells and in spheroids of human liver-derived cells. Higher MPC expression resulted in smaller cells, not by increasing TCA cycle flux as observed in other cell types, but instead by redirecting mitochondrial pyruvate metabolism toward gluconeogenesis. A key driver of this metabolic rewiring is a reduced cellular redox state, which disrupts the biosynthesis of TCA cycle-derived amino acids, such as aspartate and glutamate, ultimately reducing protein synthesis. These observations highlight how cells with specialized functions, like hepatocytes, employ distinct metabolic adaptations to respond to organismal demands under varying nutritional conditions.

## Results

### Increased mitochondrial pyruvate transport reduces the size of *Drosophila* fat body cells

The *Drosophila* fat body is functionally analogous to mammalian white adipose tissue and liver, serving as a buffer to store excess nutrients in fat droplets and glycogen and deploy them to support the animal with fuel during times of fasting (*Arrese and Soulages, 2010*; *Musselman et al., 2013*). During larval development, fat body cells halt cell division and dramatically increase in size during the third instar stage (from 72 to 120 hr After Egg Laying (AEL)) (*Edgar and Orr-Weaver, 2001*; *Zheng et al., 2016*; *Figure 1a–c*). As a first step toward understanding the metabolic programs that enable

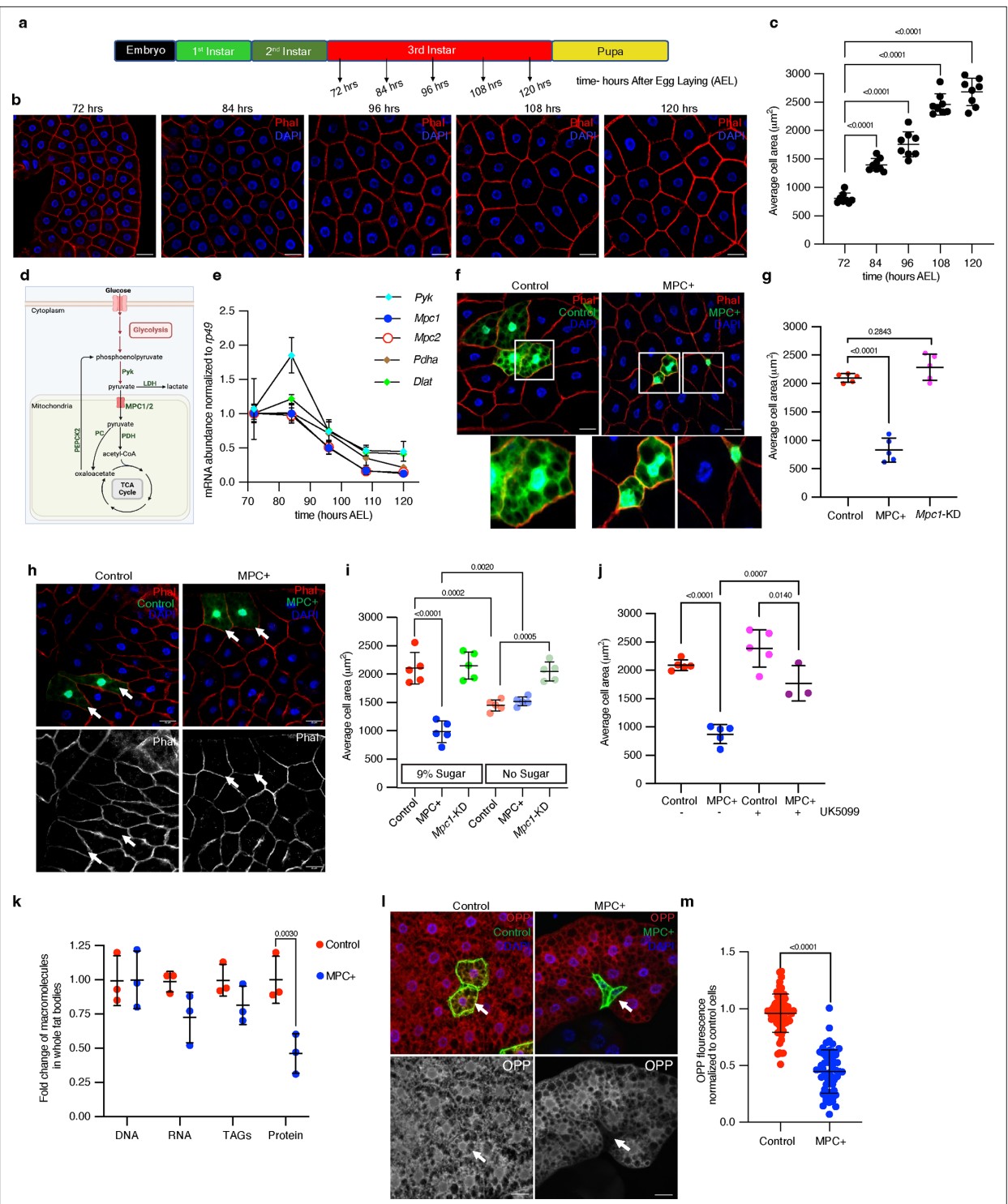

**Figure 1.** Increased mitochondrial pyruvate transport reduces size of *Drosophila* fat body cells. (**a**) A schematic representation of *Drosophila* developmental stages with specified time points (hours after egg laying (AEL) at 25°C) at which larvae were dissected to collect fat bodies. (**b**) Representative images of larval fat bodies at the indicated times (hours AEL) stained with rhodamine phalloidin to visualize cell membranes and DAPI to visualize DNA. The scale bar represents 25 μm. (**c**) Quantification of fat body cell area based on rhodamine phalloidin stained cell membranes at the indicated time points. Data are presented as mean ± standard deviation (s.d.) from six biological replicates, with each replicate averaging the size of 50 randomly selected cells from fat bodies dissected from five male larvae. (**d**) A schematic of pyruvate metabolism. In the cytoplasm, pyruvate is a product of glycolysis, synthesized by pyruvate kinase (Pyk) or from lactate via lactate dehydrogenase (LDH). Pyruvate is transported into mitochondria by the mitochondrial pyruvate carrier (MPC) complex. Within mitochondria, pyruvate is converted into acetyl-CoA by pyruvate dehydrogenase (PDH) or into

*Figure 1 continued on next page*

*Figure 1 continued*

oxaloacetate by pyruvate carboxylase (PC), both of which are substrates for the TCA cycle. PEPCK2 converts oxaloacetate into phosphoenolpyruvate. (**e**) *Pyk*, *Mpc1*, *Mpc2*, *Pdha*, and *Dlat* transcripts were quantified from larval fat bodies collected at the indicated times. (**f**) Representative confocal microscope images of phalloidin- and DAPI-stained fat bodies with flip-out Gal4 clones expressing MPC1-P2A-MPC2 (MPC+), marked with GFP at 120 hr AEL. The images at the bottom show magnified insets of GFP-positive cells in control and MPC-expressing clones. (**g**) Quantification of GFP-positive clonal cell area with the indicated genetic manipulations—control, MPC+, and *Mpc1*-KD. Data are presented as mean ± s.d. from six biological replicates, with each set averaging the size of 20 clonal cells from fat bodies collected from five male larvae. (**h**) Images showing control and MPC+ clones from larvae fed on no sugar diet. (**i**) Quantification of the area of control, MPC+ and *Mpc1*-KD fat body clonal cells from larvae fed on a diet containing either 9% sugar or no sugar. (**j**) Quantification of the area of MPC+ fat body clonal cells from larvae fed a diet supplemented with or without 20 μM UK5099. (**k**) Fold change of the abundances of the indicated macromolecules in fat bodies expressing MPC (MPC+) in all fat body cells using CG-Gal4. The abundance of each individual macromolecule is normalized to that of the respective macromolecular abundance in GFP-expressing, control fat bodies. Data are represented as mean ± s.d. from three biological replicates with fat bodies collected from 10 larvae at 120 hr AEL. (**l**) Representative images of fat body clones stained with O-propargyl-puromycin (OPP, 20 μM for 30 min), showing control and MPC expressing GFP-positive cells. The top panels show GFP-positive clones and OPP staining in red, while the bottom panels show respective OPP channel. Arrows indicate cells with the specified genetic manipulation. The scale bar represents 20 μm. (**m**) Quantification of OPP fluorescence intensity of control or MPC+ fat body cells compared to neighboring non-clonal cells. Data are presented as mean ± s.d. from six biological replicates, with each set averaging the size of 20 clonal cells from fat bodies collected from five male larvae. Unpaired *t*-tests or one-way ANOVA tests were performed to evaluate the statistical significance of the data, with p-values mentioned in the graphs where significance was noted. Panel d was created with BioRender.com.

The online version of this article includes the following source data and figure supplement(s) for figure 1:

**Source data 1.** Source data related to *Figure 1*.

**Figure supplement 1.** Transcriptomic changes in fat body during larval development.

**Figure supplement 1—source data 1.** Source data related to *Figure 1—figure supplement 1*.

**Figure supplement 2.** Increased mitochondrial pyruvate transport reduces size of *Drosophila* fat body cells.

**Figure supplement 2—source data 1.** Source data related to *Figure 1—figure supplement 2*.

this rapid cell growth, we performed RNA sequencing of the *Drosophila* fat body across this developmental period. We observed a time-dependent change in mRNAs encoding proteins that have well-characterized roles in supporting cell growth, including components of the insulin and mTORC1 signaling pathways and the Myc transcriptional network, which correlated with increased cell size (*Figure 1—figure supplement 1a*). Among metabolic genes, we observed modest differences in those that function in amino acid synthesis and fatty acid metabolism (*Figure 1—figure supplement 1b*). The abundances of mRNAs encoding proteins involved in glycolysis, oxidative phosphorylation (OxPhos), and the TCA cycle were distinctly altered in fat bodies during development (*Figure 1—figure supplement 1b*). Since pyruvate metabolism is the central node connecting carbohydrate metabolism and the TCA cycle, we studied the abundances of mRNAs that encode proteins that regulate pyruvate metabolism (*Figure 1d*). We found that the expression of genes that link pyruvate to the TCA cycle was reduced, including *Pyk*, the *Drosophila* pyruvate kinase homolog, which converts phosphoenolpyruvate into pyruvate. *Mpc1* and *Mpc2*, which encode the two subunits of the MPC, which transports pyruvate into the mitochondrial matrix; as well as *Pdha* and *Dlat*, which encode subunits of PDH complex (*Figure 1e*). In contrast, some mRNAs encoding proteins that regulate pyruvate abundance were upregulated, such as *Pepck* and *Pepck2*, which make phosphoenolpyruvate from oxaloacetate (*Figure 1—figure supplement 1c*). Based on these data, we hypothesized that the suppression of mitochondrial pyruvate metabolism, which is gated by the action of the MPC, might support the rapid cell growth observed in fat body cells.

To test this hypothesis, we prevented the downregulation of the MPC during larval development (*Figure 1e*) by ectopically expressing *Mpc1* and *Mpc2* (termed 'MPC+' *Figure 1—figure supplement 2a and b*). The sustained expression of the MPC (both *Mpc1* and *Mpc2*) throughout the fat body significantly reduced the rate of cell growth compared to a GFP-expressing control (*Figure 1—figure supplement 2c*). Given the important role of the fat body in controlling organismal growth, we wanted to assess the cell-autonomous effects of MPC expression using mosaic analysis in individual fat body cells (*Figure 1—figure supplement 2d*; *Ito et al., 1997*). We generated GFP-labeled clones with expression of the MPC (MPC+), which we confirmed by immunofluorescence (*Figure 1—figure supplement 2e, f*). The MPC-expressing, GFP-positive cells were significantly smaller in size compared with either mock clones (control) or neighboring GFP-negative cells within the same tissue. *Mpc1* knockdown (*Mpc1*-KD) clones, in contrast, were marginally larger (*Figure 1f, g*). These results

demonstrate that sustained expression of the MPC in developing fat body cells is sufficient to prevent cell growth in a cell-autonomous manner.

If the effects of MPC expression were related to the mitochondrial transport of pyruvate, then limiting the production of pyruvate should mitigate these effects. Therefore, we measured cell size in larvae raised on either a normal (9% sugar) diet or a diet with no added sugars, which limits the production of pyruvate from glucose and fructose. Limiting dietary sugars significantly reduced the size of control clones but increased the size of MPC-expressing clones. Notably, the size of MPC-expressing fat body clones was comparable to that of control clones when larvae were grown in sugar-limited medium, suggesting that limiting pyruvate synthesis abolishes the effect of MPC expression on fat body cell size (*Figure 1h, i*). *Mpc1*-KD clones were again larger than control cells, and their size was unaffected by the sugar-limited diet. Inhibiting MPC activity by feeding larvae a normal diet supplemented with the MPC inhibitor UK5099 ameliorated the cell size effects of MPC expression (*Figure 1j*; *Figure 1—figure supplement 2g*). These results indicate that pyruvate transport into mitochondria inversely correlates with the size gain in fat body cells. These observations also suggest that the suppression of mitochondrial pyruvate import and metabolism is required for the rapid cell size expansion observed in the fat body during larval development.

The size of a cell is determined by its content of proteins, nucleic acids, and lipids (*Björklund, 2019*; *Lloyd, 2013*; *Schmoller and Skotheim, 2015*). To understand how MPC expression affects the abundance of these macromolecules, we dissected fat bodies from control and fat body-wide MPC-expressing larvae at 120 hr AEL and quantified DNA, RNA, triacylglycerides, and protein. Control and MPC+ fat bodies had equivalent DNA content (*Figure 1k*; *Figure 1—figure supplement 2h*) and similar levels of EdU incorporation (*Figure 1—figure supplement 2o, p*), suggesting that MPC expression does not impact DNA endoreplication. RNA and lipid content were modestly decreased in MPC+ fat bodies compared with control tissues (*Figure 1k*; *Figure 1—figure supplement 2i, j*), although the number of lipid droplets was higher with MPC expression (*Figure 1—figure supplement 2l–n*). In contrast, MPC expression dramatically decreased protein abundance in fat bodies (*Figure 1k*, *Figure 1—figure supplement 2k*), and reduced protein synthesis as assessed by staining for the puromycin analog, O-propargyl-puromycin (OPP) (*Deliu et al., 2017*; *Villalobos-Cantor et al., 2023*; *Figure 1l and m*). These data suggest that MPC-mediated mitochondrial pyruvate import decreases protein synthesis, which likely contributes to the reduced size of MPC-expressing cells. Conversely, in developing fly larvae, repression of the MPC and the subsequent decrease in mitochondrial pyruvate appear to provide a metabolic mechanism to support a rapid expansion in cell size.

## Growth factor signaling pathways are hyperactivated in MPC-expressing cells

The best understood mechanisms that govern cell size involve conserved signaling and transcriptional networks (*Björklund, 2019*; *Grewal, 2009*; *Lang et al., 1998*; *Lloyd, 2013*). For example, the mTORC1 pathway coordinates both extracellular and intracellular growth regulatory signals to dictate the synthesis and degradation of macromolecules, including proteins, lipids, and nucleic acids (*Gonzalez and Rallis, 2017*; *Gonzalez and Rallis, 2017*; *Saxton and Sabatini, 2017*). mTORC1 increases protein synthesis through the phosphorylation of several proteins, including S6 kinase (S6K) and 4EBP1 (*Figure 2a*). Since MPC-expressing clones were smaller in size and had reduced protein synthesis compared with control clones, we assessed mTORC1 activity in the fat body using our mosaic expression system. As a control, we confirmed that *S6k* (the *Drosophila* gene encoding S6K) over-expressing clones were larger in size and had elevated phospho-S6 staining compared with wild-type clones (*Figure 2b and c*). Surprisingly, MPC+ clones also had elevated phospho-S6 staining (*Figure 2b, c*), suggesting that despite their small size, these cells have high mTORC1 activity. Over-expression of *Rheb*, which is an upstream activator of mTORC1, resulted in clones that were larger than wild-type controls and which had increased phospho-4EBP1 staining (*Figure 2d, e*). Again, even though the MPC+ clones were smaller in size, we observed robust p-4EBP1 staining, indicating that mTORC1 is hyperactive in these cells.

Gene set enrichment analysis of RNA-sequencing data from MPC-expressing fat bodies showed enrichment for signatures associated with pro-growth signaling (*Figure 2—figure supplement 1a*). We assessed markers of several of these pathways in MPC+ clones. In addition to the mTORC1 pathway, Myc increases cell growth by regulating the transcription of ribosome subunits and biosynthetic

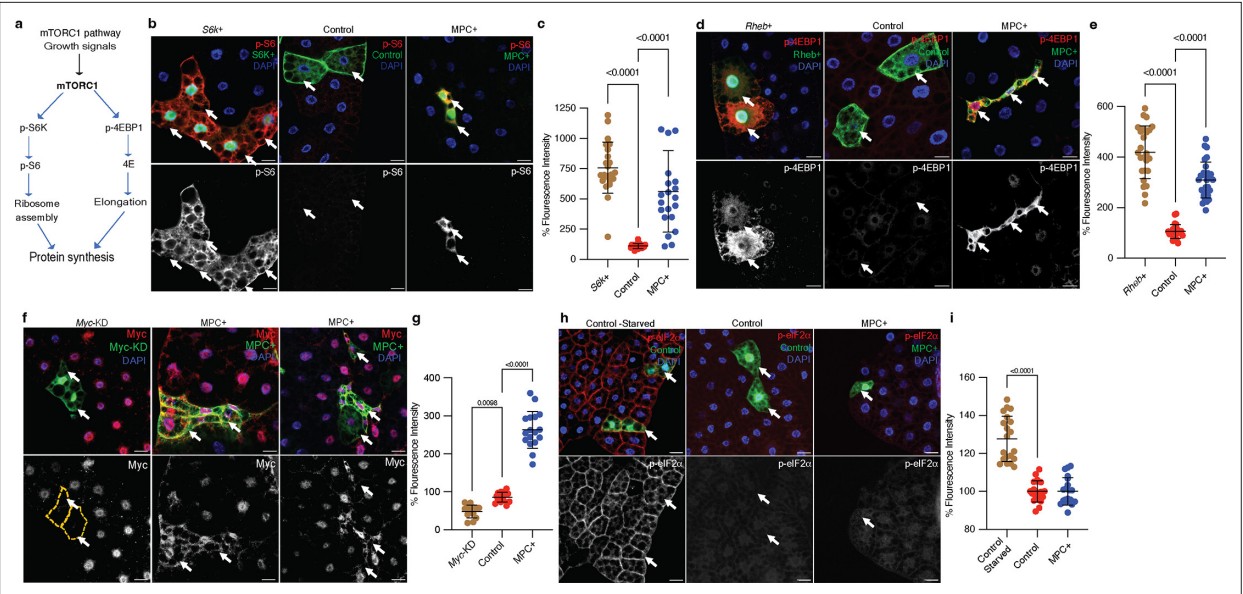

**Figure 2.** mTORC1 and Myc pathways are hyperactivated in MPC expressing fat body cells. (**a**) Schematic of the mTORC1 pathway. The mTORC1 pathway is activated by pro-growth signals such as insulin, leading to the phosphorylation of S6 kinase (S6K) and 4EBP. S6K phosphorylates ribosomal protein S6, while 4EBP phosphorylation inactivates 4EBP and releases elongation factor eIF4E. These events increase ribosomal assembly and elongation rates, thereby enhancing protein synthesis. (**b**) Representative images of fat body clones with *S6k* overexpression, control clones, and MPC+ clones, stained for phosphorylated S6 (p-S6) in red (top panels) and white (bottom panels). Arrows indicate clonal cells with the specified genetic manipulation. The scale bar represents 20 µm. (**c**) Quantification of total p-S6 fluorescence intensity in GFP-positive cells compared to neighboring GFP-negative cells in MPC+ versus control clones. Data are presented as mean ± s.d. with statistical significance as scored with one-way ANOVA test. (**d**) Representative images of fat body clones with *Rheb* over-expression, control clones, and MPC+ clones stained for phosphorylated 4EBP (p-4EBP) in red (top panels) and white (bottom panels). Arrows indicate clonal cells with the specified genetic manipulation. The scale bar represents 20 µm. (**e**) Quantification of total p-4EBP fluorescence intensity in MPC+ versus control clones with statistical significance as scored with One-way ANOVA test. (**f**) Representative images of fat body clones with *Myc* knockdown, control clones, and MPC+ clones stained for Myc protein in red (top panels) and white (bottom panels). Arrows indicate clonal cells with the specified genetic manipulation. The scale bar represents 20 µm. (**g**) Quantification of total Myc fluorescence intensity in MPC+ versus control clones with statistical significance as scored with one-way ANOVA test. (**h**) Representative images of fat body clones from starved wild type, control clones, and MPC+ clones, stained for phosphorylated eIF2α (p-eIF2α) in red (top panels) and white (bottom panels). Arrows indicate clonal cells with the specified genetic manipulation. The scale bar represents 20 µm. (**i**) Quantification of total p-eIF2α fluorescence intensity in MPC+ versus control clones with statistical significance as scored with one-way ANOVA test. Data are presented as mean ± s.d. One-way ANOVA tests were performed to evaluate the statistical significance of the data, with p-values noted in the graph if significance was observed.

The online version of this article includes the following source data and figure supplement(s) for figure 2:

**Source data 1.** Source data related to *Figure 2*.

**Figure supplement 1.** mTORC1 and Myc pathways are hyperactivated in MPC-expressing fat body cells.

**Figure supplement 1—source data 1.** Source data related to *Figure 2—figure supplement 1*.

metabolic genes. We found that MPC+ clones had an elevated abundance of the Myc protein, both in the cytoplasm and nucleus (***Figure 2f, g***), suggesting that the growth-promoting Myc transcriptional program is active in these cells. MPC+ clones also had reduced expression of the transcription factor Foxo, consistent with its downregulation by pro-growth signaling pathways (***Figure 2—figure supplement 1b***). Finally, we stained for phospho-eIF2α to assess the activity of the integrated stress response, which restricts global protein synthesis. Starvation robustly induced the integrated stress response in control clones, but MPC+ clones exhibited no evidence of activation of this pathway under normal growth conditions (***Figure 2h, i***). Collectively, these data indicate that conventional pro-growth pathways are activated in MPC-expressing clones, which is incongruent with the small size of the clones. This suggests that mitochondrial pyruvate metabolism controls cell size via alternative molecular mechanisms.

We next performed genetic epistasis analysis to better understand the relationship between MPC expression and the mTORC1, PI3K, and Myc pathways. Activation of the PI3K or mTORC1 pathways, via overexpression of *PI3K*, increased the size of control but not MPC-expressing fat body cells

(*Figure 2—figure supplement 1c*). Similarly, activation of mTORC1 via knockdown of its inhibitor, Tuberous Sclerosis Complex 1 (*Tsc1*), increased the size of control cells but had no effect on MPC+ cell size (*Figure 2—figure supplement 1d*). Overexpression of *Myc*, on the other hand, increased the size of control and MPC+ fat body clones to a similar degree (*Figure 2—figure supplement 1e*). Knockdown of *Myc* was sufficient to decrease cell size to a similar extent as MPC expression; however, *Myc* knockdown had no additional effect on cell size in MPC+ clones (*Figure 2—figure supplement 1f*). These results suggest that neither mTORC1, PI3K, nor Myc is epistatic to MPC and suggest that MPC likely acts independently of these canonical pathways to regulate the size of fat body cells.

We next performed genetic epistasis analysis to better understand the relationship between MPC expression and the mTORC1, PI3K, and Myc pathways. Activation of the PI3K or mTORC1 pathways, via overexpression of *PI3K*, increased the size of control but not MPC-expressing fat body cells (*Figure 2—figure supplement 1c*). Similarly, activation of mTORC1 via knockdown of its inhibitor, Tuberous Sclerosis Complex 1 (*Tsc1*), increased the size of control cells but had no effect on MPC+ cell size (*Figure 2—figure supplement 1d*). Overexpression of *Myc*, on the other hand, increased the size of control and MPC+ fat body clones to a similar degree (*Figure 2—figure supplement 1e*). Knockdown of *Myc* was sufficient to decrease cell size to a similar extent as MPC expression; however, *Myc* knockdown had no additional effect on cell size in MPC+ clones (*Figure 2—figure supplement 1f*). These results suggest that neither mTORC1, PI3K, nor Myc is epistatic to MPC and suggest that MPC likely acts independently of these canonical pathways to regulate the size of fat body cells.

## Increased mitochondrial pyruvate transport reduces the size of HepG2 cells and spheroids

Since the *Drosophila* fat body has many features reminiscent of the mammalian liver, we tested whether MPC expression might similarly restrict cell size in cells derived from this tissue. We engineered HepG2 cells, which were originally isolated from a hepatocellular carcinoma, to express epitope-tagged MPC1 and MPC2 in a doxycycline-inducible manner. We observed expression of ectopic MPC1 and MPC2 starting at 4 hr after doxycycline treatment, which increased over the duration of the time course, peaking at 24 hr post-induction (*Figure 3a*). We stained doxycycline-treated cells with phalloidin and measured cell size using microscopy images taken at 2-hr intervals. Induction of MPC expression coincided with a significant reduction in cell size, which was first apparent 6 hr post-doxycycline treatment and was sustained for the remainder of the time course (*Figure 3b*). Doxycycline had no effect on the size of control HepG2 cells (EV). Treatment with the MPC inhibitor UK5099 for 24 hr markedly increased the size of control cells and partially reversed the small size phenotype of MPC-expressing cells (*Figure 3c*). We also assessed cell volume by analyzing a 3D reconstruction of confocal images of phalloidin-stained HepG2 cells and found that the cell volume was lower in MPC+ cells (*Figure 3— figure supplement 1a*).

We have observed that the physiological consequences of altered pyruvate metabolism are more apparent in cells grown in a three-dimensional culture environment (*Schell et al., 2014*; *Wei et al., 2022*). Using our doxycycline-inducible cells, we found that MPC expression resulted in spheroids that were significantly smaller than control spheroids as assessed by microscopy (*Figure 3d, e*). Cells from MPC-expressing spheroids were also smaller, as shown by flow cytometry (*Figure 3f, g*). When compared with control (EV) spheroids, expression of MPC did not affect the number of cells per spheroid, the cell cycle phase distribution, or the number of apoptotic or necrotic cells (*Figure 3— figure supplement 1b-d*). These analyses suggest that the smaller size observed in MPC-expressing spheroids is not due to effects on cell proliferation or cell death.

Consistent with our observations in the *Drosophila* fat body, protein content was lower in MPC-expressing HepG2 spheroids compared with EV controls, whereas there was no difference in the abundances of DNA, RNA, or triacylglycerides (*Figure 3h*; *Figure 3—figure supplement 1e-h*). To directly assess the effect of MPC expression on protein synthesis, we induced the MPC in HepG2 cells and treated the cells acutely with a low concentration of puromycin to label nascent proteins. We found that MPC expression decreased the incorporation of the amino acid analog L-homopropargyl-glycine (HPG) (*Shen et al., 2021*; *Tom Dieck et al., 2012*) into nascent proteins, as assessed by fluorescence microscopy (*Figure 3i, j*). Protein synthesis was similarly reduced in MPC-expressing HepG2 spheroids (*Figure 3k, l*). These observations were further supported by the decreased abundance of the short-lived, destabilized GFP (d2GFP) (*Li et al., 1998*; *Pavlova et al., 2020*) in MPC-expressing

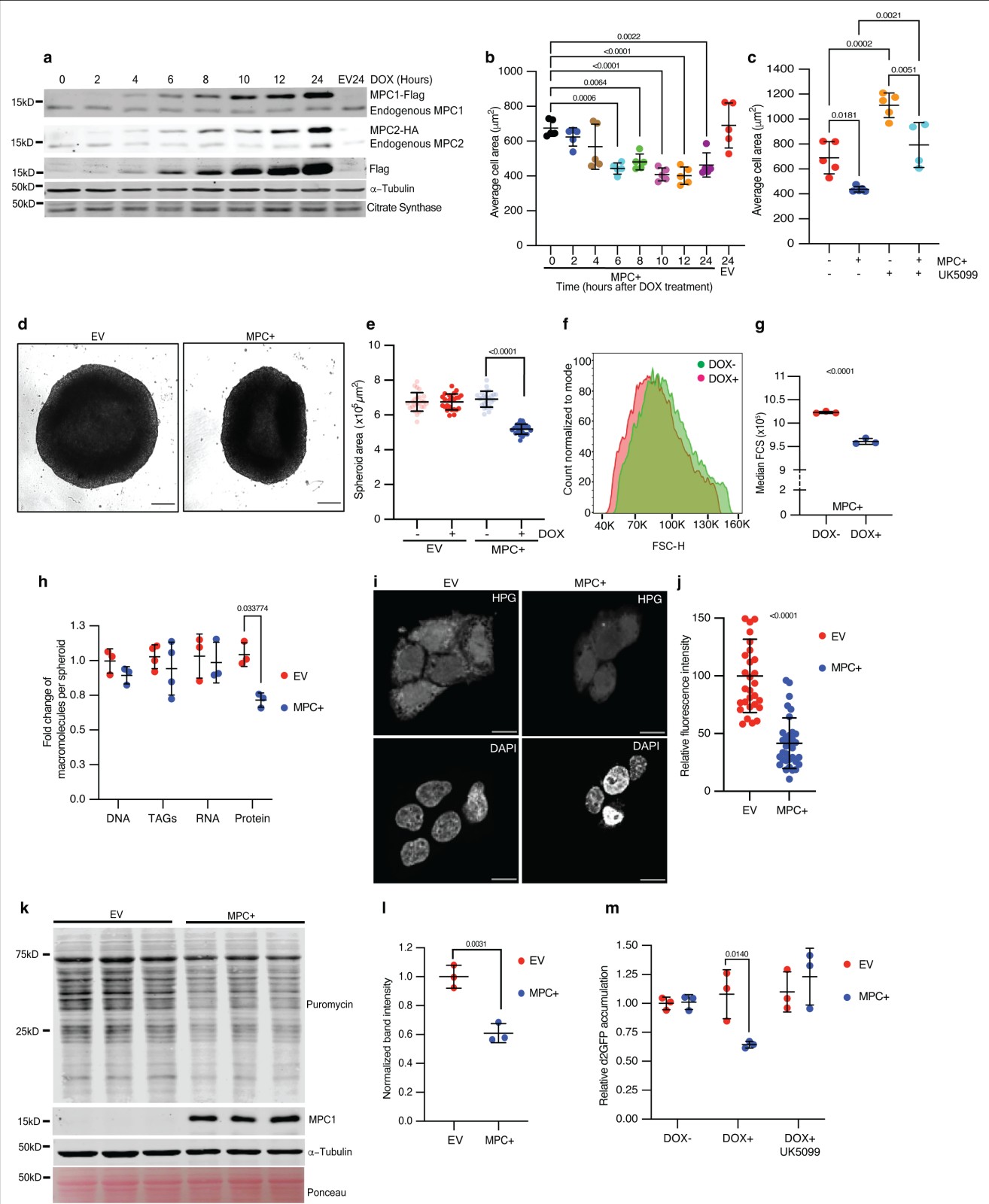

**Figure 3.** Increased mitochondrial pyruvate transport reduces size of HepG2 spheroids. (**a**) Western blots showing inducible expression of MPC1 and MPC2 at 2-hr intervals following treatment with 1 μg/ml doxycycline. Citrate synthase and tubulin were used as loading controls. Both endogenous and epitope tag bands are shown. (**b**) Quantification of the 2D area of HepG2 cells with MPC expression (MPC+) or empty vector (EV) fixed and stained with rhodamine phalloidin at the indicated times after doxycycline treatment. Data are presented as mean ± s.d. from five biological replicates, with each

*Figure 3 continued on next page*

*Figure 3 continued*

replicate representing the average size of 25 randomly selected cells. (**c**) Quantification of the 2D area of MPC+ or EV HepG2 cells treated with 10 μM UK5099. Data are presented as mean ± s.d. from five biological replicates, with each replicate representing the average size of 25 randomly selected cells. (**d**) Representative brightfield images of HepG2 spheroids with empty vector (EV) or MPC expression (MPC+) treated with 1 μg/ml doxycycline for 6 days. The scale bar represents 200 μm. (**e**) Quantification of spheroid area from images of MPC+ or EV HepG2 spheroids, with or without doxycycline treatment. Data are presented as mean ± s.d. from 30 technical replicates. (**f**) Forward scatter (FSC) of cells dissociated from MPC+ (red) or EV spheroids with cell count normalized to mode. (**g**) Median FSC of MPC+ HepG2 spheroids treated with or without 1 μg/ml doxycycline. Data are presented as mean ± s.d. from three biological replicates. (**h**) Fold change in macromolecules—DNA, TAGs, RNA, and protein—fractionated from EV or MPC+ HepG2 spheroids normalized to that in EV HepG2 cells. (**i**) Representative images showing L-homopropargylglycine (HPG)-labeled newly synthesized proteins in EV or MPC+ HepG2 cells. The top panels show HPG staining, and the lower panels show nuclei stained with DAPI. The scale bar represents 20 μm. (**j**) Quantification of HPG fluorescence intensity is presented as mean ± s.d. from 35 cells, for both EV and MPC+ cells. (**k**) Western blot analysis of nascent protein synthesis using a puromycin incorporation assay (20 μg/ml puromycin for 30 min) in either EV or MPC+ HepG2 spheroids (protein lysates from 16 spheroids loaded in each lane). (**l**) Quantification of 70 kD band intensity in puromycin blot normalized with tubulin band intensity and represented as mean ± s.d. from three independent experiments. (**m**) Relative accumulation of destabilized GFP (d2GFP) in EV or MPC+ HepG2 spheroids treated with or without 1 μg/ml doxycycline ± 10 μM UK5099. Data are presented as mean ± s.d. from three biological replicates. Unpaired *t*-tests, one-way, or two-way ANOVA tests were performed to evaluate the statistical significance of the data, with p-values noted in the graph if significance was observed.

The online version of this article includes the following source data and figure supplement(s) for figure 3:

**Source data 1.** Source data related to *Figure 3*.

**Source data 2.** Original files for western blot analysis displayed in *Figure 3a*.

**Source data 3.** PDF files containing original western blots for *Figure 3a*, indicating the relevant bands and treatments.

**Source data 4.** Original files for western blot analysis displayed in *Figure 3k*.

**Source data 5.** PDF files containing original western blots for *Figure 3k*, indicating the relevant bands and treatments.

**Figure supplement 1.** MPC overexpression reduces size of HepG2 cells in 2D and spheroid culture.

**Figure supplement 1—source data 1.** Source data related to *Figure 3—figure supplement 1*.

HepG2 spheroids compared with controls (*Figure 3m*). This reduction in the level of d2GFP was prevented by treatment with the MPC inhibitor UK5099 (*Figure 3m*). Together, these data suggest that increased transport of pyruvate into mitochondria, mediated by MPC expression, reduces protein synthesis and cell size in both fly and mammalian models.

## Excess mitochondrial pyruvate promotes gluconeogenesis

We have previously shown that loss of the MPC reduces the contribution of glucose and pyruvate to the TCA cycle (*Bensard et al., 2020*; *Cluntun et al., 2021*). To investigate how MPC expression impacts carbohydrate metabolism in HepG2 cells, we performed metabolic tracing using $^{13}$C-glucose (*Figure 4a*). We observed that MPC expression reduced the labeling fraction of the glycolytic intermediates 3-phosphoglycerate (*Figure 4—figure supplement 1a*) and pyruvate (*Figure 4—figure supplement 1b*), as well as alanine (*Figure 4—figure supplement 1c*), which is derived from pyruvate, all of which suggests a reduction in glycolysis in these cells. Once imported into mitochondria, glycolytic-derived pyruvate has two major fates: conversion to acetyl-CoA by PDH or to oxaloacetate by pyruvate carboxylase (PC). These fates can be differentiated by assessing the abundances of M+2 (derived from PDH) and M+3 (derived from PC) TCA cycle intermediates (*Figure 4a*). We found that both M+2 and M+3 isotopomers of TCA cycle intermediates were modestly increased following MPC expression (*Figure 4b and c*; *Figure 4—figure supplement 1d–i*). The fact that we observe increases in TCA cycle intermediates despite decreased glycolytic labeling in pyruvate suggests that the apparent labeling through PDH and PC might be underestimating the effect on metabolic flux through these enzymes. Thus, it appears that MPC expression increases the activity of both enzymes that utilize pyruvate. Typically, when TCA cycle flux is increased, one observes an increase in activity of the electron transport chain (ETC) to oxidize the resulting NADH. However, MPC expression had no impact on ETC activity as assessed by measuring oxygen consumption (*Figure 4d*). The implications of elevated NADH production without a concomitant increase in NADH oxidation will be discussed in *Figure 5*.

To probe the impact of increased flux through the PC and PDH reactions, we conducted genetic epistatic analysis in fat body cells. We found that *Drosophila* fat body clones in which we over-expressed *Pcb* (the *Drosophila* gene encoding pyruvate carboxylase) were significantly smaller than controls and

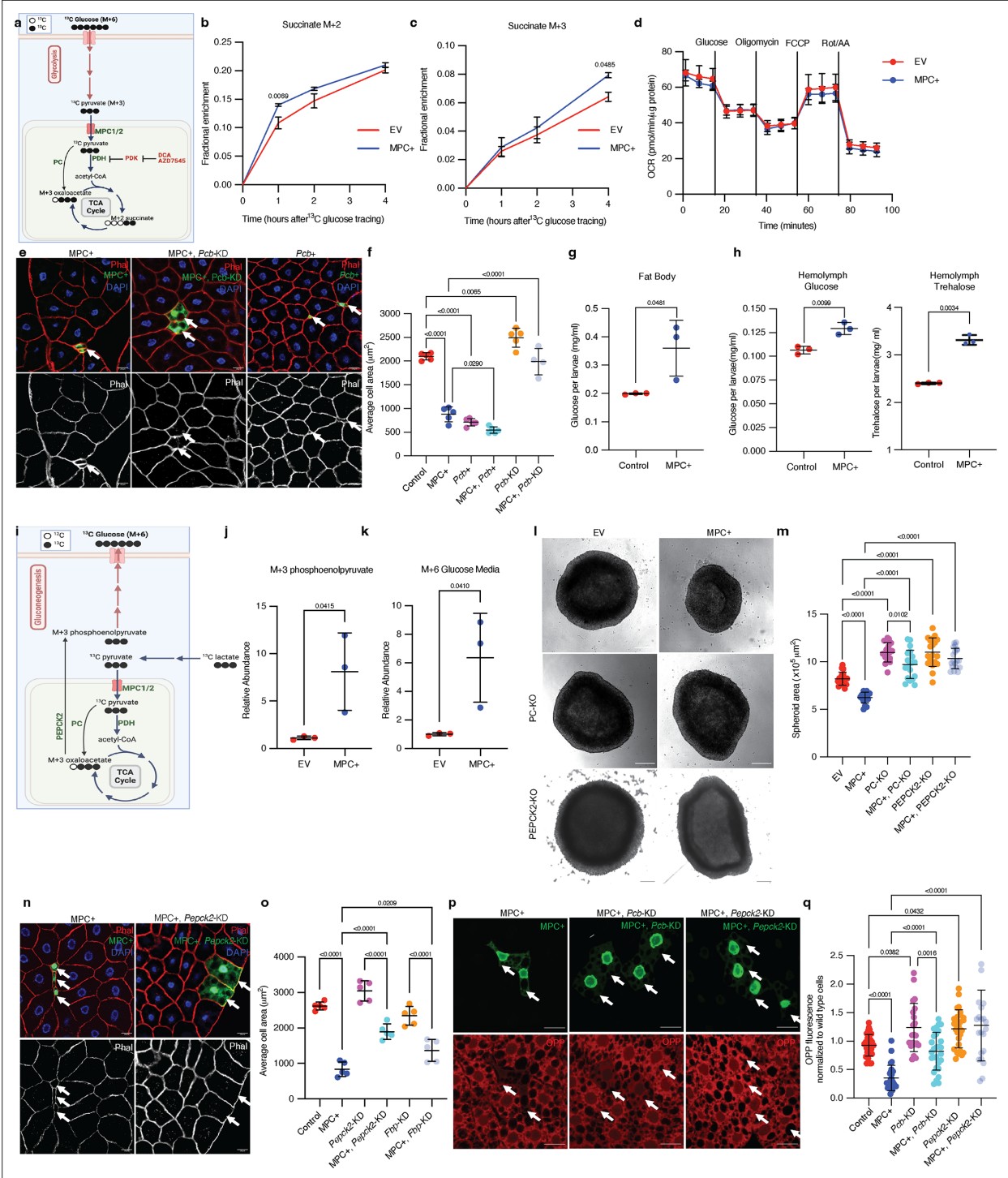

**Figure 4.** Increased mitochondrial pyruvate metabolism promotes gluconeogenesis via pyruvate carboxylase to suppress protein synthesis. (**a**) Schematic illustration of the ¹³C-glucose tracing strategy used to measure the activity of pyruvate dehydrogenase (PDH) and pyruvate carboxylase (PC). TCA metabolites labeled with two heavy carbons (¹³C or M+2 TCA pool) result from PDH activity, whereas M+3 TCA metabolites result from PC activity. PDH is inhibited by PDK-mediated phosphorylation. DCA and AZD7545 are inhibitors of PDK. (**b**) Fractional enrichment of M+2 succinate in empty vector (EV; red) and MPC-expressing (MPC+; blue) HepG2 cells at the indicated times after ¹³C-glucose tracing. MPC expression was induced for 24 hr by treatment with 1 µg/ml doxycycline and media was changed to ¹²C-glucose. The change in M+2 succinate is significant (by two-way ANOVA test) at 1 hr after ¹³C glucose incubation. (**c**) Fractional enrichment of M+3 succinate in EV (red) and MPC+ (blue) HepG2 at the indicated times after ¹³C-glucose tracing. MPC expression was induced for 24 hr by treatment with 1 µg/ml doxycycline. The change in M+3 succinate is significant (by two-way ANOVA test) at 4 hr after ¹³C-glucose incubation. (**d**) Rate of oxygen consumption (OCR) in EV and MPC+ HepG2 cells. (**e**) Representative images of phalloidin-

*Figure 4 continued on next page*

*Figure 4 continued*

and DAPI-stained fat body cells. Arrows indicate GFP-positive clones with MPC expression (MPC+), *Pcb* knockdown with MPC expression (MPC+, *Pcb*-KD), or *Pcb* overexpression (*Pcb*+). The scale bar represents 20 µm. (**f**) Quantification of the area of GFP-positive clones with control, MPC+, *Pcb* over-expression (*Pcb*+), *Pcb* and MPC co-expression (MPC+, *Pcb*+), *Pcb* knockdown (*Pcb*-KD), and *Pcb* knockdown with MPC expression (MPC+, *Pcb*-KD) shown as mean ± s.d. of five biological replicates, with each group representing the analysis of 20 the indicated clonal cells. Concentration of glucose in the fat body (**g**) and concentration of glucose and trehalose in hemolymph (**h**) of larva with fat body-specific expression MPC or control. Data is presented as mean ± s.d. of three biological replicates analyzed by unpaired *t*-tests. (**i**) Schematic illustration of the strategy to analyze gluconeogenesis from $^{13}$C-lactate. Cells convert $^{13}$C-lactate into $^{13}$C-pyruvate, which is transported into mitochondria by the MPC. PC converts $^{13}$C-pyruvate (M+3) into oxaloacetate (M+3). PEPCK2 converts oxaloacetate (M+3) into phosphoenolpyruvate (M+3), which is converted into M+6 glucose and excreted from cells. (**j**) Relative abundances of M+3 phosphoenolpyruvate (PEP) in EV and MPC+ HepG2 cells and (**k**) M+6 glucose in their respective media following treatment with 20 mM $^{13}$C-lactate for 4 hr. Data is presented as mean ± s.d. of three biological replicates, each with an average of three technical replicates. (**l**) Representative brightfield images of EV, MPC+, and PC knockout (KO) or PEPCK2 KO with or without MPC expression HepG2 spheroids. The scale bar represents 200 µm. (**m**) Quantification of spheroid area is presented as mean ± s.d. of 30 technical replicates. (**n**) Representative images of phalloidin- and DAPI-stained fat body cells. Arrows indicate GFP-positive clones with MPC expression (MPC+), and *Pepck2* knockdown with MPC expression (MPC+, *Pepck2*-KD). The scale bar represents 20 µm. (**o**) Quantification of the area of GFP-positive clones with MPC+, *Pepck2* knockdown (*Pepck2*-KD), *Pepck2* knockdown with MPC+ (MPC+, *Pepck2*-KD), *Fbp* knockdown (*Fbp*-KD), *Fbp* knockdown with MPC+ (MPC+, *Fbp*-KD). Data is presented as mean ± s.d. of five biological replicates, with each group analyzing 20 clonal cells of the mentioned genetic manipulations. (**p**) Representative images of fat body clones stained with OPP (red). Arrows indicate GFP-positive clones with MPC expression (MPC+), *Pcb* knockdown with MPC expression (MPC+, *Pcb*-KD), or *Pepck2* knockdown with MPC expression (MPC+, *Pepck2*-KD). The scale bar represents 20 µm. (**q**) Quantification of OPP intensity in the indicated clones compared with adjacent wild-type cells. Data is presented as mean ± s.d. Unpaired *t*-tests, one-way ANOVA tests, or two-way ANOVA tests were performed to evaluate the statistical significance of the data, with p-values mentioned in the graph if significance is noted. Panel a was created with BioRender.com. Panel i was created with BioRender.com.

The online version of this article includes the following source data and figure supplement(s) for figure 4:

**Source data 1.** Source data related to *Figure 4*.

**Figure supplement 1.** Surge of mitochondrial pyruvate reduced glycolysis and promotes gluconeogenesis via pyruvate carboxylase.

**Figure supplement 1—source data 1.** Source data related to *Figure 4—figure supplement 1*.

**Figure supplement 2.** Surge of mitochondrial pyruvate promotes gluconeogenesis via pyruvate carboxylase to suppress protein synthesis.

**Figure supplement 2—source data 1.** Source data related to *Figure 4—figure supplement 2*.

**Figure supplement 2—source data 2.** PDF files containing original western blots for *Figure 4—figure supplement 2e*, indicating the relevant bands and treatments.

**Figure supplement 2—source data 3.** Original files for western blot analysis displayed in *Figure 4—figure supplement 2e*.

**Figure supplement 2—source data 4.** PDF files containing original western blots for *Figure 4—figure supplement 2m*, indicating the relevant bands and treatments.

**Figure supplement 2—source data 5.** Original files for western blot analysis displayed in *Figure 4—figure supplement 2m*.

were equivalent in size to MPC+ clones (*Figure 4e, f*; *Figure 4—figure supplement 2a–c*). Clones expressing both the MPC and *Pcb* were even smaller (*Figure 4e, f*; *Figure 4—figure supplement 2a–c*). Conversely, knockdown of *Pcb* (*Pcb*-KD) in MPC-expressing fat body clones completely rescued the cell size phenotype (*Figure 4e, f*; *Figure 4—figure supplement 2d*). Knockdown of *Pcb* in MPC-expressing HepG2 cells also eliminated the small cell size phenotype (*Figure 4—figure supplement 2e*). These data suggested that the small size of cells expressing the MPC is likely due to increased flux through the PC reaction. Consistent with this, knockdown of the PDH-E1 (*Drosophila* gene *Pdha*) or PDH-E2 (*Drosophila* gene *Dlat*) subunits of PDH, which should divert mitochondrial pyruvate, so it is preferentially used by PC, also resulted in smaller fat body cells (*Figure 4—figure supplement 2f–k*). In contrast, activating PDH by knocking down the inhibitory PDH kinases (mammalian PDKs, *Drosophila* gene *Pdk*) (*Stacpoole, 2017*; *Wang et al., 2021*), which should promote the flux of pyruvate through PDH and away from PC, rescued cell size in MPC-expressing fat body clones (*Figure 4—figure supplement 2j, k*). Similarly, in MPC-expressing HepG2 cells, activation of PDH via treatment with the PDK inhibitors DCA and AZD7545 was sufficient to restore cell size (*Figure 4—figure supplement 2l*), (*Stacpoole, 2017*; *Wang et al., 2021*). These analyses suggest that the reduction in cell size observed with MPC expression is due to PC-mediated metabolism of pyruvate.

The product of PC, oxaloacetate, has three major metabolic fates: (1) feeding the TCA cycle via citrate synthase as discussed above, (2) conversion to aspartate by glutamic-oxaloacetic transaminase 2 (GOT2), and (3) conversion to phosphoenolpyruvate by PEPCK2, leading to gluconeogenesis, synthesis of glucose (*Jitrapakdee et al., 2008*; *Kiesel et al., 2021*). Sustained expression of the MPC

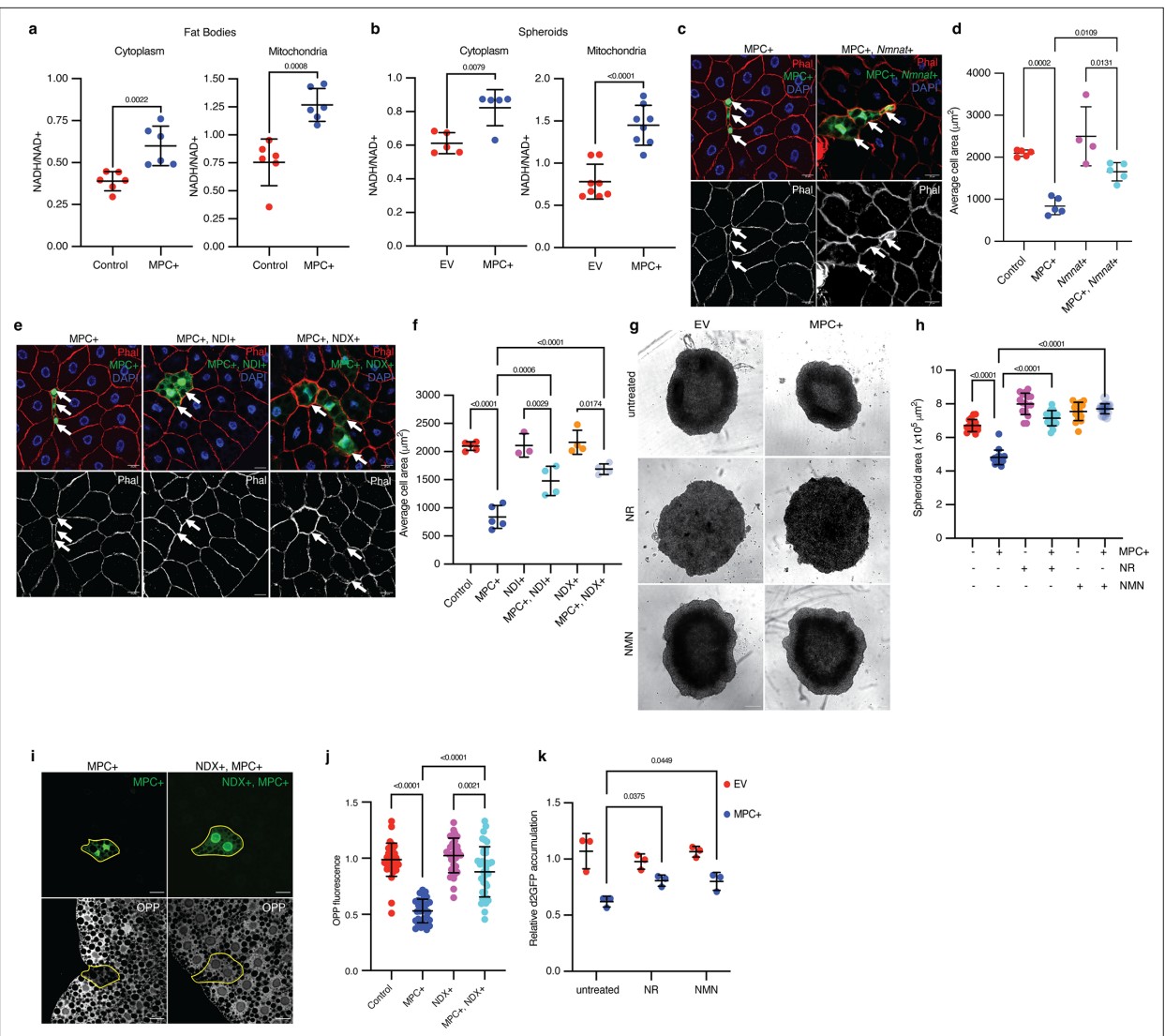

**Figure 5.** Redox imbalance impedes protein synthesis and cell growth from elevated pyruvate metabolism in mitochondria. (**a**) NADH/NAD+ ratios from the cytoplasmic and mitochondrial fractions of control and MPC-expressing (MPC+) fat bodies. Data is presented as mean ± s.d. of six biological replicates. (**b**) NADH/NAD+ ratios from the cytoplasmic and mitochondrial fractions of empty vector (EV) and MPC-expressing (MPC+) HepG2 spheroids. Data is presented as mean ± s.d. of six biological replicates. (**c**) Representative images of phalloidin- and DAPI-stained fat bodies. Arrows indicate GFP-positive cells with either MPC+ or MPC and *Nmnat* co-expression (MPC+, *Nmnat*+). The scale bar represents 20 μm. (**d**) Quantification of the areas of GFP-positive cells with control, MPC+, *Nmnat* overexpression (*Nmnat*+), MPC and *Nmnat* co-expression (MPC+, *Nmnat*+). Data is presented as mean ± s.d. of five biological replicates, with 20 clonal cells analyzed for each of the indicated genetic manipulations. (**e**) Representative images of Phalloidin and DAPI-stained fat body tissues. Arrows indicate GFP-positive cells showing MPC expression (MPC+); NDI and MPC expression (MPC+, NDI+); and NDX and MPC expression (MPC+, NDX+). The scale bar represents 20 μm. (**f**) Quantification of the area of GFP-positive cells with control, MPC+, NDI expression (NDI+), NDI and MPC expression (MPC+, NDI+), NDX expression (NDX+), NDX and MPC co-expression (MPC+, NDX+). Data is presented as mean ± s.d. of five biological replicates, with 20 clonal cells analyzed for each of the indicated genetic manipulations. (**g**) Representative bright field images of EV or MPC+HepG2 spheroids cultured with NAD+ supplements (100 nM nicotinamide riboside or 1 μM NMN) as indicated. The scale bar represents 200 μm. (**h**) Quantification of spheroid area is presented as mean ± s.d. of 30 technical replicates. (**i**) Representative images MPC+ or MPC+, NDI+ GFP-positive clones and of fat body cells stained with OPP (bottom). Clonal cells are mapped with dotted lines. The scale bar represents 20 μm. (**j**) Fold change in OPP intensity of 35 GFP-positive cells compared with adjacent wild-type cells. Data is presented as mean ± s.d. (**k**) Relative accumulation of destabilized GFP (d2GFP) in spheroids of EV or MPC+ HepG2 cells treated with NAD+ supplements (100 nM nicotinamide riboside or 1 μM NMN) as indicated. Unpaired t-tests, one-way ANOVA tests, or two-way ANOVA tests were performed to assess the statistical significance of the data, with p-values indicated in the graph where significance was observed.

The online version of this article includes the following source data and figure supplement(s) for figure 5:

**Source data 1.** Source data related to *Figure 5*.

*Figure 5 continued on next page*

*Figure 5 continued*

**Figure supplement 1.** Redox imbalance impedes protein synthesis and cell growth from elevated pyruvate metabolism in mitochondria.

**Figure supplement 1—source data 1.** Source data related to *Figure 5—figure supplement 1*.

**Figure supplement 1—source data 2.** PDF files containing original western blots for *Figure 5—figure supplement 1k*, indicating the relevant bands and treatments.

**Figure supplement 1—source data 3.** Original files for western blot analysis displayed in *Figure 5—figure supplement 1k*.

**Figure supplement 1—source data 4.** PDF files containing original western blots for *Figure 5—figure supplement 1b*, indicating the relevant bands and treatments.

**Figure supplement 1—source data 5.** Original files for western blot analysis displayed in *Figure 5—figure supplement 1b*.

in fat bodies increased the concentration of glucose in this tissue (*Figure 4g*) as well as glucose and trehalose in larval circulation (known as hemolymph) (*Figure 4h*), suggesting that MPC expression increases glucose production in fat body cells. Next, we tested whether PEPCK2-mediated gluconeogenesis was elevated in these cells compared with controls. We used $^{13}$C-lactate to trace the $^{13}$C-labeling of phosphoenolpyruvate and resultant glucose synthesis via gluconeogenesis (*Figure 4i*). We found that the relative abundance of M+3 phosphoenolpyruvate was higher in MPC-expressing cells compared with controls (*Figure 4j*), suggesting increased activity of PEPCK2. The gluconeogenic pathway employs a series of biochemical reactions to convert phosphoenolpyruvate into glucose, which is then excreted from the cell. This production of M+6 glucose from lactate was also higher in MPC-expressing HepG2 cells (*Figure 4k*). To test whether gluconeogenesis contributes to the small size phenotype of MPC-expressing cells, we knocked down enzymes in the pathway (*Drosophila* phosphoenol carboxykinase, *Pepck2,* and fructose bisphosphatase, *Fbp*) and assessed cell size in *Drosophila* and HepG2 models. Knockout of PEPCK2 rescued the size of MPC-expressing HepG2 cells grown in spheroids (*Figure 4l*) and cells (*Figure 4—figure supplement 2m*). Knockdown of either *Pepck2* or *Fbp* partially rescued the size defect in MPC-expressing *Drosophila* fat body clones (*Figure 4n, o*; *Figure 4—figure supplement 2n–r*), and *Pepck2* knockdown also increased protein synthesis in these clones (*Figure 4p, q*). Collectively, these data suggest that the cell size and protein synthesis phenotypes observed in MPC-expressing cells require PC-mediated gluconeogenesis, and this relationship is found in both the *Drosophila* fat body and in HepG2 cells.

## A redox imbalance impairs protein synthesis and cell growth in MPC-expressing cells

We were intrigued by the observation that MPC expression exerted its impact on cell size through PC, rather than via increased flux through PDH. We did not observe any differences in the abundances of PDH1a, phosphorylated (Ser-293, inactive) PDH1a, PC, PEPCK2, or G6PC proteins in MPC-expressing HepG2 cells (*Figure 5—figure supplement 1a, b*). We therefore hypothesized that the shift in pyruvate metabolism in response to MPC expression might be driven by changes in the abundances of enzyme cofactors, metabolite regulators, or cellular redox balance. PC and PDH show reciprocal regulation by such factors: PC utilizes ATP as a substrate and is allosterically activated by acetyl-CoA, whereas PDH is inhibited by ATP, acetyl CoA, and NADH (*Chai et al., 2022*; *Sugden and Holness, 2011*; *Figure 5—figure supplement 1c*). In addition, several reactions in gluconeogenesis require ATP or NADH (*Hausler et al., 2006*; *Siess et al., 1977*; *Figure 5—figure supplement 1c*). Upon induction of MPC expression in HepG2 cells, we observed increased abundance of acetyl-CoA (*Figure 5—figure supplement 1d*), a higher ATP to ADP ratio (*Figure 5—figure supplement 1e*), a greater abundance of NADH without an increase in NAD$^+$, and thus an increase in the cellular NADH/NAD$^+$ ratio (*Figure 5—figure supplement 1f–h*). To determine how MPC expression impacts the subcellular distribution of redox factors, we separated cytoplasmic and mitochondrial fractions from control or MPC-expressing *Drosophila* fat bodies and measured the NADH/NAD$^+$ ratio. We found that MPC expression increased the NADH/NAD$^+$ ratio in both the cytoplasm and mitochondria (*Figure 5a*). We observed a similar increase in the NADH/NAD$^+$ ratio in both fractions in MPC-expressing HepG2 spheroids (*Figure 5b*). It is important to note that these measurements can be confounded by the very rapid redox reactions that can happen post-harvest. Nonetheless, these combined results suggest that the increased abundances of acetyl-CoA, ATP, and NADH in MPC-expressing cells could promote the rewiring of mitochondrial pyruvate metabolism through PC and support gluconeogenesis.

To test the hypothesis that ATP or NADH concentrations might affect cell size in MPC-expressing *Drosophila* fat body clones or HepG2 cells, we utilized pharmacological or genetic modulation of these molecules. Treatment with gramicidin, which decreases the ATP to ADP ratio (*Xue et al., 2022*), did not alter the size of MPC-expressing HepG2 cells (*Figure 5—figure supplement 1i*). We used several orthogonal approaches to reduce the cellular NADH/NAD⁺ ratio in the MPC-expressing systems and measured their effect on cell size. Co-expression of the *Drosophila* nicotinamide mononucleotide adenylyl transferase (*Nmnat*), which increases NAD⁺ biosynthesis, almost normalized the small size phenotype of MPC-expressing clones in the *Drosophila* fat body (*Figure 5c, d*). We observed a similar rescue of cell size following expression of the *Ciona intestinalis* alternate Complex I enzyme NADH dehydrogenase (NDX) (*Gospodaryov et al., 2020*) or the yeast NADH dehydrogenase (NDI) (*Sanz et al., 2010*), both of which oxidize NADH to NAD⁺ without concomitant proton translocation and energy capture (*Figure 5e, f*). Expression of NDI in HepG2 cells also mitigated the effect of MPC expression on spheroid size (*Figure 5—figure supplement 1j, k*), as did treatment with duroquinone (*Merker et al., 2006*; *Figure 5—figure supplement 1l*), which oxidizes NADH to NAD⁺. To extend these investigations to three-dimensional culture, we supplemented the growth medium of MPC-expressing HepG2 spheroids with the NAD⁺ precursors nicotinamide riboside (NR) or nicotinamide mononucleotide (NMN), both of which recovered spheroid size (*Figure 5g, h*).

Since MPC expression reduced protein synthesis in both *Drosophila* fat bodies and HepG2 cells, we tested how cellular redox status might contribute to this phenotype in both systems. Expression of NDX, which lowers the NADH/NAD⁺ ratio, increased translation in MPC-expressing *Drosophila* fat body clones (*Figure 5i, j*). In HepG2 spheroids, boosting NAD⁺ biosynthesis by supplementing growth media with NR or NMN partially rescued the abundance of destabilized GFP (*Figure 5k*). These results suggest that the elevated NADH/NAD⁺ ratio in MPC-expressing cells limits protein synthesis and that normalizing that ratio increases protein synthesis and cell size.

## Reduced amino acid abundance impairs the size of MPC-expressing cells

Given the reduced protein synthesis observed upon MPC expression, we assessed amino acid concentrations in HepG2 using steady-state metabolomics. We found that MPC expression reduced the abundance of most amino acids (*Figure 6a*). To test whether the low abundances of amino acids contribute to the smaller size of MPC-expressing cells, we supplemented the growth media of HepG2 spheroids with excess non-essential amino acids (NEAAs)—either two or three times the recommended dilution of a commercially available amino acid cocktail that includes glycine, L-alanine, L-asparagine, L-aspartate, L-glutamate, L-proline, and L-serine. MPC-expressing spheroids grown with excess NEAAs were comparable in size to controls (*Figure 6b, c*). In parallel, we provided *Drosophila* larvae with food containing excess (5x) amino acids from 72 to 120 hr AEL, which partially rescued cell size in MPC-expressing fat body clones (*Figure 6d, e*). To genetically augment intracellular amino acids, we expressed the amino acid importer *slimfast* (*Colombani et al., 2003*) in control or MPC-expressing fat body clones and found that it prevented the decrease in cell size (*Figure 6f, g)*.

To determine which amino acid(s) contribute to the cell size effects of MPC expression, we cultured control and MPC-expressing HepG2 cells in media supplemented with an excess of each amino acid from the NEAA cocktail. Treatment with excess glycine, L-alanine, or L-serine had no effect on cell size (*Figure 6h*). However, the size of MPC-expressing cells was normalized by supplementation with L-aspartate, L-glutamate, or L-proline (*Figure 6h*), all of which are derived from TCA cycle intermediates. Supplementation with either L-aspartate or L-glutamate also rescued the small size phenotype of MPC-expressing HepG2 spheroids (*Figure 6—figure supplement 1a, b*). Increasing L-aspartate uptake by over-expressing the aspartate transporter SLC1A3 also recovered size of HepG2 spheroids (*Figure 6—figure supplement 1c, d*). In addition, treatment with excess (3x) NEAA partially restored the abundance of d2GFP in MPC-expressing HepG2 spheroids (*Figure 6—figure supplement 1e*), suggesting rescued protein synthesis.

Like glutamate and proline, aspartate is derived from a TCA cycle intermediate, specifically via transamination of oxaloacetate by glutamic-oxaloacetate transaminase 2 (GOT2). Since GOT2 and PEPCK2 both utilize oxaloacetate as a substrate, we hypothesized that knocking down GOT2 might phenocopy MPC expression by driving PEPCK2-mediated conversion of oxaloacetate into phosphoenolpyruvate and thereby suppressing aspartate biosynthesis. Knockdown of *Got2* (the *Drosophila* gene

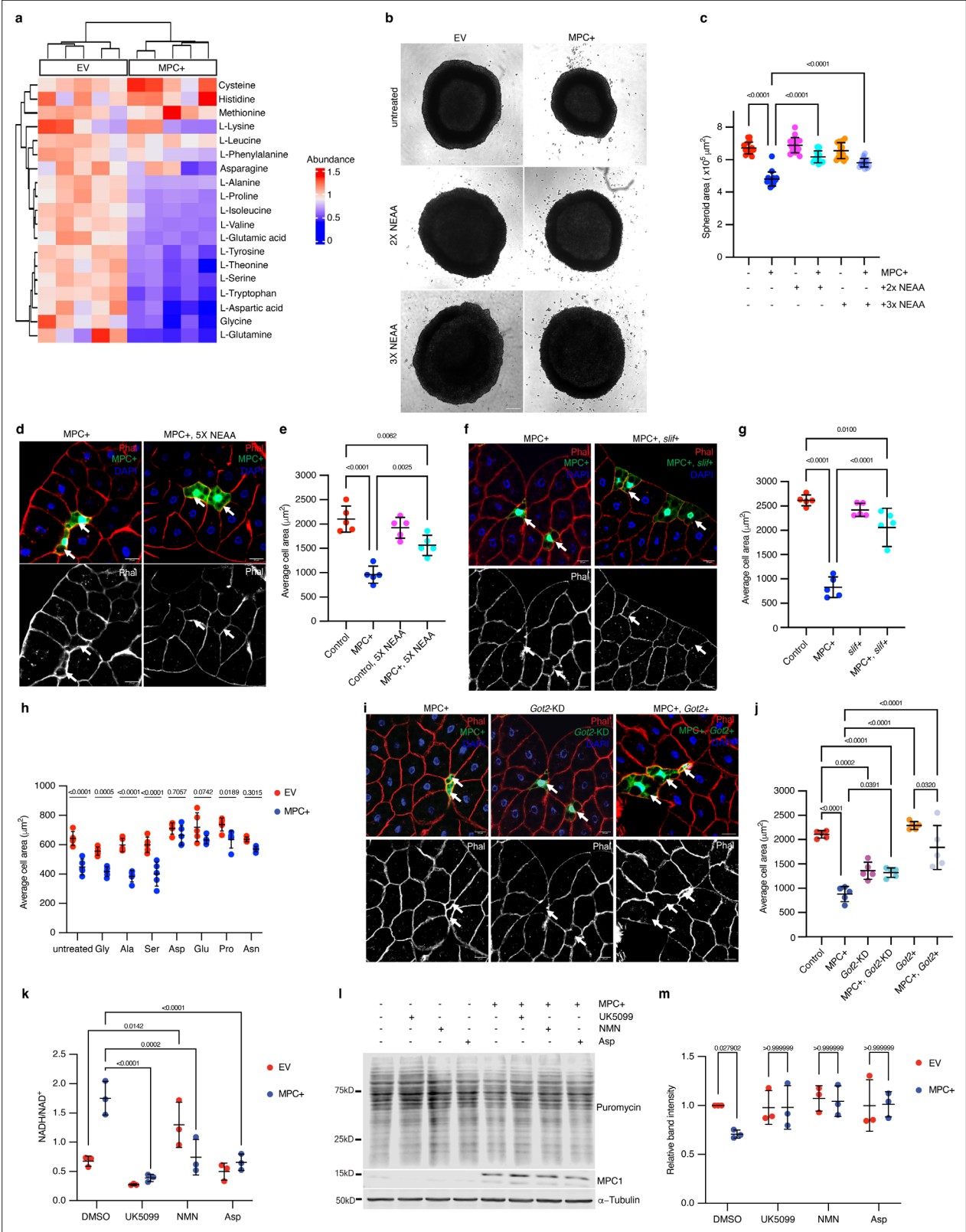

**Figure 6.** Reduced amino acid abundance impairs size of MPC-expressing cells. (**a**) A heatmap of the abundances of amino acids in empty vector (EV) and MPC expressing (MPC+) HepG2 cells cultured under standard conditions. Color codes indicate relative abundances for each amino acid: blue (low), green (similar), and yellow (high). (**b**) Representative bright field images of EV or MPC+ HepG2 spheroids cultured with 2x or 3x the recommended concentration of non-essential amino acids cocktail (NEAA). The scale bar represents 200 µm. (**c**) Quantification of spheroid areas from EV and MPC+

*Figure 6 continued on next page*

*Figure 6 continued*

HepG2 spheroids cultured with 2x or 3x NEAA. Data is presented as mean ± s.d. of 30 technical replicates. (**d**) Representative images of phalloidin- and DAPI-stained fat body cells from animals fed a standard diet or a diet supplemented with 5x NEAA. Arrows indicate GFP-positive clones with MPC expression (MPC+). The scale bar represents 20 μm. (**e**) Quantification of the area of MPC+ fat body clonal cells. Data is presented as mean ± s.d. of five biological replicates, with each data point representing the average size of 20 clones collected from five male larvae. (**f**) Representative images of phalloidin- and DAPI-stained fat body cells. Arrows indicate GFP-positive MPC+ clones and MPC+ and *slimfast* over-expression (MPC+, *Slif*+). The scale bar represents 20 μm. (**g**) Quantification of the area of GFP-positive clones with control, MPC expression, *Slimfast* overexpression (*Slif*+) and MPC+ clones with *Slimfast* overexpression (MPC+, *Slif*+). Data presented as mean ± s.d. of five biological replicates, with each 20 clonal cells analyzed for each of the indicated genetic manipulations. (**h**) Quantification of the cell areas of EV or MPC+ HepG2 cells cultured under standard conditions or with excess of the indicated amino acid—10 mM glycine, 5 mM alanine, 5 mM serine, 5 mM asparagine, 5 mM aspartic acid, 5 mM glutamic acid, or 5 mM proline. Data is presented as mean ± s.d. of five biological replicates. (**i**) Representative images of phalloidin- and DAPI-stained fat body cells. Arrows indicated GFP-positive cells with MPC expression (MPC+), *Got2* knockdown (*Got2* KD), and *Got2* overexpression with MPC expression (MPC+, *Got2*+). (**j**) Quantification of the area of GFP-positive clones with control, MPC expression (MPC+), *Got2* knockdown (*Got2*-KD), *Got2* knockdown with MPC expression (MPC+, *Got2*-KD), *Got2* overexpression (*Got2*+), and *Got2* overexpression with MPC expression (MPC+, *Got2*+). Data is presented as mean ± s.d. of five biological replicates, with each 20 clonal cells analyzed for each of the specified genetic manipulations. (**k**) NADH/NAD$^+$ ratio in cells treated with 10 μM UK5099, 1 μM NMN, or 5 mM aspartate. Data is presented as mean ± s.d. of three biological replicates. (**l**) Western blot analysis of puromycin-labeled (20 μg/ml puromycin for 30 min) nascent protein in EV or MPC+ HepG2 cells cultured with 10 μM UK5099, 1 μM NMN, or 5 mM aspartate. (**m**) Quantification of intensities of puromycin labeling in EV and MPC+ cell lysates. Unpaired *t*-tests and one-way ANOVA tests were performed to evaluate the statistical significance of the data, and p-values are noted in the graph if significance is observed.

The online version of this article includes the following source data and figure supplement(s) for figure 6:

**Source data 1.** Source data related to *Figure 6*.

**Source data 2.** PDF files containing original western blots for *Figure 6i*, indicating the relevant bands and treatments.

**Source data 3.** Original files for western blot analysis displayed in *Figure 6i*.

**Figure supplement 1.** Reduced aspartate/glutamate abundance impairs size of MPC-expressing cells.

**Figure supplement 1—source data 1.** Source data related to *Figure 6—figure supplement 1*.

**Figure supplement 1—source data 2.** PDF files containing original western blots for *Figure 6—figure supplement 1k*, indicating the relevant bands and treatments.

**Figure supplement 1—source data 3.** Original files for western blot analysis displayed in *Figure 6—figure supplement 1k*.

**Figure supplement 1—source data 4.** PDF files containing original western blots for *Figure 6—figure supplement 1i*, indicating the relevant bands and treatments.

**Figure supplement 1—source data 5.** Original files for western blot analysis displayed in *Figure 6—figure supplement 1i*.

encoding GOT2) reduced cell size in *Drosophila* fat body clones and phenocopied MPC-expressing cells (*Figure 6i, j*; *Figure 6—figure supplement 1f–h*). Similarly, both GOT2 knock-out and MPC-expressing HepG2 cells were smaller than EV (*Figure 6—figure supplement 1i*). Expression of the MPC in these GOT2 knockdown systems had no significant impact on cell size (*Figure 6i, j*; *Figure 6—figure supplement 1i*), suggesting that the effects of MPC expression and GOT2 knockdown act similarly to limit amino acid synthesis and cell size. We next performed the reciprocal experiment by over-expressing *Got2* to favor the production of aspartate from oxaloacetate. *Got2* expression normalized cell size in both MPC-expressing fat body clones and in HepG2 spheroids (*Figure 6i, j*; *Figure 6—figure supplement 1j, k*). The efflux of aspartate from mitochondria into the cytoplasm is a critical component of the malate-aspartate shuttle, which is a major redox shuttle in human cells. To test whether increasing the abundance of aspartate would ameliorate the high NADH/NAD$^+$ ratio observed in MPC-expressing cells, we supplemented growth media with exogenous aspartate and assessed cellular redox status. We found that excess aspartate reduced the cellular NADH/NAD$^+$ ratio in these cells such that it was comparable with control cells (*Figure 6k*). We observed similar results when we treated these cells with NMN or UK5099 (*Figure 6k*). Moreover, all these treatments increased protein synthesis in MPC-expressing cells (*Figure 6l, m*).

## Mitochondrial pyruvate import reduces the size of rat primary hepatocytes

Although HepG2 cells exhibit some features of hepatocytes, they are a transformed, immortalized, and proliferative hepatocellular carcinoma cell line. We wanted to test whether the MPC expression phenotypes that we observed in *Drosophila* fat bodies and HepG2 cells could be recapitulated in a

more physiologically relevant mammalian model. We chose primary rat hepatocytes, which have been used extensively to interrogate hepatocyte cell metabolism and signaling and also have the advantage of being genetically tractable. We expressed MPC1 and MPC2 in cultured primary rat hepatocytes (*Figure 7a*), and, consistent with our results in other systems, we found that expression of MPC reduced cell size (*Figure 7b, c*) and decreased protein synthesis (*Figure 7d*). MPC-expressing primary hepatocytes also had a higher NADH to NAD$^+$ ratio in both the cytoplasmic and mitochondrial fractions (*Figure 7e*). We assessed gluconeogenesis in primary hepatocytes by quantifying glucose in the culture media following incubation with the gluconeogenic precursors, pyruvate, and lactate. Glucose production was higher in MPC-expressing hepatocytes compared with controls (*Figure 7f*). Treatment with UK5099 eliminated this effect and reduced glucose production to a similar rate in both cells (*Figure 7f*). These results demonstrate that augmented mitochondrial pyruvate in hepatocytes and related cells in *Drosophila* drives a metabolic program that results in an increased NADH/NAD$^+$ ratio. This scenario results in accelerated gluconeogenesis, decreased protein synthesis, and reduced cell size.

## Discussion

We investigated whether pyruvate metabolism influences biosynthetic capacity and cell size in the *Drosophila* fat body and in HepG2 spheroids. During the third instar phase of *Drosophila* larval growth, when cells are rapidly expanding in size, we observed a profoundly decreased expression of the MPC in liver-like fat body cells. We found that this rewiring of pyruvate metabolism is essential for cell growth, as forced maintenance of MPC expression resulted in dramatically smaller cells. By combining *Drosophila* genetic analyses and metabolomics studies in HepG2 cells and spheroids, we demonstrated that excess mitochondrial pyruvate elevates the cellular NADH/NAD$^+$ ratio and redirects carbohydrate metabolism to favor gluconeogenesis over glycolysis (*Figure 7g*). This shift reduces the availability of oxaloacetate for aspartate and glutamate biosynthesis, triggering a broader imbalance in amino acid abundances within the cell. We conclude that altering the fate of pyruvate to support biomass accumulation is required for the cell size expansion that occurs during fat body development. We speculate that this phenomenon also applies to the mammalian liver, which is the closest analog of the *Drosophila* fat body, as both HepG2 cells and primary rat hepatocytes show similar effects following ectopic MPC expression.

Why does simply reorienting the metabolism of pyruvate have this profound effect on cell growth? Our data suggest that a central mediator of the phenotype is an elevated NADH/NAD$^+$ ratio, which likely results from MPC expression driving an acceleration of the TCA cycle, as evidenced by an increase in the abundances of M+2 isotopomers of TCA cycle intermediates. Although the increase in labeled succinate, fumarate, and malate is modest, it occurs despite a reduction in glycolytic labeling. This suggests that less labeled pyruvate feeds the TCA in MPC-expressing cells compared with controls and that we are likely underestimating the actual increase in flux. The TCA cycle generates NADH and appears to do so more actively in cells with ectopically expressed MPC. However, our oxygen consumption data suggest that the oxidative phosphorylation system in these cells does not have the capacity to increase its activity in response to the enhanced availability of mitochondrial pyruvate and an increased NADH/NAD$^+$ ratio. As a result, the increased TCA cycle flux and limited ETC activity together elevate the NADH/NAD$^+$ ratio in both the mitochondria and cytoplasm, disrupting cellular redox balance leading to a rewiring of cellular metabolism.

This redox situation causes two distinct but related perturbations that appear to both contribute to decreased cell growth. First, we observed a clear depletion of amino acids and evidence of decreased synthesis of amino acids that are primarily derived from TCA cycle intermediates (aspartate, glutamate, and proline). A reduced NAD$^+$ pool impairs the capacity of cells to synthesize aspartate (*Birsoy et al., 2015*; *Sullivan et al., 2015*; *Sullivan et al., 2018*), which is used to synthesize glutamate and proline and which plays a crucial role in maintaining redox balance in both the mitochondria and cytoplasm (*Alkan et al., 2018*; *Holeček, 2023a*; *Lieu et al., 2020*; *Wei et al., 2020*; *Yoo et al., 2020*). Replenishing any of these TCA cycle-derived amino acids via genetic or nutritional increases in their availability was sufficient to reverse the effect of ectopic MPC expression on cell size. Thus, amino acid depletion is a key driver of the small size phenotype. Second, the increased NADH/NAD$^+$ ratio also drives a particular metabolic program that favors the conversion of mitochondrial pyruvate to oxaloacetate and a subsequent increase in gluconeogenesis. This program is enforced by allosteric

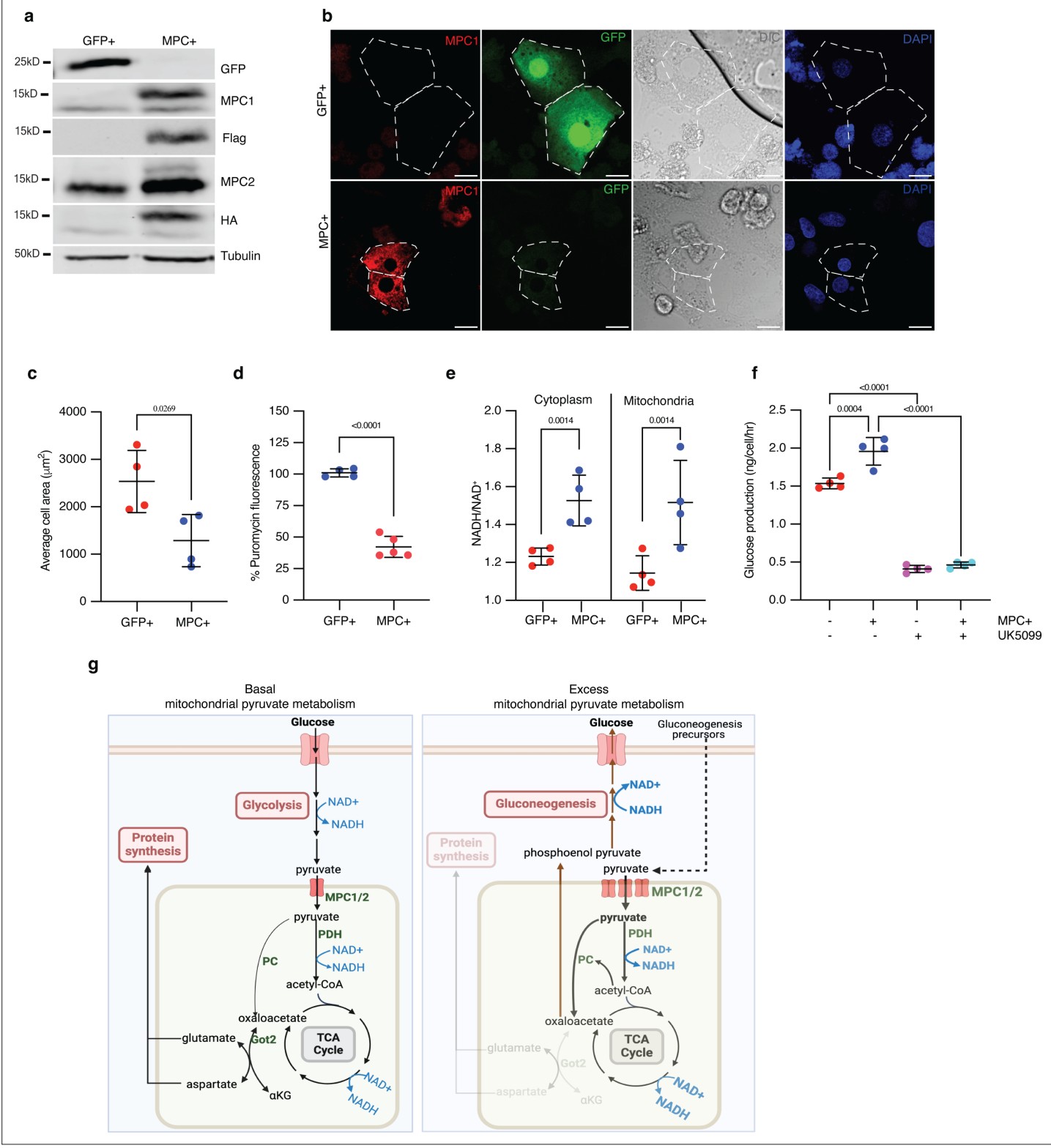

**Figure 7.** Increased mitochondrial pyruvate transport rewires metabolism and redox status to reduce protein synthesis and cell size in rat primary hepatocytes. (**a**) Western blot analysis of MPC expression in cultured rat primary hepatocytes. (**b**) Representative confocal images of rat primary hepatocytes expressing exogenous GFP or MPC. DIC images and DAPI staining of nuclei are also shown. Cell boundaries are marked with dotted lines. The scale bar represents 20 μm. (**c**) Quantification of the areas of GFP and MPC expressing primary hepatocytes. Data is presented as mean ± s.d. of six biological replicates, with 20 hepatocytes analyzed for GFP and MPC+. (**d**) Fluorescence intensity of puromycin-labeled nascent proteins in

*Figure 7 continued on next page*

*Figure 7 continued*

GFP and MPC+ primary hepatocytes. Data is presented as mean ± s.d. of six biological replicates, with 20 hepatocytes analyzed for each condition. (**e**) Quantification of glucose in the culture media of primary hepatocytes transfected with GFP or MPC constructs, conditioned with 20 mM lactate and 2 mM pyruvate for 4 hr. 10 µM UK5099 was used to inhibit MPC's downstream impact on gluconeogenesis. Data is presented as mean ± s.d. of three biological replicates. (**f**) NADH/NAD$^+$ ratios in the cytoplasmic and mitochondrial fractions of GFP or MPC+ primary hepatocytes. Data is presented as mean ± s.d. of three biological replicates. (**g**) Schematics illustrating the metabolic consequences of excess mitochondrial pyruvate in hepatocytes. Under normal conditions, mitochondrial pyruvate fuels the TCA cycle, maintaining redox balance and generating sufficient amino acids for cellular homeostasis. However, when mitochondrial pyruvate transport is increased and excess pyruvate is metabolized, both mitochondrial and cytoplasmic redox states are altered. This excess pyruvate enhances the activities of pyruvate carboxylase (PC), pyruvate dehydrogenase (PDH), and the TCA cycle, leading to an elevated NADH/NAD$^+$ ratio. The oxaloacetate produced by PC is converted into phosphoenolpyruvate via PEPCK2, promoting gluconeogenesis. This shift reduces the availability of aspartate and related amino acids necessary for protein synthesis, ultimately resulting in a reduction in cell size without impacting the canonical cell growth signaling pathways. Panel g was created with BioRender.com.

The online version of this article includes the following source data for figure 7:

**Source data 1.** Source data related to *Figure 7*.

**Source data 2.** PDF files containing original western blots for *Figure 7a*, indicating the relevant bands and treatments.

**Source data 3.** Original files for western blot analysis displayed in *Figure 7a*.

**Source data 4.** Original files for western blot analysis displayed in *Figure 7a*.

**Source data 5.** Original files for western blot analysis displayed in *Figure 7a*.

regulation via NADH, acetyl-CoA, and ATP, which act on PC and several enzymes of glycolysis and gluconeogenesis. It also appears to be important for the small size of MPC-expressing cells since loss of any of several steps along the gluconeogenic pathway, particularly PC and PEPCK2, mitigates or eliminates the cell size phenotype observed with ectopic MPC expression in fat body cells and HepG2 spheroids. It is important to note that gluconeogenesis is a critical function of these cells, and its rate is carefully controlled in response to varying physiological stimuli. Surprisingly, constitutive MPC expression is sufficient to supersede the control enacted by physiological and hormonal signals and enact the loss of biomass and impaired cell size.

We have previously shown that loss of the MPC enhanced the stem cell identity and proliferation of intestinal stem cells, whereas ectopic MPC expression had the opposite effect (*Bensard et al., 2020*; *Schell et al., 2017*). Such observations are consistent with our findings that MPC expression inversely correlates with biomass accumulation. More recent data demonstrated that enhanced MPC activity prevented the increase in cell size that occurs in cardiomyocytes in response to hypertrophic signaling (*Cluntun et al., 2021*; *Fernandez-Caggiano et al., 2020*). In these cases, the fate of mitochondrial pyruvate that determines cell growth is its oxidation in the TCA cycle. In contrast, our studies of hepatocytes and related cells described herein demonstrate that the fate of pyruvate that suppresses growth in cell size is not oxidation by the TCA cycle, but rather the production of glucose, which starts with its conversion to oxaloacetate by PC in the mitochondria. Cardiomyocytes have low expression of PC compared with liver, where this enzyme (as well as others including PEPCK2) serves a vital role during fasting by producing glucose, which fuels the brain and other organs that require glucose for their survival and function (*Hatting et al., 2018*; *Petersen et al., 2017*; *Rui, 2014*). This is an example of how the unique metabolic physiology of specialized cells plays a critical role in maintaining organismal homeostasis.

One conclusion from the data presented herein is that a cell's metabolic program plays a decisive role in determining the growth of that cell. It was striking how rapidly expression of the MPC impacted cell size in the HepG2 model. Essentially, as soon as MPC overexpressionbecame detectable, the population of cells started to show a decrease in size. Notably, the decrease in cell size following ectopic MPC expression occurred despite the upregulation of multiple pro-growth signaling networks. It is unclear why these networks are hyperactivated in this context as there is no evidence from our data indicating any nutrients are in excess abundance. We hypothesize that there may be regulators of cell size that recognize when cells are inappropriately small and engage these pathways to increase cell size. mTORC1, PI3K, and Myc pathways typically promote biomass accumulation and increased cell size but fail to do so when mitochondrial pyruvate is elevated. MPC expression reduced the abundance of amino acids, and this appears to play a dominant role in impairing protein synthesis and preventing the cell growth effects expected following hyperactivation of the mTORC1 and Myc

pathways. Thus, our data suggest that the metabolic fate of pyruvate can override canonical pathways that mediate cell size—such as mTORC1 signaling.

We demonstrate that the appropriate partitioning of pyruvate metabolism maintains the redox state of a cell to support the accumulation of biomass that is necessary for its specialized function. Increased mitochondrial pyruvate metabolism in cells from the fly 'liver-like' fat body disrupts these processes, causes cells to perform excess gluconeogenesis, and prevents cell growth. As a result, the *Drosophila* larvae became hyperglycemic and experienced developmental delay. The abundances of MPC1 and MPC2 are upregulated in mouse livers during starvation and in high-fat diet conditions, which correlates with increased rates of gluconeogenesis in both circumstances. Conversely, loss of the MPC in the liver impairs gluconeogenesis (*Gray et al., 2015*; *McCommis et al., 2015*; *Yiew and Finck, 2022*). Moreover, liver dysfunction in diabetes and hepatic steatosis is driven by reductive stress and an elevated NADH/NAD$^+$ ratio (*Goodman et al., 2020*; *Jokinen and Luukkonen, 2024*). We are intrigued by the possibility that the fate of pyruvate might have profound consequences on the redox state and gluconeogenic capacity of the mammalian liver, including by functioning as part of the metabolic milieu that drives unrestrained gluconeogenesis in diabetes. Our discovery that mitochondrial pyruvate regulates the cellular redox state, thereby controlling biosynthesis, offers insights for developing therapeutic strategies for these and other diseases.

## Materials and methods
### *Drosophila* strains and handling

*Drosophila melanogaster* stocks were maintained at 25°C on semi-defined fly food composed of 20 g agar, 80 g baker's yeast, 30 g sucrose, 60 g glucose, 0.5 g MgSO$_4$, 0.5 g CaCl$_2$, 11 ml tegosept, and 6 ml propionic acid. This was the base medium for all *Drosophila* experiments, but specific fly food modifications are mentioned in text and figure legends. To induce clones in fat bodies, synchronized eggs were transferred to 29°C for 4 days until dissection. For experiments involving genetic manipulation of all fat body cells, *tubGal80$^{ts20}$* was used to restrict activity of *CG-Gal4* at 18°C for 120 hr (57 hr equivalent time at 25°C) and larvae were shifted to 29°C till dissection at specified time points.

Following fly stocks were procured from Bloomington stock center *UAS-MPC1-P2A-MPC2* (28812582), *CG-Gal4; tubGal80$^{ts20}$, hs-Flp1.22* (BDSC, #77928), *Act >CD2>Gal4, UAS-GFP* (BDSC, #4413), *Act >CD2>Gal4, UAS-RFP* (BDSC, #30558), *UAS-S6K$^{STDETE}$* (BDSC, #6914), *UAS-Rheb$^{PA}$* (BDSC, #9689), *UAS-Myc$^{OE}$* (BDSC, #9675), *UAS-Myc$^{RNAi}$* (BDSC, #25783), *UAS-PI3K93E.$^{Excel}$* (BDSC, #8287), *UAS-PI3K93E$^{A2860C}$* (BDSC, #8288), *UAS-TSC1$^{RNAi}$* (BDSC, #31314), *UAS-hPC* (BDSC, #77928), *UAS-PC$^{RNAi}$* (BDSC, #56883), *UAS-PEPCK2$^{RNAi}$* (BDSC, #36915), *UAS-FBPase$^{RNAi}$* (BDSC, #51871), *UAS-Pdk$^{RNAi}$* (BDSC, #35142), *UAS-NMNAT* (BDSC, #37002), *UAS-CintNDX* (BDSC, #93883), *UAS-ScerNDI1* (BDSC, #93878), *UAS-Slif* (BDSC, #52661), *UAS-GOT2$^{RNAi}$* (BDSC, #78778). From VDRC *UAS-MPC1$^{RNAi(KK)}$* and from NIG *UAS-Pdha$^{RNAi}$, UAS-Dlat$^{RNAi}$ (PDH E2)* were purchased.

### Generation of overexpression and CRISPR mutant fly stock

*Pc* and *Pepck2* deletion fly stocks were generated using CRISPR–Cas9 as described (*Gratz et al., 2014*). We targeted specific nucleotide sequences of the genes of interest through homology-directed recombination using two guide RNAs and inserted a dsRed construct that expresses in adult eyes to facilitate the selection of mutant flies. We deleted exon 1 to exon 5 using guide RNAs for *Pc$^-$KO*: 5′ guide ATACATTTAAGTCCTAGGC; 3′ guide TCGATTGATCCTGGAAACA. *Pepck2$^-$KO*, being a single exon coding sequence, we generated complete deletion by using guides 5′ guide AAAG GGTGCACATCTGTGA; 3′ guide TTTGGGGCGTGGCCTAGAC. The plasmids were injected into y[1] M{GFP[E.3xP3]=vas-Cas9.RFP}ZH-2A w[1118] (BDSC, #55821) fly stock embryos (Bestgene) and one *Pc-KO* and five *Pepck2-KO* flies were picked up, confirmed for the dsRed expression in adult eyes and used in subsequent experiments.

To overexpress *Got2*, *Got2* cDNA was amplified from RNA extracted from larval fat bodies using primer 5′ GAATTC ATGAGTAGAACCATTATTATGACGCTTAAGGAC, 3′ CTCGAG CTTGGTAACCTT GTGTATGCTCTCAGCCAGG. The cDNA was then cloned into a pUAST-aatB plasmid using EcoRI and XhoI restriction enzymes, and the construct was injected into pBac{yellow[+]-attp-9A}VK00005 (BDSC, #9725) embryos to obtain the insertion.

## Mosaic analysis and phalloidin staining for cell size analysis

Fat body clones were induced by the leaky expression of heat shock flippase 1.22 during embryonic stages, and the 2D size of fat body cells was analyzed at 120 hr after egg laying (AEL) using fluorescence microscopy. Cell size analysis was conducted as reported earlier (*Toshniwal et al., 2019*). Fat bodies were dissected from larvae at the specified time points in 1X PBS buffer (pH 7.2, Invitrogen, #10010049) and fixed in 8% paraformaldehyde (Sigma-Aldrich, # P6148) for 1 hr at room temperature (RT). The tissues were then washed twice with 0.1% PBT (0.1% Triton X-100 (Sigma-Aldrich, # X100) in 1X PBS buffer) for 10 min each. Subsequently, the tissues were incubated with Rhodamine Phalloidin (Thermo Scientific, # R418) at a 1:400 dilution in 1X PBS buffer for 2 hr at RT. After staining, the tissues were washed once with 0.1% PBT and then with 1X PBS before being mounted in DAPI-supplemented VectaShield (Vector Labs, #H1200). Representative images were captured using a Laser Scanning Confocal Microscope (LSM 880, Carl Zeiss). For cell size analysis, images were captured with a fluorescence microscope (Carl Zeiss, Axio Vision) at 20× magnification, focusing on a plane where all nuclei were in focus. The 2D areas of fat body cells were measured using FIJI software. Cell membranes stained with Rhodamine Phalloidin were traced using the freehand tool, and the area of approximately 20–25 GFP-positive cells from fat bodies collected from about five animals was measured as one replicate. All cell size analyses were conducted in a blinded manner.

## Immunofluorescence on fat body clones

Larval fat bodies at 120 hr after egg laying (AEL) at 29°C were dissected in 1X PBS buffer (pH 7.2), fixed in 8% paraformaldehyde for 1 hr at RT, and then washed three times in 0.1% PBT (0.1% Triton X-100 in 1X PBS buffer) for 10 min each. The tissues were blocked in 10% normal goat serum (NGS, Jackson ImmunoResearch Laboratories, #005-000-121) in 0.1% PBT for 1 hr at RT, followed by incubation with primary antibodies overnight at 4°C. Secondary antibody incubation was performed for 2 hr at RT. Following three to four washes in 0.3% PBT (10 min each) at RT, the tissues were mounted in DAPI-supplemented Vectashield (Vector Laboratories, #H1200). Images were captured using a Laser Scanning Confocal Microscope (LSM 880, Carl Zeiss).

## Image acquisition and analysis

For all experiments, confocal images were captured using the same laser power and identical settings, with z-stacks of the dissected tissue taken at 1 μm intervals along the y-axis. The images were analyzed using Fiji software, where a similar number of z-stacks focusing on the nuclei were projected at mean intensities. Using the freehand tool, cell membranes were outlined, and the mean fluorescence intensity (mean gray value) for each cell was recorded. To account for background fluorescence, the average mean gray values of the background (measured from regions of interest [ROIs] of the same size) were subtracted from the recorded mean gray values of the cells. The resulting mean gray values, adjusted for background fluorescence, were then normalized, and the percent normalized mean gray values (with s.d.) of GFP-negative and GFP-positive cells were plotted.

## Measurement of protein synthesis in fat bodies

Protein synthesis in *Drosophila* fat bodies was analyzed using the Click-iT Plus OPP Alexa Fluor 594 kit (Molecular Probes, #C10457). Fat bodies were dissected at 120 hr AEL in Shields and Sang M3 Insect Media (SSM3, Sigma, #S8398) and incubated with 5 mM OPP in SSM3 media for 30 min at RT. Tissues were then washed three times with 1X PBS for 10 min each and fixed in 8% paraformaldehyde for 1 hr at RT. After fixation, the tissues were washed twice with 0.1% PBT (0.1% Triton X-100 in 1X PBS) supplemented with 0.5% bovine serum albumin (BSA) for 10 min each. The OPP was developed for 30 min at RT using the Click-iT reaction mixture, which included 88 μl of OPP Reaction Buffer, 20 μl of Copper (III) Sulfate (component D), 2.5 μl of component B, and 100 μl of Buffer Additive (component E), following the manufacturer's instructions. The tissues were then washed twice in Reaction Rinse Buffer for 10 min each, followed by two washes in 0.1% PBT supplemented with 0.5% BSA for 10 min each at RT. After a final wash in 1X PBS, the fat bodies were mounted in DAPI-supplemented Vectashield. Images were captured using a confocal microscope (LSM 880, Zeiss).

## LipidTOX staining in larval fat body tissues

At 120 hr AEL, 3rd instar larvae were dissected in 1X PBS. The fat bodies were fixed in 8% paraformaldehyde for 1 hr at room temperature. After fixation, tissues were rinsed twice with 1X PBS. They were then incubated for 30 min in a 1:100 dilution of LipidTOX Red (Invitrogen, #H34351) in PBS, followed by two additional rinses with PBS. The tissues were mounted in DAPI-supplemented Vectashield and imaged using a confocal microscope. To quantify lipid droplet size, the diameters of lipid droplets from 35 fat body clones were measured using FIJI software.

## EdU incorporation in fat bodies

At 120 hr AEL, fat bodies were dissected in 1X PBS and incubated with 5 µM 5-ethyl-2'-deoxyuridine (EdU) in 1X PBS for 30 min at room temperature. The tissues were then washed three times with 1X PBS for 10 min each, followed by fixation in 8% paraformaldehyde for 1 hr at room temperature. After fixation, the tissues were washed with 0.1% PBT supplemented with 0.5% bovine serum albumin. EdU detection was performed using Click-iT Plus EdU Alexa Fluor 594 (Molecular Probes, #C10639) for 30 min at room temperature, according to the manufacturer's instructions. Following development, the fat bodies were mounted in DAPI-supplemented Vectashield. Images were captured using a confocal microscope (LSM 880, Zeiss).

## RNA isolation and RNA sequencing

To collect fat body tissues, $w^{1118}$ flies were mated at 25°C. The following day, flies were starved for 1 hr, and eggs were collected every 2 hr. The first collection was discarded, and the subsequent two collections were incubated at 25°C. After 20 hr, early hatched 1st instar larvae were discarded, and 1st instar larvae collected over a 2-hr period were kept at 25°C. Fat bodies from 10 male larvae at the specified time points were dissected and preserved in 1X PBS. For RNA isolation from MPC-expressing fat bodies at 120 hr AEL, RNA was extracted and purified using the NucleoSpin RNA kit (Takara Bio USA, Inc, #740955.50) with on-column DNA digestion, as per the manufacturer's instructions. Four independent samples for each time point were prepared for sequencing.

Library preparation for poly(A)-selected RNAs was carried out using the Illumina RNA TruSeq Stranded mRNA Library Prep Kit with oligo-dT selection. Sequencing was performed using the Illumina NovaSeq Reagent Kit v1.5 (150 bp paired-end reads) at the High-Throughput Genomics Core Facility at the University of Utah. The raw sequencing data were analyzed using the BDGP6.28 genome and gene feature files. Differentially expressed genes were identified using DESeq2 version 1.30.0 with a 5% false discovery rate. RNA quality control, library preparation, and sequencing were performed by the University of Utah Huntsman Cancer Institute High Throughput Genomics and Bioinformatics Shared Resource.

## QPCR

500 ng RNA was used to make cDNA using Superscript II reverse transcriptase (Molecular Probes, #18064-022), dNTP (Molecular Probes, #18427-088) and oligodT (Molecular Probes, #18418012). QPCR analyses were performed on cDNA as described using PowerUp SYBR Green Master Mix (Applied Biosystems, #2828831) on QuantStudio 7 Flex (Applied Biosystems) instrument. Fold changes in transcript level were determined using the ΔΔCt method. Transcript levels were normalized to *rp49*. Each experiment was performed using four to five independent samples. The following primers were used to do qPCR. A list of primers is provided in Appendix 3.

## Hemolymph glucose and fat body glucose measurement

Glucose concentrations in larval hemolymph and fat bodies were measured as previously described (*Ugrankar Banerjee et al., 2023*). To isolate hemolymph, 10 third instar larvae were selected from culture tubes, thoroughly washed to remove any food residues, and then dried. Hemolymph was collected by bleeding the larvae on a parafilm strip using Dumont 5 forceps (Fine Science Tools, #11254-20). Two microliters of the colorless hemolymph were transferred to a 96-well plate and mixed with 200 µl of Autokit Glucose reagent (Wako, #997-03001). For measuring intercellular glucose, fat bodies were dissected from 10 larvae per genotype and homogenized in approximately 300 µm of ice-cold 1X PBS using a 29G1/2 syringe. The lysates were inactivated at 70°C for 10 min, then centrifuged at maximum speed at 4°C. Thirty microliters of the lysate were mixed with 170 µl of Autokit Glucose

reagent. The plates were incubated at 37°C for 30 min, and absorbance was measured at 600 nm. Glucose concentrations were determined based on the absorbance values recorded for glucose standards. For trehalose measurement from hemolymph, 2 µl of hemolymph was diluted five times, and 8 µl of this diluted hemolymph was incubated with 5 µl 8x porcine kidney trehalase (Sigma) at 37°C overnight. 10 µl of this mix was used for glucose measurement, as described above.

## HepG2 cells – knockdown and overexpression strategies

HepG2 cells were purchased from ATCC and maintained in EMEM supplemented with 10% FBS and 1% PenStrep at 37°C in a 5% $CO_2$ atmosphere. HepG2 verification was provided by ATCC and authenticated by STR profiling. Mycoplasma testing was performed every month, and all cell lines used were mycoplasma-free.

For inducible overexpression of human MPC1 and MPC2, the HA-MPC2-P2A-T2A-MPC1-FLAG sequence was cloned into the pLVX-TetOne-Zeocin vector. Lentiviral particles were generated using Gag-Pol, pMD2.G, and VSVG packaging plasmids.

Viral particles were produced by co-transfecting 293T cells with the respective packaging plasmids using polyethylenimine (PEI, Sigma, #765090) as the transfection reagent at a 3:1 mass ratio of PEI to DNA. The virus-containing medium was collected 48 hr post-transfection, filtered through a 0.45-µm filter, and added to HepG2 cells cultured in normal medium, along with polybrene (Sigma, #P1240000) at a concentration of 10 µg/ml. Transduced cells were selected with 10 µg/ml Zeocin (Gibco, #R25001) for 1 week, and the level of overexpression was assessed by western blotting and presented with respective images.

For PC over-expression, PC coding sequence was amplified from mRNA isolated from HepG2 cells using primers 5′ GAATTC ATGCTGAAGTTCCGAACAGTCCATGGG, 3′ GGATCC CTCGATCTCCAG GATGAGGTCGTCACC and cloned into pLenti-CMV-Blast. Similarly, SLC1A3 cDNA was amplified with primers 5′ GGATCC ATGACTAAAAGCAATGGAGAAGAGC, 3′ CTACATCTTGGTTTCAATGTCGAT GG and GOT2 coding sequence was amplified using primers 5′ GGATCC ATGGCCCTGCTGCACT CCGG, 3′ TCTAGA CTTGGTGACCTGGTGAATGGCATGG and cloned in pLenti-CMV-Blast (Addgene, #17486). The coding sequences of NDI (Addgene, #72876) and d2GFP (Addgene, #115665) were also cloned into pLenti-CMV-Blast vector. Cells were selected on 3 µg/ml blasticidin (Gibco, A1113903) for 3 days. To generate knockout cells, the following gRNAs were used. PCg5e-GAAGCCTATCTCATCG GCCG CGG, PCg6e-CGAAGTCCGCTCGCTCAGAG AGG, PEPCK2g2e-ATCTCCACTAAGCACT CGCA GGG, PEPCK2g3e-CATGCGTATTATGACCCGAC TGG, GOT2ga-GAGTGGCCGGGTAAGC TGAGCAG AGG, GOT2gb-GGAGTGGACCCGCTCCGGAACAG TGG. The guides were annealed and clones in lentiCRISPRv2 with blasticidin (Addgene, #83480) resistance using BsmBI.

## HepG2 cells and 2D cell size analysis

HepG2 cells were cultured on 12 mm coverslips in a 24-well plate at a low density of 10,000 cells per well in Human Plasma-Like Medium (Gibco, #A4899101) supplemented with 10% FBS (Sigma, F0926) and 1% PenStrep (Thermo, #15140). The next day, treatments were initiated as described in the figure legends, including 1 µg/ml doxycycline (Sigma, #D5207), 10 µM UK5099 (Sigma, PZ0160-5MG), 10 mM glycine (Sigma, #G7403), 5 mM alanine (Sigma, #A7469), 5 mM asparagine (Sigma, #A4159), 5 mM aspartic acid (Sigma, #A7219), 5 mM glutamic acid (Sigma, #G8415), 5 mM proline (Sigma, #P5607), 5 mM serine (Sigma, #S4311), 10 µM AZD7545 (MedChemExpress, #HY-16082), 1 mM dichloroacetate (Sigma, #347795), 100 nM duroquinone (Sigma, #D223204), and 2 nM gramicidin (Sigma, #G5002).

For time-course experiments in *Figure 3*, cells were fixed at 2, 4, 6, 8, 10, 12, 18, and 24 hr after doxycycline treatment using 4% paraformaldehyde in 1X PBS for 20 min at RT. For all other experiments, cells were fixed 24 hr after doxycycline treatment. Following fixation, cells were washed once with 0.1% PBT (0.1% Triton X-100 in 1X PBS) for 10 min and incubated with Rhodamine Phalloidin at a 1:400 dilution in 1X PBS buffer for 20 min at RT. After a couple of washes in 1X PBS, cells were mounted in DAPI-supplemented VectaShield.

Images were captured using a fluorescence microscope (Carl Zeiss, Axio Vision) focusing on the plane of the cellular nuclei at 20X magnification, where all nuclei were in focus, and the 2D area of HepG2 cells was measured using FIJI software. All cell size analyses were conducted in a blinded manner.

For 3D cell volume analysis, images were captured using a Laser Scanning Confocal Microscope (LSM 880, Carl Zeiss). The red fluorescence signal was used for 3D reconstruction, and cell volume was measured using an ImageJ Macro code in FIJI.

## Measurement of protein synthesis in HepG2 cells

HepG2 cells were cultured in either 2D monolayers or spheroid forms. At the desired time point, cells were treated with 20 µg/ml puromycin (Sigma, #P4512) in Human Plasma-Like Medium (HPLM) for 30 min at 37°C. Following treatment, cells were washed with 1X PBS, and proteins were extracted using 1X RIPA buffer at 4°C. Protein concentrations were quantified, and 15 µg of total protein was separated by SDS-PAGE using standard methods. Proteins were then transferred onto a nitrocellulose membrane, and puromycin-labeled peptides were detected using an anti-puromycin [3RH11] antibody (Kerafast, #EQ0001) followed by incubation with an appropriate secondary antibody.

Protein synthesis in HepG2 cells was also analyzed using the Click-iT HPG Alexa Fluor 594 kit (Molecular Probes, #C10429). HepG2 cells were grown on 12 mm coverslips at a density of 10,000 cells per well and treated with 1 µg/ml doxycycline for 24 hr. Cells were then incubated with 50 µM L-homopropargylglycine (HPG) in methionine-free DMEM (Gibco, #21013024) for 30 min at 37°C. After incubation, cells were washed twice with 1X PBS for 2 min each and fixed in 4% paraformaldehyde for 20 min at room temperature.

For OPP staining, appropriately fixed cells were washed twice with 0.1% PBT (0.1% Triton X-100 in 1X PBS) supplemented with 0.5% bovine serum albumin (BSA) for 10 min each. HPG was detected using the Click-iT reaction mixture, which included 88 µl of OPP Reaction Buffer, 20 µl of copper solution (component D), 2.5 µl of component B, and 100 µl of Buffer Additive (component E), following the manufacturer's instructions. Cells were washed twice in Reaction Rinse Buffer for 10 min each, followed by two washes in 0.1% PBT supplemented with 0.5% BSA for 10 min each at room temperature. After a final wash in 1X PBS, the coverslips were mounted in DAPI-supplemented Vectashield. Images of the cells were captured using a confocal microscope (LSM 880, Zeiss).

## Analysis of spheroids

HepG2 cells were cultured in ultra-low attachment 96-well plates (Costar, #7007) at a density of 10,000 cells per well in Human Plasma-Like Medium (HPLM) supplemented with 10% FBS and 1% penicillin–streptomycin. Treatments were applied as specified in the figure legends, including 1 µg/ml doxycycline, 5 mM aspartate, 5 mM glutamate, 10 µM UK5099, MEM non-essential amino acids (Gibco, #11140050), 100 nM nicotinamide riboside (Sigma, #SMB00907), and 10 µM NMN (Sigma, #N3501). Cells were incubated for 6 days at 37°C with $CO_2$ and $O_2$. Brightfield images of the spheroids were captured using a Zeiss Axio Observer Z1 microscope, and spheroid size was measured using FIJI software.

To quantify cell numbers, 12 spheroids from each condition (EV or MPC+) were pooled, dissociated by trypsinization, and the number of cells was counted using a CellQuant system (Bio-Rad).

For cell cycle analysis, cells were stained using Vybrant DyeCycle Violet stain (Molecular Probes, #V35003). After pooling and dissociating 12 spheroids from each condition (EV or MPC+), cells were centrifuged and resuspended in 200 µl of 5 µM Vybrant DyeCycle Violet stain in EMEM supplemented with 10% FBS. The staining was performed by incubating the cells at 37°C for 30 min, protected from light. Samples were analyzed using a BD Celesta flow cytometer with ~405 nm excitation and ~440 nm emission. The resulting FCS files were processed in FlowJo software, where the forward scatter of singlets was recorded, and median data were used to plot graphs. Vybrant dye staining was employed to assess the distribution of cells in the G1, S, and G2/M phases.

For apoptosis detection, an Annexin V/PI staining kit (Molecular Probes, #V13241) was used, with 5 µM camptothecin (MedChemExpress, #HY-16560) serving as a positive control. Sixteen spheroids were collected, dissociated with trypsin as described above, and resuspended in 200 µL of 1X annexin-binding buffer. Cells were incubated with 1 µl of Alexa Fluor 488 Annexin V (Component A) and 0.2 µl of 100 µg/ml PI working solution at room temperature for 15 min. Stained cells were then analyzed using a BD Canto flow cytometer, measuring fluorescence emission at 530 and 575 nm (or equivalent) with 488 nm excitation.

## Biomolecule separation and measurement

For this experiment, biomolecules were extracted and measured from both *Drosophila* larvae fat bodies and HepG2 cell spheroids. Fat bodies were dissected from 10 male larvae at 120 hr after egg laying (AEL) and collected in 150 µl of 1X PBS. The samples were lysed by performing three freeze–thaw cycles. A total of 24 spheroids were homogenized in 150 µl of radioimmunoprecipitation assay (RIPA) buffer (50 mM Tris-HCl, 1% NP-40, 0.5% sodium deoxycholate, 0.1% SDS, 150 mM NaCl, and 2 mM EDTA).

Lysates were centrifuged at $7500 \times g$ for 5 min to remove debris. The resulting supernatant was used for the following measurements. Triglyceride measurement: For triglyceride analysis, 30 µl of the supernatant was incubated for 10 min at 75°C. Following this, 10 µl of homogenate from the spheroids or 2 µl from fat bodies was added to 200 µl of Triglycerides Reagent (Thermo Fisher Scientific, #TR22421). The mixture was incubated for 10 min at 37°C in 96-well microplates with gentle shaking. Absorbance was measured at 550 nm using the Synergy Neo2 multimode plate reader (BioTek).

Protein measurement: Protein concentrations were determined by mixing 2 µl of the supernatant with 200 µl of BCA Protein Assay Reagent (Thermo Scientific, #23225). The mixture was incubated for 30 min at 37°C with gentle shaking in 96-well microplates. Absorbance was measured at 560 nm.

RNA and DNA separation: The remaining 100 µl of the supernatant was processed for RNA and DNA separation using TRIzol reagent (1 ml; Invitrogen, # 15596026). To separate RNA, 0.2 ml of chloroform was added to the sample, followed by centrifugation at $12,000 \times g$ for 15 min at 4°C. The RNA in the interphase was purified using 75% ethanol, following standard methods, and quantified using a NanoDrop spectrophotometer. DNA was extracted from the organic phase by adding 100% ethanol, followed by isopropanol precipitation. The resulting pellets were washed with 0.3 M guanidine hydrochloride in 95% ethanol, resuspended in 0.1 M sodium citrate in 10% ethanol (pH 8.5), and washed with 75% ethanol. Finally, the DNA pellets were resuspended in 0.1 ml of 8 mM NaOH by pipetting. The pH was adjusted to 7.2 with HEPES, and the DNA was quantified using a NanoDrop spectrophotometer.

## Measurement of NADH/NAD⁺ protocol

For HepG2 cells, $1 \times 10^6$ cells treated with 1 µg/ml dox for 24 hr were scraped in 1.5 ml tube. For NADH/NAD$^+$ ratio analysis from spheroids, 18 spheroids were treated with 1 µg/ml dox for 6 days, and then resuspended in 1 ml 1X PBS. Scraped cells or pooled spheroids were centrifuged ($13,500 \times g$, 10 s, 4°C) and resuspended in 250 µl lysis buffer. Fat bodies from 10 male larvae were dissected in 1X PBS and resuspended in the lysis buffer provided with the kit. To separate cytoplasmic and mitochondrial fractions by rapid subcellular fractionation, lysates were centrifuged ($13,500 \times g$, 10 s, 4°C) and the supernatant was collected for the cytosolic fraction, while the remaining pellet contained the mitochondria. Mitochondrial pellet was resuspended in 100 µl lysis buffer.

NADH to NAD$^+$ ratios were measured using Amplite Fluorimetric NAD$^+$/NADH ratio assay kit (AAT Bioquest, #15263) as directed by the instructions provided. Briefly, 25 µl of cytoplasmic or mitochondrial fractions were mixed with either NADH or NAD$^+$ extraction solution. Samples were incubated at 37°C for 15 min. Later, 25 µl of either NAD$^+$ or NADH extraction was added, which was followed by incubation with 75 µl mix of NADH sensor buffer and NAD$^+$/NADH recycling enzyme for 1 hr at RT. Fluorescence intensity was recorded at 540 nm excitation and 590 nm emission.

## Protein extraction and western blotting

HepG2 cells were directly scraped into RIPA supplemented with protease and phosphatase inhibitors (Roche Molecular, #04906845001). For spheroids, 18 spheroids were pooled, washed with 1X PBS, and then incubated in RIPA buffer with the protease and phosphatase inhibitor cocktail. After 45 min on ice with vortexing every 15 min, lysates were centrifuged at $16,000 \times g$ for 15 min at 4°C to remove insoluble material.

Protein concentration was measured using the Bicinchoninic Acid (BCA) protein assay (Thermo Fisher Scientific, 23225). Samples were mixed with 4x sample loading buffer and heated at 95°C for 5 min. Protein samples (15 µg) were separated on SDS–polyacrylamide gel electrophoresis (SDS–PAGE) at 20 mA per gel, transferred onto a 0.45 µm nitrocellulose membrane (GE Healthcare) using the Mini Trans-Blot module (Bio-Rad) at 120 V for approximately 2 hr.

The membrane was blocked with 5% non-fat milk (Serva) in Tris-buffered saline with 0.05% Tween 20 (TBS-T) for 1 hr. It was then incubated overnight with primary antibodies diluted in TBS-T. The next day, the membrane was washed with TBS-T and incubated with fluorophore-conjugated secondary antibodies in TBS-T for 1 hr. Following additional washes with TBS-T, fluorescence was detected using the Odyssey CLx imaging system (LI-COR Biosciences) and analyzed using FIJI software. Antibodies used are listed in the Key Resources Table.

## Steady-state metabolomic studies

Ten million HepG2 cells with MPC+ and EV control expression were grown in 2D culture and treated with 1 µg/ml dox in HPLM supplemented with 10% HPLM. After 24 hr, the culture medium was collected and quenched with 1:4 volume of 100% methanol. Cells were washed and quenched with 1 ml of 80% methanol in water. Cell lysates were then subjected to three rapid freeze-thaw cycles and then spun at 16,000 × g for 10 min at 4°C. The supernatants were evaporated using a SpeedVac concentrator. Each sample or treatment had 4–5 replicates.

## $^{13}$C-glucose tracing for M+3 vs. M+2 ratio of TCA cycle metabolism

Ten million HepG2 cells with MPC+ and EV control expression were grown in 2D culture and were treated with 1 µg/ml dox in HPLM. After 24 hr, the culture medium was changed to $^{13}$C-glucose tracing media: glucose-free HPLM supplemented with 4.5 g/l $^{13}$C glucose and 10% dialyzed FBS. 0, 1, 2, and 4 hr later, cells were washed and quenched with 1 ml of 80% methanol in water. Cells were scraped out in methanol, and lysates were then subjected to three rapid freeze–thaw cycles and then spun at 16,000 × g for 10 min at 4°C. The supernatants were evaporated using a SpeedVac concentrator.

## $^{13}$C-lactate tracing for gluconeogenesis assay

Ten million HepG2 cells with MPC+ and EV control expression were grown in 2D culture and treated with 1 µg/ml dox in DMEM without glucose, without glutamine (Gibco, #A1443001) with 10% dialyzed FBS. After 16 hr, the culture medium was replaced with 6 ml of 1 µg/ml dox in DMEM without glucose, without glutamine, and without FBS. 3 hr later, the culture medium was replaced with 6 ml of 1 µg/ml dox and 20 mM lactate (Sigma, #490040) containing DMEM without glucose or glutamine or FBS. After 4 hr, 300 µl culture medium was collected and quenched with 1:4 volume of 100% methanol. Cells were washed and quenched with 1 ml of 80% methanol in water. Cell lysates were then subjected to three rapid freeze-thaw cycles and then spun at 16,000xg for 10 min at 4°C. The supernatants were evaporated using a SpeedVac concentrator. Each cell type had four to five replicates.

## Gas chromatography–mass spectrometry derivatization

Dried metabolites were derivatized and prepared for Gas chromatography following standard methods. Dried samples were resuspended in 30 µl of anhydrous pyridine with methoxamine hydrochloride (10 mg/ml) and incubated at RT overnight. The next day, the samples were heated at 70°C for 15 min and centrifuged at 16,000 × g for 10 min. The supernatant was transferred to a pre-prepared gas chromatography–mass–mass spectrometry autoinjector vial with 70 µl of *N*-(*tert*-butyldimethylsilyl)-*N*-methyltrifluoroacetamide (MTBSTFA) derivatization reagent. The samples were incubated at 70°C for 1 hr, after which aliquots of 1 µl were injected for analysis. Samples were analyzed using either an Agilent 6890 or 7890 gas chromatograph coupled to an Agilent 5973N or 5975C Mass Selective Detector, respectively. The observed distributions of mass isotopologues of glucose, pyruvate, citrate, succinate, aspartate, glutamate, malate, fumarate, and phosphoenolpyruvate were corrected for natural abundance.

## Liquid chromatography–mass spectrometry

Following standard methods, dried metabolites were resuspended in 100 µl of 0.03% formic acid in analytical-grade water, vortexed, and centrifuged to remove insoluble material. 20 µl of supernatant was collected and injected to AB SCIEX QTRAP 5500 liquid chromatography/triple quadrupole mass spectrometer (Applied Biosystems SCIEX). Chromatogram review and peak area integration were performed using MultiQuant (version 2.1, Applied Biosystems SCIEX). The peak area for acetyl CoA, ADP, ATP, NAD$^+$, and NADH was normalized against the total ion count of that sample.

## Oxygen consumption rate

Oxygen consumption rates (OCR) were measured using an XFe96 Extracellular Flux Analyzer (Agilent) according to the manufacturer's instructions. Cells were plated at a density of 60,000 cells per well in Seahorse microplates (Agilent) and allowed to adhere for 6 hr. MPC was induced with 1 µg/ml doxycycline overnight. Afterward, the cell culture media was removed and replaced with Seahorse assay media, which consisted of DMEM supplemented with 4 mM glutamine. OCR was assessed under basal conditions and following sequential injections of 1 mM glucose, oligomycin (2 µM), FCCP (0.5 µM), and a mix of rotenone plus antimycin A (2 µM each). Immediately after the measurements, cells were lysed in RIPA buffer, and the protein concentration was used to normalize the OCR data.

## Primary hepatocyte cultures and analysis

Rat primary hepatocytes (Lonza, #RWCP01) were thawed and plated in a 24-well plate at a density of 0.16 million cells per well in Hepatocyte Plating Medium (Lonza, #MP100). Four hours post-plating, the medium was replaced with Hepatocyte Basal Medium (HBM) supplemented with BCM Single-Quots (Lonza, #CC-4182), including Bovine Pituitary Extract (BPE), Insulin, Hydrocortisone, Genta-micin/Amphotericin-B (GA), Transferrin, and human Epidermal Growth Factor (hEGF).

Twenty-four hours after plating, the hepatocytes were transfected with 1 µg of either pT3.GFP or pT3.MPC2Flag-P2AT2A-MPC1HA using Lipofectamine 3000 (Invitrogen, #L3000001). Forty-eight hours post-transfection, cells were lysed in 1X RIPA buffer, and Western blot analysis was performed to confirm the overexpression of MPC1 and MPC2.

Forty-eight hours post-transfection, hepatocytes were fixed in 4% paraformaldehyde in 1X PBS. Following several washes with 0.1% PBT, cells were incubated in 5% BSA and stained with anti-MPC1 antibody overnight at 4°C. MPC1 was detected using a secondary anti-rabbit Alexa Fluor antibody. Cells were also stained with Phalloidin Red. Coverslips were mounted in DAPI-supplemented Vecta-shield, and images were captured using an LSM 880 confocal microscope. Cell area marked by Phal-loidin Red was quantified using differential interference contrast filter and analyzed with FIJI software.

For protein synthesis measurement, hepatocytes were incubated with 20 µg/ml puromycin for 30 min. Puromycin-tagged peptides were visualized by immunostaining with rabbit anti-puromycin antibody (1:500) and mouse anti-Flag M2 antibody (1:1000, Sigma, F1800), followed by the appro-priate secondary antibodies. Images were captured with the LSM 880 confocal microscope. Puromycin intensity was measured using FIJI, and the percent change in puromycin intensity in Flag-positive cells was compared to Flag-negative cells.

Gluconeogenesis was assessed using lactate and pyruvate as substrates. Forty-eight hours post-transfection, hepatocytes were incubated in low-glucose DMEM with 1 mM sodium pyruvate and 4 mM glutamine without FBS. After 16 hr, hepatocytes were treated with 200 ng glucagon in no-glu-cose DMEM without FBS for 3 hr. Cells were then cultured in media with 20 mM lactate and 2 mM pyruvate, with and without 10 µM UK5099, for 2 and 4 hr. Glucose levels in the media were measured using the Amplex Red Glucose Assay Kit (Invitrogen, #A22189) according to the manufacturer's instructions. Glucose production per cell per hour was plotted on a graph.

The compartmentalized NADH to NAD$^+$ ratio in hepatocytes was quantified 48 hr post-transfection using the Amplite Fluorimetric NAD$^+$/NADH Ratio Assay Kit as described previously.

## Acknowledgements

We thank the University of Utah core facilities, especially James Cox, PhD, at the Metabolomics core, Li Ying, PhD, at the Metabolic Phenotyping core, James Marvin, PhD, at the Flow Cytometry Core, Brian Dalley, PhD, at the High-Throughput Genomics Core, and the DNA/Peptide Synthesis Core. We thank members of the Rutter lab for discussion and feedback on the manuscript. This study was supported by 24POST1201210 to AGT; K00CA212445 to AJB; 1K99HL168312-01 to AAC, R01 CA228346 to CT and JR. JR is an Investigator of the Howard Hughes Medical Institute. The funders had no role in study design, data collection, and interpretation, or the decision to submit the work for publication.

## Additional information

### Funding

| Funder | Grant reference number | Author |
|---|---|---|
| American Heart Association | 10.58275/aha.24post1201210.pc.gr.190781 | Ashish G Toshniwal |
| National Institutes of Health | K00CA212445 | Alex J Bott |
| National Institutes of Health | 1K99HL168312-01 | Ahmad A Cluntun |
| National Institutes of Health | R01CA228346 | Carl S Thummel<br>Jared Rutter |
| Howard Hughes Medical Institute | | Jared Rutter |

The funders had no role in study design, data collection, and interpretation, or the decision to submit the work for publication.

### Author contributions

Ashish G Toshniwal, Conceptualization, Resources, Data curation, Formal analysis, Validation, Investigation, Visualization, Methodology, Writing – original draft, Project administration, Writing – review and editing; Geanette Lam, Resources, Validation, Methodology; Alex J Bott, Resources, Formal analysis, Validation, Visualization, Methodology; Ahmad A Cluntun, Formal analysis, Validation, Visualization, Methodology; Rachel Skabelund, Hyuck-Jin Nam, Dona R Wisidagama, Validation, Methodology; Carl S Thummel, Conceptualization, Resources, Supervision, Funding acquisition, Validation, Investigation; Jared Rutter, Conceptualization, Resources, Data curation, Formal analysis, Supervision, Funding acquisition, Investigation, Visualization, Methodology, Writing – original draft, Project administration, Writing – review and editing

### Author ORCIDs

Ashish G Toshniwal ⓘ https://orcid.org/0000-0001-8957-7970
Alex J Bott ⓘ https://orcid.org/0000-0003-2273-8922
Ahmad A Cluntun ⓘ https://orcid.org/0000-0001-7612-8375
Carl S Thummel ⓘ https://orcid.org/0000-0001-8112-4643
Jared Rutter ⓘ https://orcid.org/0000-0002-2710-9765

Reviewer #1 (Public review): https://doi.org/10.7554/eLife.103705.3.sa1
Reviewer #2 (Public review): https://doi.org/10.7554/eLife.103705.3.sa2
Reviewer #3 (Public review): https://doi.org/10.7554/eLife.103705.3.sa3
Author response https://doi.org/10.7554/eLife.103705.3.sa4

## Additional files

### Supplementary files
MDAR checklist

### Data availability
Novel strains (*D. melanogaster*) generated in this study, such as UAS-GOT2, Pcb KO, and Pepck2 KO, are deposited at the Bloomington Fly Stock, Indiana, US. Other reagents, such as plasmids and genetically engineered cell lines, are available upon request to the lead contact Prof. Jared Rutter. Source data to reproduce the main results of the paper, as presented in all figures, are provided.

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

# Appendix 1

## Appendix 1—key resources table

| Reagent type (species) or resource | Designation | Source or reference | Identifiers | Additional information |
|---|---|---|---|---|
| Strain, strain background (*Drosophila melanogaster*) | UAS-MPC1-P2A-MPC2 | *Schell et al., 2017* | BDSC, #84087 | Expresses *Drosophila Mpc1* and *Mpc2* cDNA separated by P2A cleavage site |
| Strain, strain background (*D. melanogaster*) | hs-Flp1.22 | Bloomington *Drosophila* Stock Center | BDSC, #77928 | |
| Strain, strain background (*D. melanogaster*) | Act >CD2>Gal4, UAS-GFP | Bloomington *Drosophila* Stock Center | BDSC, #4413 | Ay-Gal4 fly stock used to induce mosaics in fat body |
| Strain, strain background (*D. melanogaster*) | CG-Gal4 | Bloomington *Drosophila* Stock Center | BDSC, #7011 | |
| Strain, strain background (*D. melanogaster*) | UAS-S6k$^{STDETE}$ | Bloomington *Drosophila* Stock Center | BDSC, #6914 | Drives constitutively active *S6k* expression |
| Strain, strain background (*D. melanogaster*) | UAS-Rheb$^{PA}$ | Bloomington *Drosophila* Stock Center | BDSC, #9689 | Drives Rheb overexpression |
| Strain, strain background (*D. melanogaster*) | UAS-Myc$^{OE}$ | Bloomington *Drosophila* Stock Center | BDSC, #9675 | Drives *Myc* overexpression |
| Strain, strain background (*D. melanogaster*) | UAS-JF01761 | Bloomington *Drosophila* Stock Center | BDSC, #25783 | Drives Myc dsRNA, used as UAS-Myc$^{RNAi}$ |
| Strain, strain background (*D. melanogaster*) | UAS-PI3K93E.$^{Excel}$ | Bloomington *Drosophila* Stock Center | BDSC, #8287 | Drives *PI3K93* overexpression |
| Strain, strain background (*D. melanogaster*) | UAS-Pi3K92E.A2860C | Bloomington *Drosophila* Stock Center | BDSC, #8288 | Drives *PI3K93* dominant negative |
| Strain, strain background (*D. melanogaster*) | UAS-Tsc1$^{RNAi}$ | Bloomington *Drosophila* Stock Center | BDSC, #31314 | Drives *Tsc1* dsRNA, used as UAS-Tsc1$^{RNAi}$ |
| Strain, strain background (*D. melanogaster*) | UAS-hPC | Bloomington *Drosophila* Stock Center | BDSC, #77928 | Drives human PC cDNA |
| Strain, strain background (*D. melanogaster*) | UAS-HMC04104 | Bloomington *Drosophila* Stock Center | BDSC, #56883 | Drives *Pc* dsRNA, used as UAS-PC$^{RNAi}$ |
| Strain, strain background (*D. melanogaster*) | UAS-HMS00200 | Bloomington *Drosophila* Stock Center | BDSC, #36915 | Drives *Pepck2* dsRNA, used as UAS-Pepck2$^{RNAi}$ |
| Strain, strain background (*D. melanogaster*) | UAS-HMC03445 | Bloomington *Drosophila* Stock Center | BDSC, #51871 | Drives *Fbp* dsRNA, used as UAS-Fbp$^{RNAi}$ |
| Strain, strain background (*D. melanogaster*) | UAS-GL00009 | Bloomington *Drosophila* Stock Center | BDSC, #35142 | Drives *Pdk* dsRNA, used as UAS-Pdk$^{RNAi}$ |
| Strain, strain background (*D. melanogaster*) | UAS-NMNAT | Bloomington *Drosophila* Stock Center | BDSC, #39702 | Drives *Nmnat* cDNA |
| Strain, strain background (*D. melanogaster*) | UAS-CintNDX | Bloomington *Drosophila* Stock Center | BDSC, #93883 | Drives *Cionia intestinalis* NDX cDNA |
| Strain, strain background (*D. melanogaster*) | UAS-ScerNDI1 | Bloomington *Drosophila* Stock Center | BDSC, #93878 | Drives *Saccharomyces cerevisiae* NDI cDNA |
| Strain, strain background (*D. melanogaster*) | UAS-slif | Bloomington *Drosophila* Stock Center | BDSC, #52661 | Drives *slimfast* cDNA |
| Strain, strain background (*D. melanogaster*) | UAS-HMS05873 | Bloomington *Drosophila* Stock Center | BDSC, #78778 | Drives *Got2* dsRNA, used as UAS-Got2$^{RNAi}$ |
| Strain, strain background (*D. melanogaster*) | UAS-Mpc1$^{RNAi}$ | *Bricker et al., 2012* | | Drives *Mpc1* dsRNA |
| Strain, strain background (*D. melanogaster*) | UAS-Pdha$^{RNAi}$ | National Institute of Genetics, Japan | NIG 7010 R-3 | Drives *Pdha* dsRNA |
| Strain, strain background (*D. melanogaster*) | UAS-Dlat$^{RNAi}$ | National Institute of Genetics, Japan | NIG 5261 R-3 | Drives *Dlat* dsRNA |
| Strain, strain background (*D. melanogaster*) | w$^{1118}$ | Bloomington *Drosophila* Stock Center | BDSC:3605 | Wild-type fly strain |

*Appendix 1 Continued on next page*

*Appendix 1 Continued*

| Reagent type (species) or resource | Designation | Source or reference | Identifiers | Additional information |
|---|---|---|---|---|
| Strain, strain background (*D. melanogaster*) | *UAS-GOT2* | This paper | see materials and methods | Drives *Got2* cDNA, available at Bloomington *Drosophila* Stock Center |
| Strain, strain background (*D. melanogaster*) | *Pcb KO* | This paper | see materials and methods | *Pcb CRISPR KO*, available at Bloomington *Drosophila* Stock Center |
| Strain, strain background (*D. melanogaster*) | *Pepck2 KO* | This paper | see materials and methods | *Pepck2 CRISPR KO*, available at Bloomington *Drosophila* Stock Center |
| Cell line (*Homo sapiens*) | HepG2 | ATCC | Cat# HB-8065 | a hepatocellular carcinoma cell line |
| Cell line (*Rattus rattus*) | Cryopreserved Rat Primary Hepatocytes | Lonza | Cat# RWCP01 | Plateable, Rat Wistar Hannover Hepatocytes, cryopreserved |
| Antibody | anti-puromycin [3RH11] (host: mouse monoclonal) | Kerafast | Cat# EQ0001 | Dilution factor 1:1000 for western blot 1:200 for immunofluorescence |
| Antibody | anti-MPC1 (*Drosophila*) (host: rabbit monoclonal) | gift from R. Kletzein | | Dilution factor 1:200 for immunofluorescence |
| Antibody | anti-MPC2 (*Drosophila*) (host: rabbit monoclonal) | gift from R. Kletzein | | Dilution factor 1:200 for immunofluorescence |
| Antibody | anti-p-S6 (host: rabbit monoclonal) | gift from A. Teleman | | Dilution factor 1:200 for immunofluorescence |
| Antibody | anti-dFoxo (host: rabbit monoclonal) | gift from P. Bellosta | | Dilution factor 1:500 for immunofluorescence |
| Antibody | anti-p-4EBP (host: rabbit monoclonal) | Cell Signaling Technologies | Cat# 2855 | Dilution factor 1:1000 for western blot 1:500 for immunofluorescence |
| Antibody | anti-p-eIF2α (host: rabbit monoclonal) | Cell Signaling Technologies | Cat# 9721 | Dilution factor 1:500 for immunofluorescence |
| Antibody | Cy3 conjugated anti-rabbit (host: donkey polyclonal) | Jacksons Immuno Research Laboratories | Cat# 711–165–152 | Dilution factor 1:400 for immunofluorescence |
| Antibody | Cy3 conjugated anti-mouse (host: donkey polyclonal) | Jacksons Immuno Research Laboratories | Cat# 115-165-166 | Dilution factor 1:400 for immunofluorescence |
| Antibody | anti-MPC1 (host: rabbit monoclonal) | Cell Signaling Technologies | Cat# 14462 | Dilution factor 1:1000 for western blot |
| Antibody | anti-MPC2 (host: rabbit monoclonal) | Cell Signaling Technologies | Cat# 46141 | Dilution factor 1:1000 for western blot |
| Antibody | anti-PDH (host: rabbit monoclonal) | Cell Signaling Technologies | Cat# 3205 | Dilution factor 1:1000 for western blot |
| Antibody | anti- p-PDH (host: rabbit monoclonal) | Cell Signaling Technologies | Cat# 31866 | Dilution factor 1:1000 for western blot |
| Antibody | anti-PC (host: rabbit monoclonal) | Cell Signaling Technologies | Cat# 66470 | Dilution factor 1:1000 for western blot |
| Antibody | anti-PEPCK2 (D3E11) (host: rabbit monoclonal) | Cell Signaling Technologies | Cat# 8565 | Dilution factor 1:1000 for western blot |
| Antibody | anti-Got2 (host: rabbit monoclonal) | Sigma | Cat# HPA018139 | Dilution factor 1:1000 for western blot |
| Antibody | anti-tubulin (DM1A) (host: mouse monoclonal) | Cell Signaling Technologies | Cat# 3873 | Dilution factor 1:1000 for western blot |
| Antibody | anti- Flag M2 (host: mouse monoclonal) | Sigma-Aldrich | Cat# F1800 | Dilution factor 1:10000 for western blot |
| Antibody | IRDye 680RD anti-mouse (host: Donkey monoclonal) | Li-COR | Cat# 926–68072 | Dilution factor 1:5000 for western blot |
| Antibody | IRDye 800RD anti-Rabbit (host: Donkey monoclonal) | Li-COR | Cat# 926–32213 | Dilution factor 1:5000 for western blot |
| Recombinant DNA reagent | pUAST-aatB | | | |
| Recombinant DNA reagent | pLVX-TetOne-Zeocin | Takara | | |

*Appendix 1 Continued on next page*

*Appendix 1 Continued*

| Reagent type (species) or resource | Designation | Source or reference | Identifiers | Additional information |
|---|---|---|---|---|
| Recombinant DNA reagent | Gag-Pol | Addgene | | |
| Recombinant DNA reagent | VSVG | Addgene | | |
| Recombinant DNA reagent | pMD2.G | Addgene | | |
| Recombinant DNA reagent | pLenti-CMV-Blast | Addgene | Cat# 17486 | |
| Recombinant DNA reagent | pLVX-TetOne- HA-MPC2-P2A-T2A-MPC1-FLAG-Zeo | This paper | HA-tagged MPC2 and Flag-tagged MPC1 cDNA separated by P2A/T2A cleavage site cloned into pLenti-CMV-Blast | Drives *Got2* cDNA, available at Bloomington *Drosophila* Stock Center |
| Recombinant DNA reagent | pLenti-CMV-PC-V5-Blast | This paper | V5-tagged PC cDNA cloned into pLenti-CMV-Blast | Drives human PC cDNA tagged with V5, available at request to lead contact |
| Recombinant DNA reagent | pLenti-CMV-GOT2-V5-Blast | This paper | V5-tagged GOT2 cDNA cloned into pLenti-CMV-Blast | Drives human GOT2 cDNA tagged with V5, available at request to lead contact |
| Recombinant DNA reagent | PB-CAG-GFPd2 | Addgene | Cat# 115665 | |
| Recombinant DNA reagent | PMXS-NDI | Addgene | Cat# 72876 | |
| Recombinant DNA reagent | pLenti-CMV-d2GFP-Blast | This paper | d2GFP cloned into pLenti-CMV-Blast | Drives human d2GFP cDNA, available at request to lead contact |
| Recombinant DNA reagent | pLenti-CMV-NDI-V5-Blast | This paper | V5-tagged NDI cloned into pLenti-CMV-Blast | Drives yeast NDI cDNA tagged with V5, available at request to lead contact |
| Recombinant DNA reagent | pLenti-CMV-SLC1A3-V5-Blast | This paper | V5-tagged SLC1A3 cDNA cloned into pLenti-CMV-Blast; see materials and methods | Drives human SLC1A3 cDNA tagged with V5, available at request to lead contact |
| Recombinant DNA reagent | lentiCRISPRv2-blasticidin | Addgene | Cat# 83480 | |
| Recombinant DNA reagent | lentiCRISPRv2-hPCg5e-blast | This paper | sgRNA targeting human PC exon 5 in lentiCRISPRv2 vector; see Materials and methods | Drives sgRNA targeting human PC exon 5, available at request to lead contact |
| Recombinant DNA reagent | lentiCRISPRv2-hPCg6e-blast | This paper | sgRNA targeting human PC exon 6 in lentiCRISPRv2 vector; see Materials and methods | Drives sgRNA targeting human PC exon 6, available at request to lead contact |
| Recombinant DNA reagent | lentiCRISPRv2- hPEPCK2g2e-blast | This paper | sgRNA targeting human PEPCK2 exon 2 in lentiCRISPRv2 vector; see Materials and methods | Drives sgRNA targeting human PEPCK2 exon 2, available at request to lead contact |
| Recombinant DNA reagent | lentiCRISPRv2- hPEPCK2g3e-blast | This paper | sgRNA targeting human PEPCK2 exon 3 in lentiCRISPRv2 vector; see Materials and methods | Drives sgRNA targeting human PEPCK2 exon 3, available at request to lead contact |
| Recombinant DNA reagent | lentiCRISPRv2- hGOT2ga-blast | This paper | sgRNA targeting human GOT2 in lentiCRISPRv2 vector; see Materials and methods | Drives sgRNA targeting human GOT2, available at request to lead contact |
| Recombinant DNA reagent | lentiCRISPRv2- hGOT2gb-blast | This paper | sgRNA b targeting human GOT2 in lentiCRISPRv2 vector; see Materials and methods | Drives sgRNA targeting human GOT2, available at request to lead contact |
| Recombinant DNA reagent | pUAST-aatB-*dGOT2* | This paper | *dGOT2* cDNA was cloned into pLenti-CMV-Blast | Drives human GOT2, available at request to lead contact |
| Commercial assay or kit | Annexin V/PI staining kit | Molecular Probes | Cat# V13241 | |
| Commercial assay or kit | Click-iT Plus OPP Alexa Fluor 594 kit | Molecular Probes | Cat# C10457 | |
| Commercial assay or kit | Click-iT HPG Alexa Fluor 594 kit | Molecular Probes | Cat# C10429 | |
| Commercial assay or kit | Triglycerides Reagent | Thermo Fisher Scientific | Cat# TR22421 | |
| Commercial assay or kit | BCA Protein Assay Reagent | Thermo Fisher Scientific | Cat# 23225 | |
| Commercial assay or kit | TRIzol Reagent | Thermo Fisher Scientific | Cat# 15596026 | |
| Commercial assay or kit | Amplite Fluorimetric NAD/NADH ratio assay kit | AAT Bioquest | Cat# 15263 | |
| Commercial assay or kit | Click-iT Plus EdU Alexa Fluor 594 | Molecular Probes | Cat# C10639 | |

*Appendix 1 Continued*

| Reagent type (species) or resource | Designation | Source or reference | Identifiers | Additional information |
| --- | --- | --- | --- | --- |
| Commercial assay or kit | NucleoSpin RNA kit | Takara Bio USA, Inc | Cat# 740955.50 | |
| Commercial assay or kit | Autokit Glucose reagent | Wako | Cat# 997–03001 | |
| Commercial assay or kit | Amplex Red Glucose Assay Kit | Thermo Fisher Scientific | Cat# A22189 | |
| Chemical compound, drug | paraformaldehyde | Sigma-Aldrich | Cat# P6148 | |
| Chemical compound, drug | Rhodamine Phalloidin | Thermo Scientific | Cat# R418 | |
| Chemical compound, drug | DAPI-supplemented VectaShield | Vector Labs | Cat# H1200 | |
| Chemical compound, drug | Normal Goat Serum | Jackson ImmunoResearch Laboratories | Cat# 005-000-121 | |
| Chemical compound, drug | Human Plasma-Like Medium | Gibco | Cat# 765090 | |
| Chemical compound, drug | doxycycline | Sigma-Aldrich | Cat# #D5207 | |
| Chemical compound, drug | UK5099 | Sigma-Aldrich | Cat# PZ0160-5MG | |
| Chemical compound, drug | glycine | Sigma-Aldrich | Cat# G7403 | |
| Chemical compound, drug | alanine | Sigma-Aldrich | Cat# A7469 | |
| Chemical compound, drug | asparagine | Sigma-Aldrich | Cat# A4159 | |
| Chemical compound, drug | aspartic acid | Sigma-Aldrich | Cat# A7219 | |
| Chemical compound, drug | glutamic acid | Sigma-Aldrich | Cat# G8415 | |
| Chemical compound, drug | proline | Sigma-Aldrich | Cat# P5607 | |
| Chemical compound, drug | serine | Sigma-Aldrich | Cat# S4311 | |
| Chemical compound, drug | AZD7545 | MedChemExpress, | Cat# HY-16082 | |
| Chemical compound, drug | dichloroacetate | Sigma-Aldrich | Cat# 347795 | |
| Chemical compound, drug | duroquinone | Sigma-Aldrich | Cat# D223204 | |
| Chemical compound, drug | gramicidin | Sigma-Aldrich | Cat# G5002 | |
| Chemical compound, drug | nicotinamide riboside | Sigma-Aldrich | Cat# SMB00907 | |
| Chemical compound, drug | MEM non-essential amino acids | Gibco | Cat# 11140050 | |
| Chemical compound, drug | Nicotinamide mononucleotide | Sigma-Aldrich | Cat# N3501 | |
| Chemical compound, drug | Vybrant DyeCycle Violet stain | Molecular Probes | Cat# V35003 | |
| Chemical compound, drug | Eagle's Minimal Essential Medium | ATCC | Cat# 30–2003 | |
| Chemical compound, drug | Camptothecin | MedChemExpress | Cat# HY-16560 | |
| Chemical compound, drug | Shields Sang M3 Insect Media | Sigma-Aldrich | Cat# S8398 | |
| Chemical compound, drug | puromycin | Sigma-Aldrich | Cat# P4512 | |
| Chemical compound, drug | methionine-free DMEM | Gibco | Cat# 21013024 | |
| Chemical compound, drug | LipidTOX Red | Invitrogen | Cat# H34351 | |
| Chemical compound, drug | SuperScript II Reverse Transcriptase | Molecular Probes | Cat# 18064–022 | |
| Chemical compound, drug | PowerUp SYBR Green Master Mix | Applied Biosystems | Cat# 2828831 | |
| Chemical compound, drug | $^{13}$C Glucose | Millipore Sigma | Cat# 389374 | |
| Chemical compound, drug | DMEM, no glucose, no glutamine, and no phenol red | Gibco | Cat# A1443001 | |
| Chemical compound, drug | Sodium L-Lactate-C13 solution | Millipore Sigma | Cat# 490040 | |
| Chemical compound, drug | Hepatocyte Plating Medium | Lonza | Cat# MP100 | |
| Chemical compound, drug | Hepatocyte Basal Medium | Lonza | Cat# CC-4182 | |
| Chemical compound, drug | Lipofectamine 3000 | Invitrogen | Cat# L3000001 | |

*Appendix 1 Continued*

| Reagent type (species) or resource | Designation | Source or reference | Identifiers | Additional information |
|---|---|---|---|---|
| Chemical compound, drug | Triton X-100 | Sigma-Aldrich | Cat# X100 | |
| Chemical compound, drug | FBS | Sigma-Aldrich | Cat# F0926 | |
| Chemical compound, drug | PenStrep | Thermo Fisher Scientific | Cat# 15140 | |
| Chemical compound, drug | NP-40 | Millipore | Cat# 492018 | |
| Chemical compound, drug | sodium deoxycholate | Sigma-Aldrich | Cat# D6750 | |
| Chemical compound, drug | SDS | Sigma-Aldrich | Cat# L3771 | |
| Chemical compound, drug | EDTA | Sigma-Aldrich | Cat# E9884 | |
| Chemical compound, drug | Tris-HCl | Roche | Cat# 10812846001 | |
| Software | FIJI | NIH Image | RRID:SCR_002285 | https://fiji.sc/ |
| Software | Prism | GraphPad | RRID:SCR_002798 | https://www.graphpad.com/ |
| Software | FlowJo | FlowJo | RRID:SCR_008520 | https://www.flowjo.com/solutions/flowjo/downloads |
| Other | Ultra Low Cluster, 96 well, Ultra Low Attachment Polystyrene | Costar | Cat# 7007 | Corning 96-well Clear Round Bottom Ultra-Low Attachment Microplate used to make spheroids of HepG2 |
| Other | Dumont 5 forceps | Fine Science Tools | Cat# 11254–20 | Forceps used to dissect *Drosophila* fat bodies |

# Appendix 2

## Fly genotypes
### Figure 1

| # | Acronym | Detailed genotype |
|---|---------|-------------------|
| b–d | w[1118] | w[1118] |
| f | Control | hs-Flp 1.22; +/+; act >CD2>Gal4, UAS-GFP/+ |
| | MPC+ | hs-Flp 1.22; UAS-MPC1-P2A-MPC2/+; act >CD2>Gal4, UAS-GFP/+ |
| g | Control | hs-Flp 1.22; +/+; act >CD2>Gal4, UAS-GFP/+ |
| | MPC+ | hs-Flp 1.22; UAS-MPC1-P2A-MPC2/+; act >CD2>Gal4, UAS-GFP/+ |
| | Mpc1-KD | hs-Flp 1.22; UAS-Mpc1[RNAi]/+; act >CD2>Gal4, UAS-GFP/+ |
| h | Control | hs-Flp 1.22; +/+; act >CD2>Gal4, UAS-GFP/+ |
| | MPC+ | hs-Flp 1.22; UAS-MPC1-P2A-MPC2/+; act >CD2>Gal4, UAS-GFP/+ |
| I | Control | hs-Flp 1.22; +/+; act >CD2>Gal4, UAS-GFP/+ |
| | MPC+ | hs-Flp 1.22; UAS-MPC1-P2A-MPC2/+; act >CD2>Gal4, UAS-GFP/+ |
| | Mpc1-KD | hs-Flp 1.22; UAS-Mpc1[RNAi]/+; act >CD2>Gal4, UAS-GFP/+ |
| j | Control | hs-Flp 1.22; +/+; act >CD2>Gal4, UAS-GFP/+ |
| | MPC+ | hs-Flp 1.22; UAS-MPC1-P2A-MPC2/+; act >CD2>Gal4, UAS-GFP/+ |
| k | Control | y, w; CG-Gal4/UAS-GFP; tubGal[80ts20]/+ |
| | MPC+ | y, w; CG-Gal4/UAS-MPC1-P2A-MPC2; tubGal[80ts20]/+ |
| l and m | Control | hs-Flp 1.22; +/+; act >CD2>Gal4, UAS-GFP/+ |
| | MPC+ | hs-Flp 1.22; UAS-MPC1-P2A-MPC2/+; act >CD2>Gal4, UAS-GFP/+ |

### Figure 1—figure supplement 1

| # | Acronym | Detailed genotype |
|---|---------|-------------------|
| a-c | w[1118] | w[1118] |

### Figure 1—figure supplement 2

| # | Acronym | Detailed genotype |
|---|---------|-------------------|
| b and c | Control | y, w; CG-Gal4/UAS-GFP; tubGal[80ts20]/+ |
| | MPC+ | y, w; CG-Gal4/UAS-MPC1-P2A-MPC2; tubGal[80ts20]/+ |
| e and f | MPC+ | hs-Flp 1.22; UAS-MPC1-P2A-MPC2/+; act >CD2>Gal4, UAS-GFP/+ |
| g | Control | hs-Flp 1.22; +/+; act >CD2>Gal4, UAS-GFP/+ |
| | MPC+ | hs-Flp 1.22; UAS-MPC1-P2A-MPC2/+; act >CD2>Gal4, UAS-GFP/+ |
| h–k | Control | y, w; CG-Gal4/UAS-GFP; tubGal[80ts20]/+ |
| | MPC+ | y, w; CG-Gal4/UAS-MPC1-P2A-MPC2; tubGal[80ts20]/+ |
| l–n | Control | hs-Flp 1.22; +/+; act >CD2>Gal4, UAS-GFP/+ |
| | MPC+ | hs-Flp 1.22; UAS-MPC1-P2A-MPC2/+; act >CD2>Gal4, UAS-GFP/+ |
| o and p | Control | hs-Flp 1.22; +/+; act >CD2>Gal4, UAS-GFP/+ |
| | MPC+ | hs-Flp 1.22; UAS-MPC1-P2A-MPC2/+; act >CD2>Gal4, UAS-GFP/+ |

## Figure 2

| # | Acronym | Detailed genotype |
|---|---------|-------------------|
| | S6K+ | hs-Flp 1.22; UAS-S6K$^{STDETE}$/+; act >CD2>Gal4, UAS-GFP/+ |
| | Control | hs-Flp 1.22; +/+; act >CD2>Gal4, UAS-GFP/+ |
| b and c | MPC+ | hs-Flp 1.22; UAS-MPC1-P2A-MPC2/+; act >CD2>Gal4, UAS-GFP/+ |
| | Rheb+ | hs-Flp 1.22; +/+; act >CD2>Gal4, UAS-GFP/ UAS-Rheb$^{PA}$ |
| | Control | hs-Flp 1.22; +/+; act >CD2>Gal4, UAS-GFP/+ |
| d and e | MPC+ | hs-Flp 1.22; UAS-MPC1-P2A-MPC2/+; act >CD2>Gal4, UAS-GFP/+ |
| | Myc-KD | hs-Flp 1.22; +/+; act >CD2>Gal4, UAS-GFP/ UAS-Myc$^{RNAi}$ |
| | Control | hs-Flp 1.22; +/+; act >CD2>Gal4, UAS-GFP/+ |
| f and g | MPC+ | hs-Flp 1.22; UAS-MPC1-P2A-MPC2/+; act >CD2>Gal4, UAS-GFP/+ |
| | Control | hs-Flp 1.22; +/+; act >CD2>Gal4, UAS-GFP/+ |
| h and i | MPC+ | hs-Flp 1.22; UAS-MPC1-P2A-MPC2/+; act >CD2>Gal4, UAS-GFP/+ |

## Figure 2—figure supplement 1

| # | Acronym | Detailed genotype |
|---|---------|-------------------|
| | Control | y, w; CG-Gal4/UAS-GFP; tubGal$^{80ts20}$/+ |
| a | MPC+ | y, w; CG-Gal4/UAS-MPC1-P2A-MPC2; tubGal$^{80ts20}$/+ |
| | PI3K-DN | hs-Flp 1.22; +/+; act >CD2>Gal4, UAS-GFP/ UAS-Pi3K92E.A2860C |
| | Control | hs-Flp 1.22; +/+; act >CD2>Gal4, UAS-GFP/+ |
| b | MPC+ | hs-Flp 1.22; UAS-MPC1-P2A-MPC2/+; act >CD2>Gal4, UAS-GFP/+ |
| | Control | hs-Flp 1.22; +/+; act >CD2>Gal4, UAS-GFP/+ |
| | MPC+ | hs-Flp 1.22; UAS-MPC1-P2A-MPC2/+; act >CD2>Gal4, UAS-GFP/+ |
| | PI3K+ | hs-Flp 1.22; +/+; act >CD2>Gal4, UAS-GFP/ UAS-PI3K93E$^{.Excel}$ |
| c | MPC+, PI3K+ | hs-Flp 1.22; UAS-MPC1-P2A-MPC2/+; act >CD2>Gal4, UAS-GFP/ UAS- PI3K93E$^{.Excel}$ |
| | Control | hs-Flp 1.22; +/+; act >CD2>Gal4, UAS-GFP/+ |
| | MPC+ | hs-Flp 1.22; UAS-MPC1-P2A-MPC2/+; act >CD2>Gal4, UAS-GFP/+ |
| | Tsc1-KD | hs-Flp 1.22; +/+; act >CD2>Gal4, UAS-GFP/ UAS-Tsc1$^{RNAi}$ |
| d | MPC+, Tsc1-KD | hs-Flp 1.22; UAS-MPC1-P2A-MPC2/+; act >CD2>Gal4, UAS-GFP/ UAS-Tsc1$^{RNAi}$ |
| | Control | hs-Flp 1.22; +/+; act >CD2>Gal4, UAS-GFP/+ |
| | MPC+ | hs-Flp 1.22; UAS-MPC1-P2A-MPC2/+; act >CD2>Gal4, UAS-GFP/+ |
| | Myc+ | hs-Flp 1.22; +/+; act >CD2>Gal4, UAS-GFP/ UAS-Myc |
| e | MPC+, Myc+ | hs-Flp 1.22; UAS-MPC1-P2A-MPC2/+; act >CD2>Gal4, UAS-GFP/ UAS-Myc |
| | Control | hs-Flp 1.22; +/+; act >CD2>Gal4, UAS-GFP/+ |
| | MPC+ | hs-Flp 1.22; UAS-MPC1-P2A-MPC2/+; act >CD2>Gal4, UAS-GFP/+ |
| | Myc-KD | hs-Flp 1.22; +/+; act >CD2>Gal4, UAS-GFP/ UAS-Myc$^{RNAi}$ |
| f | MPC+, Myc-KD | hs-Flp 1.22; UAS-MPC1-P2A-MPC2/+; act >CD2>Gal4, UAS-GFP/ UAS-Myc$^{RNAi}$ |

## Figure 4

| # | Acronym | Detailed genotype |
|---|---------|-------------------|
| | MPC+ | hs-Flp 1.22; UAS-MPC1-P2A-MPC2/+; act >CD2>Gal4, UAS-GFP/+ |
| | MPC+, Pcb-KD | hs-Flp 1.22; UAS-MPC1-P2A-MPC2/+; act >CD2>Gal4, UAS-GFP/ UAS- -Pcb[RNAi] |
| e | Pcb+ | hs-Flp 1.22; +/+; act >CD2>Gal4, UAS-GFP/ UAS-hPC |
| | Control | hs-Flp 1.22; +/+; act >CD2>Gal4, UAS-GFP/+ |
| | MPC+ | hs-Flp 1.22; UAS-MPC1-P2A-MPC2/+; act >CD2>Gal4, UAS-GFP/+ |
| | Pcb+ | hs-Flp 1.22; +/+; act >CD2>Gal4, UAS-GFP/ UAS-hPC |
| | MPC+, Pcb+ | hs-Flp 1.22; UAS-MPC1-P2A-MPC2/+; act >CD2>Gal4, UAS-GFP/ UAS-hPC |
| | Pcb-KD | hs-Flp 1.22; +/+; act >CD2>Gal4, UAS-GFP/ UAS-Pcb[RNAi] |
| f | MPC+, Pcb-KD | hs-Flp 1.22; UAS-MPC1-P2A-MPC2/+; act >CD2>Gal4, UAS-GFP/ UAS--Pcb[RNAi] |
| | Control | y, w; CG-Gal4/UAS-GFP; tubGal[80ts20]/+ |
| g and h | MPC+ | y, w; CG-Gal4/UAS-MPC1-P2A-MPC2; tubGal[80ts20]/+ |
| | MPC+ | hs-Flp 1.22; UAS-MPC1-P2A-MPC2/+; act >CD2>Gal4, UAS-GFP/+ |
| n | MPC+, Pepck2-KD | hs-Flp 1.22; UAS-MPC1-P2A-MPC2/+; act >CD2>Gal4, UAS-GFP/UAS-Pepck2[RNAi] |
| | Control | hs-Flp 1.22; +/+; act >CD2>Gal4, UAS-GFP/+ |
| | MPC+ | hs-Flp 1.22; UAS-MPC1-P2A-MPC2/+; act >CD2>Gal4, UAS-GFP/+ |
| | Pepck2-KD | hs-Flp 1.22; +/+; act >CD2>Gal4, UAS-GFP/UAS-Pepck2[RNAi] |
| | MPC+, Pepck2-KD | hs-Flp 1.22; UAS-MPC1-P2A-MPC2/+; act >CD2>Gal4, UAS-GFP/UAS-Pepck2[RNAi] |
| | Fbp-KD | hs-Flp 1.22; +/+; act >CD2>Gal4, UAS-GFP/UAS-Fbp[RNAi] |
| o | MPC+, Fbp-KD | hs-Flp 1.22; UAS-MPC1-P2A-MPC2/+; act >CD2>Gal4, UAS-GFP/UAS-Fbp2[RNAi] |
| | MPC+ | hs-Flp 1.22; UAS-MPC1-P2A-MPC2/+; act >CD2>Gal4, UAS-GFP/+ |
| | MPC+, Pcb-KD | hs-Flp 1.22; UAS-MPC1-P2A-MPC2/+; act >CD2>Gal4, UAS-GFP/ UAS-Pcb[RNAi] |
| p | MPC+, Pepck2-KD | hs-Flp 1.22; UAS-MPC1-P2A-MPC2/+; act >CD2>Gal4, UAS-GFP/UAS-Pepck2[RNAi] |
| | Control | hs-Flp 1.22; +/+; act >CD2>Gal4, UAS-GFP/+ |
| | MPC+ | hs-Flp 1.22; UAS-MPC1-P2A-MPC2/+; act >CD2>Gal4, UAS-GFP/+ |
| | Pcb-KD | hs-Flp 1.22; +/+; act >CD2>Gal4, UAS-GFP/ UAS-Pcb[RNAi] |
| | MPC+, Pcb-KD | hs-Flp 1.22; UAS-MPC1-P2A-MPC2/+; act >CD2>Gal4, UAS-GFP/ UAS-Pcb[RNAi] |
| | Pepck2-KD | hs-Flp 1.22; +/+; act >CD2>Gal4, UAS-GFP/UAS-Pepck2[RNAi] |
| q | MPC+, Pepck2-KD | hs-Flp 1.22; UAS-MPC1-P2A-MPC2/+; act >CD2>Gal4, UAS-GFP/UAS-Pepck2[RNAi] |

## Figure 4—figure supplement 2

| # | Acronym | Detailed genotype |
|---|---------|-------------------|
| a–c | Control | hs-Flp 1.22; +/+; act >CD2>Gal4, UAS-GFP/+ |
| | MPC+ | hs-Flp 1.22; UAS-MPC1-P2A-MPC2/+; act >CD2>Gal4, UAS-GFP/+ |
| | Pcb-KD | hs-Flp 1.22; +/+; act >CD2>Gal4, UAS-GFP/ UAS-Pcb[RNAi] |
| | MPC+, Pcb-KD | hs-Flp 1.22; UAS-MPC1-P2A-MPC2/+; act >CD2>Gal4, UAS-GFP/ UAS--Pcb[RNAi] |
| | Pcb+ | hs-Flp 1.22; +/+; act >CD2>Gal4, UAS-GFP/ UAS-hPC |
| | MPC+, Pcb+ | hs-Flp 1.22; UAS-MPC1-P2A-MPC2/+; act >CD2>Gal4, UAS-GFP/ UAS-hPC |

*Continued on next page*

*Continued*

| # | Acronym | Detailed genotype |
|---|---------|-------------------|
| d | Control | hs-Flp 1.22; +/+; act >CD2>Gal4, UAS-GFP/+ |
| | MPC+ | hs-Flp 1.22; UAS-MPC1-P2A-MPC2/+; act >CD2>Gal4, UAS-GFP/+ |
| | Pcb-KO | hs-Flp 1.22; Pcb-KO/+; act >CD2>Gal4, UAS-GFP/+ |
| | MPC+, Pcb-KO | hs-Flp 1.22; UAS-MPC1-P2A-MPC2/Pcb KO; act >CD2>Gal4, UAS-GFP/+ |
| f–k | Control | hs-Flp 1.22; +/+; act >CD2>Gal4, UAS-GFP/+ |
| | MPC+ | hs-Flp 1.22; UAS-MPC1-P2A-MPC2/+; act >CD2>Gal4, UAS-GFP/+ |
| | MPC+, Pdha-KD | hs-Flp 1.22; UAS-MPC1-P2A-MPC2/UAS-Pdha$^{RNAi}$; act >CD2>Gal4, UAS-GFP/+ |
| | Dlat-KD | hs-Flp 1.22; UAS-Dlat$^{RNAi}$ /+; act >CD2>Gal4, UAS-GFP/ UAS-hPC |
| | MPC+, Dlat-KD | hs-Flp 1.22; UAS-MPC1-P2A-MPC2/UAS-Dlat$^{RNAi}$; act >CD2>Gal4, UAS-GFP/+ |
| | Pdk-KD | hs-Flp 1.22; +/+; act >CD2>Gal4, UAS-GFP/ UAS-Pdk$^{RNAi}$ |
| | MPC+, Pdk-KD | hs-Flp 1.22; UAS-MPC1-P2A-MPC2/+; act >CD2>Gal4, UAS-GFP/ UAS-Pdk$^{RNAi}$ |
| n–q | Control | hs-Flp 1.22; +/+; act >CD2>Gal4, UAS-GFP/+ |
| | MPC+ | hs-Flp 1.22; UAS-MPC1-P2A-MPC2/+; act >CD2>Gal4, UAS-GFP/+ |
| | Pepck2-KD | hs-Flp 1.22; +/+; act >CD2>Gal4, UAS-GFP/UAS-Pepck2$^{RNAi}$ |
| | MPC+, Pepck2-KD | hs-Flp 1.22; UAS-MPC1-P2A-MPC2/+; act >CD2>Gal4, UAS-GFP/UAS-Pepck2$^{RNAi}$ |
| | Fbp-KD | hs-Flp 1.22; +/+; act >CD2>Gal4, UAS-GFP/UAS-Fbp$^{RNAi}$ |
| | MPC+, Fbp-KD | hs-Flp 1.22; UAS-MPC1-P2A-MPC2/+; act >CD2>Gal4, UAS-GFP/UAS-Fbp2$^{RNAi}$ |
| r | Control | hs-Flp 1.22; +/+; act >CD2>Gal4, UAS-GFP/+ |
| | MPC+ | hs-Flp 1.22; UAS-MPC1-P2A-MPC2/+; act >CD2>Gal4, UAS-GFP/+ |
| | Pepck2-KO | hs-Flp 1.22; Pepck2-KO/+; act >CD2>Gal4, UAS-GFP/UAS-Pepck2$^{RNAi}$ |
| | MPC+, Pepck2-KO | hs-Flp 1.22; UAS-MPC1-P2A-MPC2/Pepck2-KO; act >CD2>Gal4, UAS-GFP/UAS-Pepck2$^{RNAi}$ |

## Figure 5

| # | Acronym | Detailed genotype |
|---|---------|-------------------|
| | Control | y, w; CG-Gal4/UAS-GFP; tubGal$^{80ts20}$/+ |
| a | MPC+ | y, w; CG-Gal4/UAS-MPC1-P2A-MPC2; tubGal$^{80ts20}$/+ |
| | MPC+ | hs-Flp 1.22; UAS-MPC1-P2A-MPC2/+; act >CD2>Gal4, UAS-GFP/+ |
| c | MPC+, Nmnat+ | hs-Flp 1.22; UAS-MPC1-P2A-MPC2/UAS-Nmnat; act >CD2>Gal4, UAS-GFP/+ |
| | Control | hs-Flp 1.22; +/+; act >CD2>Gal4, UAS-GFP/+ |
| | MPC+ | hs-Flp 1.22; UAS-MPC1-P2A-MPC2/+; act >CD2>Gal4, UAS-GFP/+ |
| | Nmnat+ | hs-Flp 1.22; UAS-Nmnat/+; act >CD2>Gal4, UAS-GFP/+ |
| d | MPC+, Nmnat+ | hs-Flp 1.22; UAS-MPC1-P2A-MPC2/UAS-Nmnat; act >CD2>Gal4, UAS-GFP/+ |
| | MPC+ | hs-Flp 1.22; UAS-MPC1-P2A-MPC2/+; act >CD2>Gal4, UAS-GFP/+ |
| | MPC+, NDI+ | hs-Flp 1.22; UAS-MPC1-P2A-MPC2/+; act >CD2>Gal4, UAS-GFP/ UAS-NDI |
| e | MPC+, NDX+ | hs-Flp 1.22; UAS-MPC1-P2A-MPC2/+; act >CD2>Gal4, UAS-GFP/ UAS-NDX |

*Continued on next page*

*Continued*

| # | Acronym | Detailed genotype |
|---|---------|-------------------|
|   | Control | hs-Flp 1.22; +/+; act >CD2>Gal4, UAS-GFP/+ |
|   | MPC+ | hs-Flp 1.22; UAS-MPC1-P2A-MPC2/+; act >CD2>Gal4, UAS-GFP/+ |
|   | NDI+ | hs-Flp 1.22; +/+; act >CD2>Gal4, UAS-GFP/ UAS-NDI |
|   | MPC+, NDI+ | hs-Flp 1.22; UAS-MPC1-P2A-MPC2/+; act >CD2>Gal4, UAS-GFP/ UAS-NDI |
|   | NDX+ | hs-Flp 1.22; +/+; act >CD2>Gal4, UAS-GFP/ UAS-NDX |
| f | MPC+, NDX+ | hs-Flp 1.22; UAS-MPC1-P2A-MPC2/+; act >CD2>Gal4, UAS-GFP/ UAS-NDX |
|   | MPC+ | hs-Flp 1.22; UAS-MPC1-P2A-MPC2/+; act >CD2>Gal4, UAS-GFP/+ |
| i | MPC+, NDX+ | hs-Flp 1.22; UAS-MPC1-P2A-MPC2/+; act >CD2>Gal4, UAS-GFP/ UAS-NDX |
|   | Control | hs-Flp 1.22; +/+; act >CD2>Gal4, UAS-GFP/+ |
|   | MPC+ | hs-Flp 1.22; UAS-MPC1-P2A-MPC2/+; act >CD2>Gal4, UAS-GFP/+ |
|   | NDX+ | hs-Flp 1.22; +/+; act >CD2>Gal4, UAS-GFP/ UAS-NDX |
| j | MPC+, NDX+ | hs-Flp 1.22; UAS-MPC1-P2A-MPC2/+; act >CD2>Gal4, UAS-GFP/ UAS-NDX |

## Figure 6

| # | Acronym | Detailed genotype |
|---|---------|-------------------|
| d | MPC+ | hs-Flp 1.22; UAS-MPC1-P2A-MPC2/+; act >CD2>Gal4, UAS-GFP/+ |
| e | Control | hs-Flp 1.22; +/+; act >CD2>Gal4, UAS-GFP/+ |
|   | MPC+ | hs-Flp 1.22; UAS-MPC1-P2A-MPC2/+; act >CD2>Gal4, UAS-GFP/+ |
| f | MPC+ | hs-Flp 1.22; UAS-MPC1-P2A-MPC2/+; act >CD2>Gal4, UAS-GFP/+ |
|   | MPC+, slif+ | hs-Flp 1.22; UAS-MPC1-P2A-MPC2/+; act >CD2>Gal4, UAS-GFP/ UAS-slif |
| g | Control | hs-Flp 1.22; +/+; act >CD2>Gal4, UAS-GFP/+ |
|   | MPC+ | hs-Flp 1.22; UAS-MPC1-P2A-MPC2/+; act >CD2>Gal4, UAS-GFP/+ |
|   | slif+ | hs-Flp 1.22; +/+; act >CD2>Gal4, UAS-GFP/ UAS-slif |
|   | MPC+, slif+ | hs-Flp 1.22; UAS-MPC1-P2A-MPC2/+; act >CD2>Gal4, UAS-GFP/ UAS-slif |
| i | MPC+ | hs-Flp 1.22; UAS-MPC1-P2A-MPC2/+; act >CD2>Gal4, UAS-GFP/+ |
|   | Got2-KD | hs-Flp 1.22; +/+; act >CD2>Gal4, UAS-GFP/ UAS-Got2$^{RNAi}$ |
|   | MPC+, Got2+ | hs-Flp 1.22; UAS-MPC1-P2A-MPC2/+; act >CD2>Gal4, UAS-GFP/UAS-Got2 |
| j | Control | hs-Flp 1.22; +/+; act >CD2>Gal4, UAS-GFP/+ |
|   | MPC+ | hs-Flp 1.22; UAS-MPC1-P2A-MPC2/+; act >CD2>Gal4, UAS-GFP/+ |
|   | Got2-KD | hs-Flp 1.22; +/+; act >CD2>Gal4, UAS-GFP/ UAS-Got2$^{RNAi}$ |
|   | MPC+, Got2-KD | hs-Flp 1.22; UAS-MPC1-P2A-MPC2/+; act >CD2>Gal4, UAS-GFP/ UAS-Got2$^{RNAi}$ |
|   | Got2+ | hs-Flp 1.22; +/+; act >CD2>Gal4, UAS-GFP/UAS- Got2+ |
|   | MPC+, Got2+ | hs-Flp 1.22; UAS-MPC1-P2A-MPC2/+; act >CD2>Gal4, UAS-GFP/UAS- Got2+ |

## Appendix 3

### Oligos for *Drosophila* genes

| Name of gene | Sequence | Additional information |
| --- | --- | --- |
| rp49 | GACGCTTCAAGGGA CAGTATCTG | QPCR- forward primer |
| rp49 | AAACGCGGTTCTGCATGA | QPCR- reverse primer |
| Pyk | TCTTGGTGACTGGCTGAAGG | QPCR- forward primer |
| Pyk | GCCGTTCTTCTTTCCGAC | QPCR- reverse primer |
| Mpc1 | CTCAAAGGAGTGGCGGGATT | QPCR- forward primer |
| Mpc1 | CAGGGTCAGAGCCAATGTCA | QPCR- reverse primer |
| Mpc2 | CAGCTGGTCCCAAGACGATA | QPCR- forward primer |
| Mpc2 | CGCATCCAGACACGGAGAT | QPCR- reverse primer |
| Pdha | ATCATCTCGGCGTACCGTG | QPCR- forward primer |
| Pdha | GCCTCCGTAGAAGTTCGGTG | QPCR- reverse primer |
| Dlat | CTGGAGTCCAAGACACAACTG | QPCR- forward primer |
| Dlat | TGAAGTCGTTTACAGAGACGC | QPCR- reverse primer |
| Pepck | TGATCCCGAACGCACCATC | QPCR- forward primer |
| Pepck | CTCAGGGCGAAGCACTTCTT | QPCR- reverse primer |
| Pepck2 | AATGCTGGGTAACTGGATAGCC | QPCR- forward primer |
| Pepck2 | GGTGCGACCTTTCATGCAG | QPCR- reverse primer |
| Pc-KO | ATACATTTAAGTCCTAGGC | CRISPR 5′ guide |
| Pc-KO | TCGATTGATCCTGGAAACA | CRISPR 3′ guide |
| Pepck2-KO | AAAGGGTGCACATCTGTGA | CRISPR 5′ guide |
| Pepck2-KO | TTTGGGGCGTGGCCTAGAC | CRISPR 3′ guide |
| Got2 | GAATTC ATGAGTAGAACCATTATTATGACGCTTAAGGAC | Primer for cDNA clone |
| Got2 | CTCGAG CTTGGTAACCTTGTGTATGCTCTCAGCCAGG | Primer for cDNA clone |

### Oligos for human genes

| Name of gene | Sequence | Additional information |
| --- | --- | --- |
| PC | GAATTC ATGCTGAAGTTCCGAACAGTCCATGGG | Forward primer for cDNA clone |
| PC | GGATCC CTCGATCTCCAGGATGAGGTCGTCACC | Reverse primer for cDNA clone |
| SLC1A3 | GGATCC ATGACTAAAAGCAATGGAGAAGAGC | Forward primer for cDNA clone |
| SLC1A3 | CTACATCTTGGTTTCAATGTCGATGG | Reverse primer for cDNA clone |
| GOT2 | GGATCC ATGGCCCTGCTGCACTCCGG | Forward primer for cDNA clone |
| GOT2 | TCTAGA CTTGGTGACCTGGTGAATGGCATGG | Reverse primer for cDNA clone |
| PC | GAAGCCTATCTCATCGGCCG CGG | CRISPR guide targeting exon 5 |
| PC | CGAAGTCCGCTCGCTCAGAG AGG | CRISPR guide targeting exon 6 |
| PEPCK2 | ATCTCCACTAAGCACTCGCA GGG | CRISPR guide targeting exon 2 |
| PEPCK2 | CATGCGTATTATGACCCGAC TGG | CRISPR guide targeting exon 3 |
| GOT2 | GAGTGGCCGGGTAAGCTGAGCAG AGG | CRISPR guide a |
| GOT2 | GGAGTGGACCCGCTCCGGAACAG TGG | CRISPR guide b |

