## [Editor Report · eLife Assessment]

This **fundamental** work demonstrates that compartmentalized cellular metabolism is a dominant input into cell size control in a variety of mammalian cell types and in Drosophila. The authors show that increased pyruvate import into the mitochondria in liver-like cells and in primary hepatocytes drives gluconeogenesis but reduces cellular amino acid production, suppressing protein synthesis. The evidence supporting the conclusions is **compelling**, with a variety of genetic and pharmacologic assays rigorously testing each step of the proposed mechanism. This work will be of interest to cell biologists, physiologists, and researchers interested in cell metabolism, and is significant because stem cells and many cancers exhibit metabolic rewiring of pyruvate metabolism.

---

## [Referee Report · Reviewer #1 (Public review)]

Summary:

The study examines how pyruvate, a key product of glycolysis that influences TCA metabolism and gluconeogenesis, impacts cellular metabolism and cell size. It primarily utilizes the Drosophila liver-like fat body, which is composed of large post-mitotic cells that are metabolically very active. The study focuses on the key observations that over-expression of the pyruvate importer MPC complex (which imports pyruvate from the cytoplasm into mitochondria) can reduce cell size in a cell-autonomous manner. They find this is by metabolic rewiring that shunts pyruvate away from TCA metabolism and into gluconeogenesis. Surprisingly, mTORC and Myc pathways are also hyper-active in this background, despite the decreased cell size, suggesting a non-canonical cell size regulation signaling pathway. They also show a similar cell size reduction in HepG2 organoids. Metabolic analysis reveals that enhanced gluconeogenesis suppresses protein synthesis. Their working model is that elevated pyruvate mitochondrial import drives oxaloacetate production and fuels gluconeogenesis during late larval development, thus reducing amino acid production and thus reducing protein synthesis.

Strengths:

The study is significant because stem cells and many cancers exhibit metabolic rewiring of pyruvate metabolism. It provides new insights into how the fate of pyruvate can be tuned to influence Drosophila biomass accrual, and how pyruvate pools can influence the balance between carbohydrate and protein biosynthesis. Strengths include its rigorous dissection of metabolic rewiring and use of Drosophila and mammalian cell systems to dissect carbohydrate:protein crosstalk.

Comments on revised version:

The study remains an important dissection of how metabolic compartmentalization can influence cell size. It nicely uses Drosophila and a variety of metabolic approaches. The various pathway analyses and rigorous quantitation are strengths.

---

## [Referee Report · Reviewer #2 (Public review)]

In this manuscript, the authors leverage multiple cellular models including the drosophila fat body and cultured hepatocytes to investigate the metabolic programs governing cell size. By profiling gene programs in the larval fat body during the third instar stage - in which cells cease proliferation and initiate a period of cell growth - the authors uncover a coordinated downregulation of genes involved in mitochondrial pyruvate import and metabolism. Enforced expression of the mitochondrial pyruvate carrier restrains cell size, despite active signaling of mTORC1 and other pathways viewed as traditional determinants of cell size. Mechanistically, the authors find that mitochondrial pyruvate import restrains cell size by fueling gluconeogenesis through the combined action of pyruvate carboxylase and phosphoenolpyruvate carboxykinase. Pyruvate conversion to oxaloacetate and use as a gluconeogenic substrate restrains cell growth by siphoning oxaloacetate away from aspartate and other amino acid biosynthesis, revealing a tradeoff between gluconeogenesis and provision of amino acids required to sustain protein biosynthesis. Overall, this manuscript is extremely rigorous, with each point interrogated through a variety of genetic and pharmacologic assays. The major conceptual advance is uncovering the regulation of cell size as a consequence of compartmentalized metabolism, which is dominant even over traditional signaling inputs. The work has implications for understanding cell size control in cell types that engage in gluconeogenesis but more broadly raise the possibility that metabolic tradeoffs determine cell size control in a variety of contexts.

Comments on revised version:

I have had a chance to review the manuscript and response to reviewer comments. I was extremely positive about this manuscript at first submission, and thought that the manuscript rigorously reported a surprising observation with broad implications across fields. The notion that intracellular metabolic networks can be dominant determinants of cell size, even over traditional signaling inputs, is surprising and important. The authors also provide convincing mechanistic insights into how the observed metabolic changes could affect cell size regulation. I think my previous comments and summary remain applicable for the revised manuscript.

---

## [Referee Report · Reviewer #3 (Public review)]

Summary:

In this article, Toshniwal et al. investigate the role of pyruvate metabolism in controlling cell growth. They find that elevated expression of the mitochondrial pyruvate carrier (MPC) leads to decreased cell size in the Drosophila fat body, a transformed human hepatocyte cell line (HepG2), and primary rat hepatocytes. Using genetic approaches and metabolic assays, the authors find that elevated pyruvate import into cells with forced expression of MPC increases the cellular NADH/NAD+ ratio, which drives the production of oxaloacetate via pyruvate carboxylase. Genetic, pharmacological, and metabolic approaches suggest that oxaloacetate is used to support gluconeogenesis rather than amino acid synthesis in cells over-expressing MPC. The reduction in cellular amino acids impairs protein synthesis, leading to impaired cell growth.

Strengths:

This study shows that the metabolic program of a cell, and especially its NADH/NAD+ ratio, can play a dominant role in regulating cell growth.

The combination of complementary approaches, ranging from Drosophila genetics to metabolic flux measurements in mammalian cells, strengthens the findings of the paper and shows a conservation of MPC effects across evolution.

---

## [Author Response]

The following is the authors’ response to the original reviews.

**Reviewer #1 (Public review):**
The study examines how pyruvate, a key product of glycolysis that influences TCA metabolism and gluconeogenesis, impacts cellular metabolism and cell size. It primarily utilizes the Drosophila liver-like fat body, which is composed of large post-mitotic cells that are metabolically very active. The study focuses on the key observations that overexpression of the pyruvate importer MPC complex (which imports pyruvate from the cytoplasm into mitochondria) can reduce cell size in a cell-autonomous manner. They find this is by metabolic rewiring that shunts pyruvate away from TCA metabolism and into gluconeogenesis. Surprisingly, mTORC and Myc pathways are also hyper-active in this background, despite the decreased cell size, suggesting a non-canonical cell size regulation signaling pathway. They also show a similar cell size reduction in HepG2 organoids. Metabolic analysis reveals that enhanced gluconeogenesis suppresses protein synthesis. Their working model is that elevated pyruvate mitochondrial import drives oxaloacetate production and fuels gluconeogenesis during late larval development, thus reducing amino acid production and thus reducing protein synthesis.Strengths:The study is significant because stem cells and many cancers exhibit metabolic rewiring of pyruvate metabolism. It provides new insights into how the fate of pyruvate can be tuned to influence Drosophila biomass accrual, and how pyruvate pools can influence the balance between carbohydrate and protein biosynthesis. Strengths include its rigorous dissection of metabolic rewiring and use of Drosophila and mammalian cell systems to dissect carbohydrate:protein crosstalk.Weaknesses:However, questions on how these two pathways crosstalk, and how this interfaces with canonical Myc and mTORC machinery remain. There are also questions related to how this protein:carbohydrate crosstalk interfaces with lipid biosynthesis. Addressing these will increase the overall impact of the study.

We thank the reviewer for recognizing the significance of our work and for providing constructive feedback. Our findings indicate that elevated pyruvate transport into mitochondria acts independently of canonical pathways, such as mTORC1 or Myc signaling, to regulate cell size. To investigate these pathways, we utilized immunofluorescence with well-validated surrogate measures (p-S6 and p-4EBP1) in clonal analyses of MPC expression, as well as RNAseq analyses in whole fat body tissues expressing MPC. These methods revealed surprising hyperactivation of mTORC1 and Myc signaling in Drosophila fat body cells expressing MPC, which are dramatically smaller than control cells. One explanation of these seemingly contradictory observations could be an excess of nutrients that activate mTORC1 or Myc pathways. However, our data is inconsistent with a nutrient surplus that could explain this hyperactivation. Instead, we observed reduced amino acid abundance upon MPC expression, which is very surprising given the observed hyperactivation of mTORC1. This led us to hypothesize the existence of a feedback mechanism that senses an inappropriate reduction in cell size and activates signaling pathways to promote cell growth. The best-characterized “sizer” pathway for mammalian cells is the Cyclin D/CDK4 complex, which has been well studied in the context of cell size regulation of the cell cycle (PMID 10970848, 34022133). However, the mechanisms that sense cell size in post-mitotic cells, such as fat body cells and hepatocytes, remain poorly understood. Investigating the hypothesized size-sensing mechanisms at play here is a fascinating direction for future research.

For the current study, we conducted epistatic analyses with mTORC1 pathway members by overexpressing PI3K and knocking down the TORC1 inhibitor Tuberous Sclerosis Complex 1 (Tsc1). These manipulations increased the size of control fat body cells but not those overexpressing the MPC (Supplementary Fig. 3c, 3d). Regarding Myc, its overexpression increased the size of both control and MPC+ clones (Supplementary Fig. 3e), but Myc knockdown had no additional effect on cell size in MPC+ clones (Supplementary Fig. 3f). These results suggest that neither mTORC1, PI3K, nor Myc is epistatic to the cell size effects of MPC expression. Consequently, we shifted our focus to metabolic mechanisms regulating biomass production and cell size.

When analyzing cellular biomolecules contributing to biomass, we observed a significant impact on protein levels in Drosophila fat body cells and mammalian MPC-expressing HepG2 spheroids. Triglyceride abundance in MPC-expressing HepG2 spheroids and whole fat body cells showed a statistically insignificant decrease compared to controls. Furthermore, lipid droplets in fat body cells were comparable in MPC-expressing clones when normalized to cell size.

Interestingly, RNA-seq analysis revealed modestly increased expression of fatty acid and cholesterol biosynthesis pathways in MPC-expressing fat body cells. Upregulated genes included major SREBP targets, such as ATPCL (2.08-fold), FASN1 (1.15-fold), FASN2 (1.07-fold), and ACC (1.26-fold). Since mTORC1 promotes SREBP activation and MPC-expressing cells showed elevated mTOR activity and upregulation of SREBP targets, we hypothesize that SREBP is modestly activated in these cells. Nonetheless, our data on amino acid abundance and its impact on protein synthesis activity suggest that protein abundance is likely to play a prominent causal role in regulating cell size in response to increased pyruvate transport into mitochondria.

**Reviewer #2 (Public review):**
In this manuscript, the authors leverage multiple cellular models including the drosophila fat body and cultured hepatocytes to investigate the metabolic programs governing cell size. By profiling gene programs in the larval fat body during the third instar stage - in which cells cease proliferation and initiate a period of cell growth - the authors uncover a coordinated downregulation of genes involved in mitochondrial pyruvate import and metabolism. Enforced expression of the mitochondrial pyruvate carrier restrains cell size, despite active signaling of mTORC1 and other pathways viewed as traditional determinants of cell size. Mechanistically, the authors find that mitochondrial pyruvate import restrains cell size by fueling gluconeogenesis through the combined action of pyruvate carboxylase and phosphoenolpyruvate carboxykinase. Pyruvate conversion to oxaloacetate and use as a gluconeogenic substrate restrains cell growth by siphoning oxaloacetate away from aspartate and other amino acid biosynthesis, revealing a tradeoff between gluconeogenesis and provision of amino acids required to sustain protein biosynthesis. Overall, this manuscript is extremely rigorous, with each point interrogated through a variety of genetic and pharmacologic assays. The major conceptual advance is uncovering the regulation of cell size as a consequence of compartmentalized metabolism, which is dominant even over traditional signaling inputs. The work has implications for understanding cell size control in cell types that engage in gluconeogenesis but more broadly raise the possibility that metabolic tradeoffs determine cell size control in a variety of contexts.

We thank the reviewer for their thoughtful recognition of our efforts, and we are honored by the enthusiasm the reviewer expressed for the findings and the significance of our research. We share the reviewer’s opinion that our work might help to unravel metabolic mechanisms that regulate biomass gain independent of the well-known signaling pathways.

**Reviewer #3 (Public review):**
Summary:In this article, Toshniwal et al. investigate the role of pyruvate metabolism in controlling cell growth. They find that elevated expression of the mitochondrial pyruvate carrier (MPC) leads to decreased cell size in the Drosophila fat body, a transformed human hepatocyte cell line (HepG2), and primary rat hepatocytes. Using genetic approaches and metabolic assays, the authors find that elevated pyruvate import into cells with forced expression of MPC increases the cellular NADH/NAD+ ratio, which drives the production of oxaloacetate via pyruvate carboxylase. Genetic, pharmacological, and metabolic approaches suggest that oxaloacetate is used to support gluconeogenesis rather than amino acid synthesis in cells over-expressing MPC. The reduction in cellular amino acids impairs protein synthesis, leading to impaired cell growth.Strengths:This study shows that the metabolic program of a cell, and especially its NADH/NAD+ ratio, can play a dominant role in regulating cell growth.The combination of complementary approaches, ranging from Drosophila genetics to metabolic flux measurements in mammalian cells, strengthens the findings of the paper and shows a conservation of MPC effects across evolution.Weaknesses:In general, the strengths of this paper outweigh its weaknesses. However, some areas of inconsistency and rigor deserve further attention.

Thank you for reviewing our manuscript and offering constructive feedback. We appreciate your recognition of the significance of our work and your acknowledgment of the compelling evidence we have presented. We have carefully revised the manuscript in line with the reviewers' recommendations.

The authors comment that MPC overrides hormonal controls on gluconeogenesis and cell size (Discussion, paragraph 3). Such a claim cannot be made for mammalian experiments that are conducted with immortalized cell lines or primary hepatocytes.

We appreciate the reviewer’s insightful comment. Pyruvate is a primary substrate for gluconeogenesis, and our findings suggest that increased pyruvate transport into mitochondria increases the NADH-to-NAD+ ratio, and thereby elevates gluconeogenesis. Notably, we did not observe any changes in the expression of key glucagon targets, such as PC, PEPCK2, and G6PC, suggesting that the glucagon response is not activated upon MPC expression. By the statement referenced by the reviewer, we intended to highlight that excess pyruvate import into mitochondria drives gluconeogenesis independently of hormonal and physiological regulation.

It seems the reviewer might also have been expressing the sentiment that our in vitro models may not fully reflect the in vivo situation, and we completely agree. Moving forward, we plan to perform similar analyses in mammalian models to test the in vivo relevance of this mechanism. For now, we will refine the language in the manuscript to clarify this point.

Nuclear size looks to be decreased in fat body cells with elevated MPC levels, consistent with reduced endoreplication, a process that drives growth in these cells. However, acute, ex vivo EdU labeling and measures of tissue DNA content are equivalent in wild-type and MPC+ fat body cells. This is surprising - how do the authors interpret these apparently contradictory phenotypes?

We thank the reviewer for raising this important issue. The size of the nucleus is regulated by DNA content and various factors, including the physical properties of DNA, chromatin condensation, the nuclear lamina, and other structural components (PMID 32997613). Additionally, cytoplasmic and cellular volume also impact nuclear size, as extensively documented during development (PMID 17998401, PMID 32473090).

In MPC-expressing cells, it is plausible that the reduced cellular volume impacts chromatin condensation or the nuclear lamina in a way that slightly decreases nuclear size without altering DNA content. Specifically, in our whole-fat body experiments using CG-Gal4 (as shown in Supplementary Figure 2a-c), we noted that after 12 hours of MPC expression, cell size was significantly reduced (Supplementary Figure 2c and Author Response Figure 1A). However, the reduction in nuclear size is modestly different at 24 hours and significantly different at 36 hours (Author Response Figure 1B), suggesting that the reduction in cell size is a more acute response to MPC expression, followed only later by effects on nuclear size.

In clonal analyses, this relationship was further clarified. MPC-expressing cells with a size greater than 1000 µm² displayed nuclear sizes comparable to control cells, whereas those with a drastic reduction in cell size (less than 1000 µm²) exhibited smaller nuclei (Author Response Figure 1C and 1D). These observations collectively suggest that changes in nuclear size are more likely to be downstream rather than upstream of cell size reduction. Given that DNA content remains unaffected, we focused on investigating the rate of protein synthesis. Our findings suggest that protein synthesis might play a causal role in regulating cell size, thereby reinforcing the connection between cellular and nuclear size in this context.

**Author response image 1. sa4fig1:** 

Cell Size vs. Nuclear Size in MPC-Expressing Fat Body Cells A. Cell size comparison between control (blue, ay-GFP) and MPC+ (red, ay-MPC) fat body cells over time, measured in hours after MPC expression induction. B. Nuclear area measurements from the same fat body cells in ay-GFP and ay-MPC groups. C. Scatter plot of nuclear area vs. cell area for control (ay-GFP) cells, including the corresponding R^2^ value. D. Scatter plot of nuclear area vs. cell area for MPC-expressing (ay-MPC) cells, with the respective R² value.

This figure highlights the relationship between nuclear and cell size in MPC-expressing fat body cells, emphasizing the distinct cellular responses observed following MPC induction.

In Figure 4d, oxygen consumption rates are measured in control cells and those overexpressing MPC. Values are normalized to protein levels, but protein is reduced in MPC+ cells. Is oxygen consumption changed by MPC expression on a per-cell basis?

As described in the manuscript, MPC-expressing cells are smaller in size. In this context, we felt that it was most appropriate to normalize oxygen consumption rates (OCR) to cellular mass to enable an accurate interpretation of metabolic activity. Therefore, we normalized OCR with protein content to account for variations in cellular size and (probably) mitochondrial mass.

Trehalose is the main circulating sugar in Drosophila and should be measured in addition to hemolymph glucose. Additionally, the units in Figure 4h should be related to hemolymph volume - it is not clear that they are.

We appreciate this valuable suggestion. In the revised manuscript, we have quantified trehalose abundance in circulation and within fat bodies. As described in the Methods section and following the approach outlined in Ugrankar-Banerjee et al. 2023, we bled 10 larvae (either control or MPC-expressing) using forceps onto parafilm. From this, 2 microliters of hemolymph were collected for glucose measurement. The hemolymph was treated with trehalase overnight, and the resulting glucose derived from trehalose was measured. We have observed that trehalose levels were also elevated in hemolymph of fat body-specific MPC-expressing larvae, further supporting our conclusion that MPC expression in fat body induces a hyperglycemic state. These data are now included in Figure 4h of the revised manuscript, and the details are further mentioned in the revised materials and methods.

Measurements of NADH/NAD ratios in conditions where these are manipulated genetically and pharmacologically (Figure 5) would strengthen the findings of the paper. Along the same lines, expression of manipulated genes - whether by RT-qPCR or Western blotting - would be helpful to assess the degree of knockdown/knockout in a cell population (for example, Got2 manipulations in Figures 6 and S8).

We appreciate this suggestion, which will provide additional rigor to our study. We have already quantified NADH/NAD+ ratios in HepG2 cells under UK5099, NMN, and Asp supplementation, as presented in Figure 6k. As suggested, we have quantified the expression of Got2 manipulations mentioned in Figure 6j using RT-qPCR, this data is presented in revised Supplementary Figure 8f-h. In addition, Supplementary Figure 8i has been updated with western blot analysis of Got2 expression in knock-out cells used to perform the size analysis in HepG2 cells.

Additionally, we have also analysed the efficiency of pcb (Supplementary Figure 6a-c), pdha (Supplementary Figure 6f-h), dlat (Supplementary Figure 6f, g and i), pepck2 (Supplementary Figure 6n-p), fbp (Supplementary Figure 6n, m, q) manipulations used to modulate the expression of these genes. These validations will ensure the robustness of our findings and strengthen the conclusions of our study.

**Reviewer #1 (Recommendations for the authors):**
General questions:(1) MPC over-expression in HepG2 cells altered the redox balance and the NADH/NAD+ ratio. This is suggested to help drive the metabolic rewiring from protein to carbohydrate biosynthesis. In line with this overexpression of Nmnat (which makes NAD+) or NDX rescues cell size and elevates protein biosynthesis. However, mechanistically it is unclear exactly how these redox NAD+ changes directly impact protein biosynthesis. Some additional explanations will strengthen this portion of the study.

Our data indicate that the altered redox state of the cell, particularly elevated NADH levels, affects the rate of protein synthesis. A similar relationship between redox balance and protein synthesis has been observed during embryonic development (PMID: 39879975), although the underlying mechanism remains uncharacterized. Our study suggests that increased NADH levels reprogram cellular carbohydrate metabolism, shifting it from glycolysis toward gluconeogenesis. This metabolic shift necessitates the use of oxaloacetate by PEPCK2, instead of its diversion toward GTP-mediated aspartate synthesis. Aspartate, which can be anaplerotically converted into glutamate and proline, plays a critical role in protein biosynthesis. Thus, the conversion of oxaloacetate to phosphoenolpyruvate represents a key metabolic node influencing protein synthesis under altered redox conditions. Additionally, since aspartate serves as a precursor for NAD biosynthesis, this may suggest a feedforward loop reinforcing the metabolic rewiring. Nonetheless, the precise relationship between NADH concentration and redox status and the regulation of protein synthesis warrants further investigation in future studies.

(2) In the MPC1/2 (MPC+) over-expression background, can blocking of gluconeogenesis downstream in the carbohydrate synthesis pathway rescue the phenotype?

We knocked down FBPase (Drosophila fbp) using an RNAi construct, achieving approximately 60% reduction in FBPase expression in Drosophila. Notably, FBPase knockdown in fat body cells overexpressing MPC rescued the reduced cell size phenotype. These findings are presented in Figure 4o and Supplementary Figures 6n–q.

(3) Biomass accrual and cell size are also influenced by lipogenesis. The study suggests mTORC and Myc are uncoupled to cell size determination per se, but how lipogenesis regulatory pathways like SREBP are impacted by MPC overexpression is not really explored. How lipid membrane synthesis inter-relates to this protein/carbohydrate crosstalk would add to the understanding of the system.

As mentioned above - When analyzing cellular biomolecules contributing to biomass, we observed a significant impact on protein levels in Drosophila fat body cells and mammalian MPC-expressing HepG2 spheroids. Triglyceride abundance in MPC-expressing HepG2 spheroids and whole fat body cells showed a statistically insignificant decrease compared to controls. Furthermore, lipid droplets in fat body cells were comparable in MPC-expressing clones when normalized to cell size.

Interestingly, RNA-seq analysis revealed increased expression of fatty acid and cholesterol biosynthesis pathways in MPC-expressing fat body cells. Upregulated genes included major SREBP targets, such as ATPCL (2.08-fold), FASN1 (1.15-fold), FASN2 (1.07-fold), and ACC (1.26-fold). Since mTOR promotes SREBP activation and MPC-expressing cells showed elevated mTOR activity and upregulation of SREBP targets, we hypothesize that SREBP is modestly activated in these cells. Nonetheless, our data on amino acid abundance and its impact on protein synthesis activity suggest that protein abundance, rather than lipids, is likely to play a larger causal role in regulating cell size in response to increased pyruvate transport into mitochondria.

**Reviewer #2 (Recommendations for the authors):**
I have only minor suggestions for the authors to consider.Minor points(1) Wherever possible, scale bars should be labeled with units or indicated comparisons (e.x. Supplementary Fig. 1). To make the data as accessible as possible, it would be helpful for the authors to include the data presented in Supplementary Figure 1 as an associated table as well.

We have corrected this in the revised manuscript and included the table.

(2) To support the conclusions about TCA cycle flux (lines 280-284), it will be helpful for the authors to consider relative metabolite pool sizes (which they should have on hand) in addition to labeling rate and fraction.

We thank the reviewer for this suggestion. We have included the metabolite counts with fractional abundance changes side by side in Supplementary Figure 5.

(3) believe (?) there is a typo in lines 326-328; PEPCK KO increases (not decreases) the size of spheroids/cells.

We thank the reviewer for pointing out this error. We have corrected this in the revised manuscript.

(4) Supplementary Figure 7b: PHD has 3 phospho sites that have independent regulation; the specific phosphosite queried should be listed on the figure and unless all 3 sites are probed the claims about lack of change in phosphorylation (line 337) should be removed.

We thank the reviewer for bringing this to our attention. We have included this in the revised manuscript.

(5) (Optional) I appreciate the effort the authors undertook to acquire cytoplasmic and mitochondrial ratios of NADH/NAD. While I recognize that many labs perform this assay, it is difficult for this reviewer to envision how accurately these values reflects the ratios present in the intact cell given how quickly these redox couples interconvert and significant post-harvest metabolic flux (see for ex PMID: 31767181), even with the extremely rapid fractionation protocol described in the methods. The present data certainly support the notion that MPC+ cells are more reduced, but these ratios may reflect a capacity for reductive metabolism rather than a bona fide NADH/NAD ratio; for example, Figure 7f shows almost identical NADH/NAD ratios in the cytoplasm and mitochondria, even though these compartments are frequently considered to have (sometimes vastly) different redox states. If the authors are willing, I would support them by including a brief discussion of the caveat of this method for new readers in the field.

We agree with this important note from the reviewers. This is an important caveat of the technique that we used for these analyses. We have included a description of this caveat in the manuscript (Revised Manuscripts lines 393 to 395).

**Reviewer #3 (Recommendations for the authors):**
Minor points:(1) Line 327 - "smaller" should be "bigger".

We thank the reviewer for pointing out this error. We have corrected this in the revised manuscript.

(2) For Figure 7 - references to panels e and f in the text, and descriptions of e and f in the Figure Legend are switched with regard to the Figure itself.

We thank the reviewer for pointing out this error. We have corrected this in the revised manuscript.

(3) Line 449 - "reduced" is missing its R

We thank the reviewer for pointing out this error. We have corrected this in the revised manuscript.

(4) Some additional, careful proofreading is needed - several other punctuation errors were found.

We thank the reviewer for pointing out these errors.

We thank the reviewer for bringing this to our attention. We have conducted very careful proofreading and corrected errors.